

# Assimilation of Carbonyl Sulfide (COS) fluxes within the adjoint-based data assimilation system—Nanjing University Carbon Assimilation System (NUCAS v1.0)

Huajie Zhu[1], Mousong Wu[1]*, Fei Jiang[1,2,3,4], Michael Vossbeck[5], Thomas Kaminski[5], Xiuli Xing[1], Jun Wang[1], Weimin Ju[1], Jing M. Chen[6]

[1]International Institute for Earth System Science, Nanjing University, Nanjing, 210023, China
[2]Jiangsu Provincial Key Laboratory of Geographic Information Science and Technology, School of Geography and Ocean Science, Nanjing University, Nanjing, 210023, China
[3]Key Laboratory for Land Satellite Remote Sensing Applications of Ministry of Natural Resources, School of Geography and Ocean Science, Nanjing University, Nanjing, 210023, China
[4]Frontiers Science Center for Critical Earth Material Cycling, Nanjing University, Nanjing, 210023, China
[5]The Inversion Lab, Hamburg, Germany
[6]Department of Geography and Program in Planning, University of Toronto, ON M5S 3G3, Canada

Correspondence: Mousong Wu (mousongwu@nju.edu.cn)

**Abstract.** Modeling and predicting changes in the function and structure of the terrestrial biosphere and its feedbacks to climate change strongly depends on our ability to accurately represent interactions of the carbon and water cycles, and energy exchange. However, carbon fluxes, hydrological status and energy exchange simulated by process-based terrestrial ecosystem models are subject to significant uncertainties, largely due to the poorly calibrated parameters related to various processes. In this work, an adjoint-based data assimilation system (Nanjing University Carbon Assimilation System, NUCAS) was developed, which is capable of assimilating multiple observations to optimize process parameters of a satellite data driven ecosystem model—BEPS (Boreal Ecosystem Productivity Simulator). Data assimilation experiments were conducted to demonstrate the robustness and to investigate the feasibility and applicability of NUCAS on seven sites by assimilating the carbonyl sulfide (COS) fluxes, which were tightly related to the stomatal conductance and photosynthesis. Results showed that NUCAS is able to achieve a consistent fit to COS observations across various ecosystems. Comparing prior simulations with validation datasets, we found that the assimilation of COS can significantly improve the model performance in gross primary productivity, sensible heat, latent heat and even soil moisture. We also showed that the NUCAS is capable of constraining parameters from multiple sites simultaneously and achieving a good consistency to the single-site assimilation. Our results demonstrate that COS can provide strong constraints on parameters relevant to water, energy and carbon processes with the data assimilation system, and open new perspectives for better understanding of the ecosystem carbon, water and energy exchanges.

*Keywords*: Carbonyl sulfide; Data assimilation; Carbon cycle; Satellite-driven; Ecosystem model

## 1 Introduction

Overwhelmingly due to anthropogenic fossil fuel and carbonate emissions, as well as land use and land cover change (Arias et al., 2021), atmospheric $CO_2$ concentrations have increased at an unprecedented rate since the Industrial Revolution and the global climate has been profoundly affected. As a key component of earth system, the terrestrial biosphere has absorbed about 30% of anthropogenic $CO_2$ emissions since 1850 and has significantly mitigated climate change (Friedlingstein et al., 2022). However, in line with large-scale global warming, the structure and function of terrestrial biosphere have changed rapidly (Grimm et al., 2013; Arias et al., 2021; Moore and Schindler, 2022), which makes terrestrial carbon fluxes subject to great uncertainty (Macbean et al., 2022).



Terrestrial ecosystem models have been an important tool to investigate the net effect of complex feedback loops between the global carbon cycle and climate change (Zaehle et al., 2005; Fisher et al., 2014; Fisher and Koven, 2020). Meanwhile, with the advancement of modern observational techniques, a rapidly increasing number of satellite- and ground-based observational data have played an important role in studying the spatiotemporal distribution and mechanisms of the terrestrial ecosystem carbon fluxes (Rodell et al., 2004; Quirita et al., 2016). Various observations (Scholze et al., 2017), such as sun-induced

chlorophyll fluorescence (Schimel et al., 2015) and soil moisture (Wu et al., 2018), have been used to estimate or constrain carbon fluxes in terrestrial ecosystems. Recently, carbonyl sulfide (COS) has emerged as a promising proxy for understanding terrestrial carbon uptake and plant physiology (Sandoval-Soto et al., 2005; Montzka et al., 2007; Campbell et al., 2008; Seibt et al., 2010; Stimler et al., 2010; Stimler et al., 2011) since it is taken up by plants through the same pathway of stomatal diffusion as $CO_2$ (Goldan et al., 1988; Sandoval-Soto et al., 2005; Seibt et al., 2010) and completely removed by hydrolysis

without any back-flux in leaves under normal condition (Protoschill-Krebs et al., 1996; Stimler et al., 2010).

   Plants control the opening of leaf stomata in order to regulate the water and $CO_2$ transit during transpiration and photosynthesis (Daly et al., 2004). As an important probe for characterizing stomatal conductance, COS has shown with great potential to constrain plant photosynthesis and transpiration and to improve understanding of the water-carbon coupling (Wohlfahrt et al., 2012). A number of empirical or mechanistic COS plant uptake models (Sandoval-Soto et al., 2005; Campbell et al., 2008;

Wohlfahrt et al., 2012; Berry et al., 2013; Kooijmans et al., 2019) and soil exchange models (Kesselmeier et al., 1999; Berry et al., 2013; Launois et al., 2015; Sun et al., 2015; Whelan et al., 2016; Ogée et al., 2016; Whelan et al., 2022) have been developed to simulate COS fluxes in order to more accurately estimate gross primary productivity (GPP) as well as other key ecosystem variables. However, due to the lack of ecosystem-scale measurements of the COS flux (Wohlfahrt et al., 2012; Kooijmans et al., 2021), little experiments were conducted to systematically assess the added value of COS in simultaneously

constraining photosynthesis, transpiration and other related processes.

   Data assimilation is an approach that aims at producing physically consistent estimates of the dynamical behavior of a model by combining the information in process-based models and observational data (Liu and Gupta, 2007; Law et al., 2015). It has been widely applied in geophysics and numerical weather prediction (Tarantola, 2005). In the past few decades, substantial efforts have been put into the use of various satellite- (Knorr et al., 2010; Kaminski et al., 2012; Deng et al., 2014; Scholze et

al., 2016; Norton et al., 2018; Wu et al., 2018) and ground-based (Knorr and Heimann, 1995; Rayner et al., 2005; Santaren et al., 2007; Kato et al., 2013; Zobitz et al., 2014) observational datasets to constrain or optimize the photosynthesis, transpiration and energy-related parameters and variables of terrestrial ecosystem models via data assimilation techniques. In particular, by applying data assimilation methods to process-based models, not only can the observed dynamics of ecosystems be more accurately portrayed, but also our understanding of ecosystem processes can be deepened, with respect to their responses to

climate (Luo et al., 2011; Keenan et al., 2012; Niu et al., 2014).

   In this study, we present the newly developed adjoint-based data assimilation system NUCAS (Nanjing University Carbon Assimilation System), that is designed to assimilate multiple observational data streams including the recently promising COS flux data to improve the process-based model Boreal Ecosystem Productivity Simulator (BEPS) (Liu et al., 1997), which has been specifically developed for simulating the ecosystem COS flux with the advanced two-leaf model that is driven by satellite

observations of leaf area index (LAI).

   In this context, the main questions that we aim to answer in this paper are as follows:

   What are the main changes in the parameters through the assimilation of COS flux and which processes are constrained?

   How effective is the assimilation of COS fluxes in improving the carbon, water and energy balance for different ecosystems?

   What are the controlling factors of variability of carbon, water and energy exchange?

How robust is the NUCAS when optimizing over single-site and multiple sites simultaneously?



To achieve these objectives, COS observations across a wide range of ecosystems are assimilated into NUCAS to optimize the model parameters using the four-dimensional variational (4D-Var) data assimilation approach, and the optimization results are evaluated against in situ observations. Specifically, materials and methods used in our study are described in Sect. 2. In this section, the BEPS model and our new data assimilation system NUCAS are introduced, along with the data used and the parameters chosen to be optimized in this study. The results are presented in Sect. 3, including the fit of COS simulations to observations, the variation and impact of parameters on simulated COS, as well as the comparison and evaluation of model outputs. Sect. 4 discusses the impacts of the COS assimilation on parameters and processes related to the water-carbon cycle and energy exchange as well as the influence of uncertainty inputs, in particular of the LAI driving data on posterior parameters values. In addition, the caveats and implications of assimilating COS flux are summarized. Finally, the conclusions are laid out in Sect. 5.

## 2 Materials and Methods

### 2.1 NUCAS data assimilation system

#### 2.1.1 NUCAS framework

NUCAS is built around the generic satellite data driven ecosystem model BEPS, and applies the 4D-Var data assimilation method (Talagrand and Courtier, 1987). The BEPS model uses satellite LAI to drive the phenology dynamics and separates sunlit and shaded leaves in calculating canopy-level energy fluxes and photosynthesis. It further features detailed representations of water and energy processes (**Figure 1**). These make BEPS more advanced in representing ecosystem processes (Richardson et al., 2012) and with less parameters to be calibrated given the phenology is driven by LAI.

By assimilating the observed data, NUCAS can achieve the optimization of the model process parameters and the model state variables of BEPS in two sequential steps: First, the BEPS model is run with default parameters and the model output is combined with COS flux observations to optimize the parameters controlling photosynthesis, energy balance, hydrology and soil biogeochemical processes. Second, the posterior parameters obtained in the first step are used as input data for the second step, in which the BEPS model is re-run to obtain the posterior model variables. The schematic of the of the system is shown in **Figure 1**.

Considering model and data uncertainties, NUCAS implements a probabilistic inversion concept (Talagrand and Courtier, 1987; Tarantola, 1987; Tarantola, 2005) by using Gaussian probability density functions to combine the dynamic model and observations to obtain an estimate of the true state of the system and model parameters (Talagrand, 1997; Dowd, 2007). Hereby, we minimize the following cost function:

$$J(x) = \frac{1}{2}\left[(M_{cos}(x) - O_{cos})^T C_{cos}^{-1}(M_{cos}(x) - O_{cos}) + (x - x_0)^T C_x^{-1}(x - x_0)\right] \tag{1}$$

where M and O denotes model and observation, respectively; $x$ and $x_0$ denotes the control parameter vector and the prior control parameter vector; C denotes the uncertainty covariance matrices for observations and parameters, and both matrices are diagonal as we suppose the observation uncertainties and the parameter uncertainties to be independent (Rayner et al., 2005). This definition of the cost function contains both the mismatch between modelled and observed COS fluxes and the mismatch between the prior and current parameter values (Rayner et al., 2005).

To determine an optimal set of parameters which minimizes $J$, a gradient-based optimization algorithm (BFGS) performs an iterative search (Wu et al., 2020). In each iteration, the gradient of $J$ is calculated by applying the adjoint of the model, where the model is run backward to efficiently compute the sensitivity of $J$ and with respect to $x$ (Rayner et al., 2005), and is used to define a new search direction. The adjoint model is an efficient sensitivity analysis tool for calculating the parametric





sensitivities of complex numerical model systems (An et al., 2016). The computational cost of it is independent of the number of parameters and is in the current case comparable to 3–4 evaluations of $J$. In this study, all derivative code is generated from the model code by the automatic differentiation tool TAPENADE (Hascoët and Pascual, 2013).

Additionally, the minimization of cost function is implemented in a normalized parameter space where the parameter values are specified in multiples of their standard deviation with Gaussian priors (Kaminski et al., 2012). The model parameters are the various constants that are not influenced by the model state. Therefore, while they may change in space to reflect different

conditions and physiological mechanisms, they will not change in time (Rayner et al., 2005).

**2.1.2 BEPS basic model**

The BEPS model (Liu et al., 1997; Chen et al., 1999; Chen et al., 2012) is a process-based diagnostic model driven by remotely sensed vegetation data, including LAI, clumping index, and land cover type, as well as meteorological and soil data (Chen et al., 2019). With the consideration of coupling among terrestrial carbon, water, and nitrogen cycles (He et al., 2021), the BEPS

model now consists of photosynthesis, energy balance, hydrological, and soil biogeochemical modules (Ju et al., 2006; Liu et al., 2015). It stratifies whole canopies into sunlit and shaded leaves to calculate carbon uptake and transpiration for these two groups of leaves separately (Liu et al., 2015). For each group of leaves, the GPP is calculated by scaling Farquhar's leaf biochemical model (Farquhar et al., 1980) up to canopy-level with a new temporal and spatial scaling scheme (Chen et al., 1999), and the stomatal conductance is calculated using a modified version of the Ball–Woodrow–Berry model (Ball et al.,

1987; Ju et al., 2006). Evapotranspiration is calculated as the summation of sunlit leaf and shaded leaf transpirations, evaporation from soil and wet canopy, and sublimation from snow storage on the ground surface (Liu et al., 2003). The BEPS model stratifies the soil profile into multiple layers (five were used in this study), and simulates temperature and water content from each layer (Ju et al., 2006). The soil water content is then used to adjust stomatal conductance considering the water stress impacts (Ju et al., 2010; He et al., 2021). Over the last few decades, the BEPS model has been continuously improved

and used for a wide variety of terrestrial ecosystems (Schwalm et al., 2010; Liu et al., 2015).

The previous version of BEPS considers a total of six plant function types (PFTs) as well as eleven soil textures (see https://github.com/JChen-UToronto/BEPS_hourly_site). For NUCAS, we use the same soil texture but added four PFTs to BEPS in order to better discriminate vegetation types, especially the C4 grass and crop. Detailed information on these ten PFTs and eleven soil textures is given in **Table S1**.

**2.1.3 COS modelling**

The ecosystem COS flux, $F_{cos_{ecosystem}}$, includes both plant COS uptake $F_{cos,plant}$ and soil COS flux exchange $F_{cos,soil}$ (Whelan et al., 2016). In this study, these two components were modelled separately. The canopy-level COS plant uptake $F_{cos,plant}$ ($pmol/m^2/s$) was calculated by upscaling the resistance analog model of COS uptake (Berry et al., 2013) with the upscaling scheme (Chen et al., 1999). Specifically, considering the different responses of foliage to diffuse and direct solar radiation (Gu

et al., 2002), $F_{cos,plant}$ is calculated as:

$$F_{cos,plant} = F_{cos,sunlit}LAI_{sunlit} + F_{cos,shaded}LAI_{shaded} \qquad (2)$$

where $LAI_{sunlit}$ and $LAI_{shaded}$ are the LAI values ($m^2/m^2$) of sunlit and shaded leaves, respectively. $F_{cos,sunlit}$ and $F_{cos,shaded}$ are the leaf-level COS uptake rate ($pmol/m^2/s$) of sunlit and shaded leaves, respectively. And the leaf-level COS uptake rate $F_{cos_{leaf}}$ is calculated as:

$$F_{cos,leaf} = cos_a * \left(\frac{1.94}{g_{sw}} + \frac{1.56}{g_{bw}} + g_{cos}\right)^{-1} \qquad (3)$$



where $cos_a$ is the COS mole fraction in the bulk air. $g_{sw}$ and $g_{bw}$ are the stomatal conductance and leaf laminar boundary layer conductance to $H_2O$ vapor. $g_{cos}$ denotes the apparent conductance for COS uptake from the intercellular airspaces, combining the mesophyll conductance and the biochemical reaction rate of COS and carbonic anhydrase. It can be calculated as:

$$g_{cos} = 1.4 * 10^3 * (1.0 + 5.33 * F_{C4}) * 10^{-6} * \left(1 - e^{(-0.45*LAI)}\right) * f_{sw} * V_{cmax} \tag{4}$$

where $F_{C4}$ denotes the C4 plant flag, which takes the value of 1 when the vegetation is C4 plants and 0 otherwise. $f_{sw}$ is a parameter describing the soil water stress on stomatal conductance. $V_{cmax}$ denotes the maximum carboxylation rate.

$F_{cos,soil}$ is taken as the combination of abiotic COS flux $F_{cos,abiotic}$ and biotic COS flux $F_{cos,biotic}$ (Whelan et al., 2016).

$$F_{cos,soil} = F_{cos,abiotic} + F_{cos,biotic} \tag{5}$$

$F_{cos,abiotic}$ is described as an exponential function of the temperature of soil $T_{soil}$ (℃).

$$F_{cos,abiotic} = 0.437 * e^{0.0984*T_{soil}} \tag{6}$$

$F_{cos_{biotic}}$ is calculated according to Behrendt et al. (2014):

$$F_{cos,biotic} = F_{opt} \left(\frac{\theta_i}{\theta_{opt}}\right) * e^{-a\left(\frac{\theta_i}{\theta_{opt}} - 1\right)} \tag{7}$$

which can be rearranged to

$$a = ln\left(\frac{F_{opt}}{F_{\theta_g}}\right) * \left(ln\left(\frac{\theta_{opt}}{\theta_g}\right) + \left(\frac{\theta_g}{\theta_{opt}} - 1\right)\right)^{-1} \tag{8}$$

Here $a$ is the curve shape constant, $\theta_i$ is the soil moisture (percent volumetric water content). The maximum biotic COS uptake $F_{opt}$ and the biotic COS uptake $F_{\theta_g}$ are the COS fluxes ($pmol/m^2/s$) at optimum soil moisture $\theta_{opt}$ and $\theta_g$, and can be calculated from $T_{soil}$ using eqs. (9) and (10) respectively.

$$F_{opt} = -0.00986 * T_{soil}^2 + 0.197 * T_{soil} - 9.32 \tag{9}$$

$$F_{\theta_g} = -0.119 * T_{soil}^2 + 0.110 * T_{soil} - 1.18 \tag{10}$$

$\theta_g$ is assumed to be a constant 0.35, and $\theta_{opt}$ is assumed to be a first order function of $T_{soil}$.

$$\theta_{opt} = 0.28 * T_{soil} + 14.5 \tag{11}$$

Then ecosystem COS flux $F_{cos,ecosystem}$ can be calculated as the sum of COS plant uptake and the COS soil flux.

**2.2 Model parameters**

In this study, we optimized a total of 76 parameters belonging to BEPS, the parameters are described in **Table S3**. Of these
parameters; some are global and others differentiated by PFT or soil texture class. The prior values of the parameters are taken as model defaults which have been tuned with efforts from previous model development and validation, and the prior uncertainty of parameters is set as 25% of the prior values.

**2.3 Site description**

The NUCAS was evaluated at seven sites distributed on the Eurasian and North American continents in boreal, temperate and
subtropical regions based on field observations collected from several studies. Those sites were representative of different climate regions and land cover types (in the model represented by PFTs, and soil textures, as depicted in **Table 1**). They contained 5 of the 10 PFTs used in BEPS and 5 of the 11 soil textures. The sites comprise AT-Neu, located at an intensively managed temperate mountain grassland near the village of Neustift in the Stubai Valley, Austria (Hörtnagl et al., 2011); the Danish ICOS RI site (DK-Sor), which is dominated by European beech (Braendholt et al., 2018); the Las Majadas del Tietar
site (ES-Lma) located in western Spain with a Mediterranean savanna ecosystem (El-Madany et al., 2018); the Hyytiälä forest Station (FI-Hyy), located in Finland and is dominated by Scots Pine (Bäck et al., 2012); an agricultural soybean field



measurement site (IT-Soy) located in Italy (Spielmann et al., 2019); the Harvard Forest Environmental Monitoring Site (US-Ha1) which is dominated by red oak and red maple in Petersham, Massachusetts, USA (Urbanski et al., 2007); the Wind River Experimental Forest site (US-Wrc), located within the Gifford Pinchot National Forest in southwest Washington state, USA,

with 478 ha of preserved old growth evergreen needleleaf forest (Rastogi et al., 2018).

**2.4 Data**

The NUCAS system was driven by several temporally and spatially variant and invariant datasets. The $CO_2$ and COS mole fractions in the bulk air were assumed to be spatially invariant over the globe and to vary annually. And the other main inputs include a remotely sensed LAI dataset, a meteorological dataset and a soil dataset. Additionally, in order to conduct data

assimilation experiments and to evaluate the effectiveness of the assimilation of COS fluxes, field observations including the COS flux, GPP, sensible heat (H), latent heat (LE) and soil moisture at these sites collected at the sites were used.

**2.4.1 LAI dataset**

The LAI dataset used here are the GLOBMAP global leaf area index product (Version 3) (see GLOBMAP global Leaf Area Index since 1981 | Zenodo), the Global Land Surface Satellite (GLASS) LAI product (Version 3) (acquired from

ftp://ftp.glcf.umd.edu/) and the level-4 MODIS global LAI product (see LP DAAC - MOD15A2H (usgs.gov)). The GLOBMAP LAI product represents Leaf area index at a spatial resolution of 8 km and a temporal resolution of 8-day (Liu et al., 2012). The GLASS LAI product is generated every 8 days at a spatial resolution of 1 km (Xiao et al., 2016). And the MODIS LAI is an 8-day composite dataset with 500 m pixel size. Overall, we used GLOBMAP products for assimilation experiments as much as possible given its good performance in the BEPS applications to various cases (Chen et al., 2019).

And all of the three LAI products were used to drive the model to investigate the effect of the LAI products on the parameter optimization results. According to Spielmann et al. (2019), the GLOBMAP product had significantly underestimated the LAI at the DK-Sor site in June 2016, and we noticed it was not consistent with the vegetation phenology at ES-Lma in May 2016. Therefore, GLASS LAI was used at these two sites and the GLOBMAP product was used at the remaining five sites. In addition, these 8-days LAI data were interpolated into daily values by the nearest neighbour method.

**2.4.2 Meteorological dataset**

Standard hourly meteorological data as input for BEPS including air temperature at 2 m, shortwave radiation, precipitation, relative humidity and wind speed is available through the FLUXNET database (AT-Neu, DK-Sor, ES-Lma and FI-Hyy, see https://fluxnet.org), the AmeriFlux database (US-Ha1, US-Wrc, see https://ameriflux.lbl.gov) and the ERA5 dataset (Site AT-Neu, IT-Soy, US-Ha1 see https://cds.climate.copernicus.eu/cdsapp#!/dataset/reanalysis-era5-single-levels?tab=overview),

respectively. Since the experiments were conducted at the site scale, we used the FLUXNET and AmeriFlux data, which contains information about the downscaling of meteorological variables of the ERA-Interim reanalysis data product (Pastorello et al., 2020) as far as possible, and supplemented them with ERA5 reanalysis data. Particularly, although AT-Neu is a FLUXNET site, its FLUXNET meteorological data are only available for the years 2002-2012 while the measurement of COS was performed in 2015. Therefore, we first performed a linear fit of its ERA5-Land data and FLUXNET meteorological data

for 2002-2012, and then corrected the ERA5 data for 2015 with the fitted parameters to obtain downscaling information for the meteorological variables. In addition, the shortwave radiation data of US-Ha1 were also derived from ERA5 since there are no in situ radiation measurements at this site.



### 2.4.3 Assimilation and evaluation datasets

The hourly COS flux observations were used to perform data assimilation experiments and to evaluate the assimilation results.

They were taken from existing studies (listed in **Table 1**) and available for at least a month. Most of the ecosystem COS flux observations were obtained using the eddy-covariance (EC) technique, with the exception US-Wrc, where the COS fluxes were derived with the gradient-based approach. We then corrected the COS fluxes from FI-Hyy using the storage-correction method (Kooijmans et al., 2017). The COS soil measurements were collected using soil chamber, except at US-Ha1, where a sub-canopy flux-gradient approach was used to calculate the soil COS flux. Detailed information on the observations of COS

can be found in the publications listed in **Table 1**.

Since only the raw COS concentration data at different altitudes are provided in Rastogi et al. (2018), while the values of the parameters needed to calculate the COS fluxes by the aerodynamic gradient method are not provided, there may be significant biases in our estimates of COS fluxes at US-Wrc. Therefore, a bias correction scheme was implemented to match the simulated and estimated COS fluxes for the US-Wrc site. The objectives of this correction scheme are to obviate the need for accurate

values of parameters relevant for COS flux calculations, and to retain as much useful information from the COS concentration measurements as possible (Leung et al., 1999; Scholze et al., 2016). This was done by using the mean and standard deviation of the simulated COS flux to correct the COS flux observations:

$$C = \frac{\sigma_s(c - m_c)}{\sigma_c} + m_s \tag{12}$$

where c denotes the COS flux observations (converted to $pmol/m^2/s$). $m_c$ and $\sigma_c$ are mean and standard deviation of the

observed COS flux series. C is the corrected observed COS flux, which is matched to the simulated COS flux. $m_s$ and $\sigma_s$ are mean and standard deviation of the COS simulations, calculated from the simulations using the prior parameters for the time period corresponding to the COS flux observations.

Considering that COS soil fluxes are much lower than the anticipated plant fluxes in general (positive values indicate COS uptake) and that the relative uncertainty in COS fluxes is very large at low values, especially when negative (Kohonen et al.,

2020), we first removed the negative values of the ecosystem COS fluxes. Then, the standard deviation of the ecosystem COS fluxes within 24 hours around each observation was calculated as estimate of the observation uncertainty. For the case where there are no other observations within the surrounding 24 hours, the uncertainty was taken as the mean of the estimated uncertainties of the whole observation series.

In order to evaluate the assimilation results, gross primary productivity, sensible heat, latent heat and volumetric soil water

content (SWC) observations were also taken from FLUXNET (DK-Sor, ES-Lma and FI-Hyy), AmeriFlux (US-Ha1 and US-Wrc), and existing studies (Spielmann et al. (2019) for AT-Neu and IT-Soy). As GPP is only available for FLUXNET sites, and CO$_2$ turbulent flux (FC) or net ecosystem exchange (NEE) data are available for other sites, a night flux partitioning model was used to estimate ecosystem respiration ($R_{eco}$) and thus to calculate GPP. The model assumes that nighttime NEE represents ecosystem respiration (Reichstein et al., 2005), and thus partitions FC or NEE into GPP and $R_{eco}$ based on the semi-empirical

models of respiration, which use air temperature as a driver (Lloyd and Taylor, 1994; Lasslop et al., 2012).

### 2.5 Experimental design

Three groups of data assimilation experiments were conducted in this study: (1) 14 model-based twin experiments were performed to investigate the ability of NUCAS to assimilate COS flux data in different scenarios; (2) 13 single-site assimilation experiments were conducted at all seven sites to obtain the site-specific posterior parameters and the corresponding posterior

model outputs based on COS flux observations; (3) one multi-site assimilation experiment was carried out to refine one set of parameters over multiple sites simultaneously and to simulate the corresponding model outputs. Prior simulations using default



parameters were also performed in order to investigate the effect of the COS flux assimilation. Moreover, due to the limitation of the COS observations, all of these experiments were conducted in a one-month time window at the peak of the growing season. Detailed information of these experiments is described in the following.

**2.5.1 Twin experiment**

Model-based twin experiments were performed to investigate the model performance of the data assimilation (Irrgang et al., 2017) at all seven sites considering single-site and multi-site scenarios, and under different perturbation conditions. In each twin experiment, we first created a pseudo-observation sequence by NUCAS using the prior parameters. The pseudo-observation sequence included the prior simulated ecosystem COS fluxes with its uncertainties, and the latter were set to a constant of 1 ($pmol/m^2/s$). Then, a given perturbation ratio was applied to the prior parameters vector, and a perturbed ecosystem COS simulation sequence could be obtained based on the perturbed parameter vector. Finally, the data assimilation experiments were performed to minimize the discrepancy between the prior parameters and the perturbed parameters, and thus the discrepancy between the COS flux pseudo-observations and the perturbed ecosystem COS simulations. The effectiveness of the data assimilation methodology of NUCAS can be validated if it successfully restores the control parameters from the pseudo-observations. And as a gradient-based optimization algorithm is used in NUCAS to tune the control parameters and minimize the cost function, the changes of cost function and gradient over assimilation processes can also be used to verify the assimilation performance of the system. In this work, a total of fourteen twin experiments were conducted, including thirteen single-site twin experiments and one multi-site twin experiment. For all cases where the PFT is evergreen needleleaf forest, a perturbation ratio of 0.2 was used. And for the remaining six single-site twin experiments, a perturbation rate of 0.4 was used.

**2.5.2 Real data assimilation experiment**

After the ability of NUCAS to assimilate COS flux data was confirmed by twin experiments, we could then use the system to conduct data assimilation experiments with real COS observations under single-site and multi-site conditions to optimize the control parameters and state variables of this model, and use the evaluation dataset to test the posterior simulations of the state variables. For the single-site case, a total of thirteen data assimilation experiments were conducted at all of these sites to investigate the assimilation effect of COS flux on optimizing key ecosystem variables. In the diagnostic processes, no perturbation was applied to the default parameters, except for the experiment conducted at the FI-Hyy site in July 2017, where a perturbation ratio of 0.2 was applied. Detailed information about those single-site experiments is shown in **Table 3**.

Single-site assimilation can fully account for the site-specific information, and thus achieve accurate calibration. However, this assimilation approach often yields a range of different model parameters between sites. For large-scale model simulations, only one set of accurate and generalized model parameters is required (Salmon et al., 2022). Thus, multi-site assimilation experiment that can assimilate COS observations from multiple sites simultaneously is necessary to be conducted. Across the seven sites, DK-Sor and US-Ha1 are both dominated by deciduous broadleaved forest, while there is no overlap in the timing of the observations for their COS data. We therefore selected FI-Hyy and US-Wrc, which are both dominated by evergreen needleleaf forest, and conducted a multi-site assimilation experiment with a one-month assimilation window in August 2014.

**2.6 Model evaluation**

For the purpose of demonstrating the process of control parameter vector being continuously adjusted in the normalized parameter space in twin experiment, and quantifying the deviation of the current control vector from the prior, the distance ($D_x$) between the parameter vector and the prior parameter vector was calculated.





$$D_x = \|x - x_0\| = \sqrt{\sum_{i=1}^{n}\left(x(i) - x_0(i)\right)^2} \tag{13}$$

where i denotes the $i$ th parameter in the parameter vectors and n denotes the number of parameters in the parameter vector, and takes a value of 76.

With the aim of evaluating the performance of NUCAS in the real data assimilation experiments, we reran the model to obtain the posterior model outputs based on the posterior model parameters. Typical statistical metrics including mean bias (MB),

root mean square error (RMSE), and correlation of determination ($R^2$) are used to measure the difference between the simulations and in situ observations. They were calculated as:

$$MB = \frac{1}{N}\sum_{i=1}^{N}(obs_i - sim_i) = \overline{obs} - \overline{sim} \tag{14}$$

$$RMSE = \sqrt{\frac{1}{N}\sum_{i=1}^{N}(obs_i - sim_i)^2} \tag{15}$$

$$R^2 = 1 - \frac{\sum_{i=1}^{N}(obs_i - sim_i)^2}{\sum_{i=1}^{N}\left(obs_i - \overline{obs}\right)^2} \tag{16}$$

where "*obs*" and "*sim*" denote the observations and simulations, respectively. $sim_i$ denotes the simulation corresponding to the $i$ th observation $obs_i$. The terms $\overline{obs}$ and $\overline{sim}$ are the mean of observations and the mean of simulations corresponding to the observations. N is the total number of observations.

Given the large variation in the magnitudes of simulations and observations across experiments, the coefficient of variation of RMSE (CV(RMSE)) was employed to compare the assimilation results between different experiments, and it was calculated

by normalizing the RMSE using the mean of observations.

$$CV(RMSE) = \frac{RMSE}{\overline{obs}} \tag{17}$$

Additionally, in order to investigate the sensitivity of COS assimilation to the model parameters, we also calculated the sensitivity coefficient for each parameter at the prior value based on the sensitivity information provided by the adjoint model. The sensitivity coefficient $\Phi$ of any parameter *var* was calculated as:


$$\Phi(var) = \frac{\partial J/\partial x_0(var)}{\|\partial J/\partial x_0\|} \tag{18}$$

where $\|\partial J/\partial x_0\|$ denote the norm of the sensitivity vector of the cost function to the model parameters at the prior values.

## 3 Results

### 3.1 Twin experiments

After dozens of evaluations of the cost function and its gradients, each of the twin experiments was successfully performed. Details of those twin experiments are shown in **Table 2**. In summary, during those assimilations, the cost function values were significantly reduced by more than sixteen orders of magnitude, from greater than $4.58 \times 10^3$ to less than $3.50 \times 10^{-13}$ and the respective gradient values also reduced from greater than $3.94 \times 10^3$ to less than $2.79 \times 10^{-4}$, which verified the ability of the data assimilation algorithm to correctly complete the assimilation.





Corresponding to the PFT and soil texture of the experimental site, some PFT-dependent and texture-dependent parameters as well as global parameters showed different adjustments from others as they can affect the simulation of COS to different degrees. Those parameters are the maximum carboxylation rate at 25 °C ($V_{cmax25}$), the ratio of $V_{cmax}$ to maximum electron transport rate $J_{max}$ (VJ_slope), saturated hydraulic conductivity (Ksat), Campbell parameter (b), and the ratio of photosynthetically active radiation (PAR) to shortwave radiation (f_leaf). Particularly, as the soil textures at the FI-Hyy and

US-Wrc are different, Ksat and b corresponding to these two soil textures were both optimized in the multi-site twin experiment. The relative changes of those parameters with respect to the prior values at the ends of the experiments, as well as the initial values ($D_{itial}$) and the maximums ($D_{max}$) and the final values ($D_{final}$) of $D_x$ are reported in **Table S3**. Results show that the relative differences of those parameters from the "true" values reached very small values at the ends of twin experiments, with the maximum of the absolute values of the relative changes below $2*10^{-8}$. $D_x$ was also reduced to nearly zero with the

maximum value below $2*10^{-7}$, which indicates that all parameters in the control parameter vectors were almost fully recovered from the pseudo observations. In conclusion, these results demonstrate that NUCAS has excellent data assimilation capability under various scenarios with different perturbations, and can effectively perform iterative computations to obtain reliable parameter optimization results during the assimilation process.

**3.2 Single-site assimilation**

With an average of approximately 118 cost function evaluations, all of the 13 single-site experiments were performed successfully. The experiments reduced cost function values significantly, with an average cost function reduction of 33.78% (**Table 3**). However, the minimization efficiency of the experiment varies considerably with PFT, site and assimilation window, ranging from 1.64% to 64.92%. The single-site assimilations tend to achieve greater minimization efficiency at the deciduous broadleaf forest sites and the evergreen needleleaf forest sites, with mean minimization efficiency of 42.74% and 42.39%,

respectively. For the other three PFTs, i.e. grass, crop and shrub, the minimization efficiencies were quite small, ranging from 1.64% to 10.48%, as the simulations of COS using the default parameters at these three sites are already very close to the corresponding observations (**Figure 3**). We found that for different sites with the same PFT, their average minimization efficiencies of the assimilation are in good agreement. However, for the same site, the minimization efficiencies varied considerably from year to year, yet were very similar for the same year. For example, at FI-Hyy, the cost function reduction

in July and August 2014 were almost identical, with 62.23% and 64.92% respectively, both much greater than the reduction rates in other years.

   For all single-site experiments, the model parameters were continuously adjusted during the assimilation and eventually stabilized. Similar to the single-site twin experiments, only five parameters have been efficiently adjusted. **Figure 2** illustrates the evolution of the values of those parameters during the single-site assimilation experiment at the DK-Sor site in June 2016.

At the beginning of the assimilation, each parameter had a great adjustment. As the iterations continued, the parameters gradually stabilized and the minimization was eventually completed. Specifically, $V_{cmax25}$, VJ_slope and f_leaf varied over a very large range during the assimilation, up to 47.92 in the normalized model parameter space. In contrast, the texture-dependent parameter Ksat and b, varied in a very small range between 3.99 and 4.01.

   **Figure 3** illustrates the mean diurnal cycle of observed and simulated COS fluxes. Results show that the prior simulations can

accurately reflect the magnitude of ecosystem COS fluxes and effectively capture the daily variation and the diurnal cycle of COS. On average across all sites, the prior simulated and observed ecosystem COS fluxes were very close, with 21.92 $pmol/m^2/s$ and 21.88 $pmol/m^2/s$, respectively. However, there was substantial variability between sites and even between experiments at the same site. At DK-Sor, the prior simulated COS fluxes were greatly underestimated by 55.72%. In contrast, the prior simulated COS fluxes were overestimated at FI-Hyy, while the overestimation is only significant in 2014, with MBs



of 11.59 $pmol/m^2/s$ and 8.34 $pmol/m^2/s$ in July and August respectively. In general, the MBs of COS fluxes are largely determined by the simulations and observations at daytime due to the larger magnitude (**Figure 3**). However, the model-observation differences at nighttime are also non-negligible. As shown in **Figure 3**, the simulated COS fluxes during nighttime were almost constant and lower than the observations for all experiments. Moreover, the underestimation is particularly evident in AT-Neu, ES-Lma and FI-Hyy.

After the single-site optimizations, both the daily variation and diurnal cycle of COS simulations were improved. This was reflected in the reduction of mean RMSE between the simulated and the observed COS fluxes from 16.69 $pmol/m^2/s$ in the prior case to 13.64 $pmol/m^2/s$ in the posterior case. And similar to the values of cost function, the RMSEs were also reduced in all single-site experiments. Moreover, the assimilation of COS observations also effectively corrected the bias between prior simulations and observations, with mean absolute MB significantly decreased from 6.94 $pmol/m^2/s$ to 3.84 $pmol/m^2/s$.

In contrast, $R^2$ remained almost unchanged by the optimizations, with its mean value increasing slightly from 0.2956 to 0.3037. In addition, the results also demonstrate that the assimilation of COS mainly optimizes the simulated COS fluxes at daytime, while the simulated nighttime COS fluxes are almost unchanged.

The impacts of the assimilation of COS in improving the COS posterior simulations were particularly evident at forest sites, where the prior simulated COS often deviated significantly from the observations, and less evident at low-stature vegetation

(including grass, crop and shrub) sites, as the model using prior parameters already performed very well in the simulations. This result is very reasonable since a similar pattern was also found in the cost function reductions at these sites. For example, with the largest cost function reduction, the assimilation of COS significantly corrected the overestimation of the COS simulations at FI-Hyy in August 2014, with RMSE decrease from 16.13 $pmol/m^2/s$ to 10.11 $pmol/m^2/s$. In contrast, with a reduction in the cost function of only 2.08%, the assimilation of COS had little effect at the IT-Soy site, where the RMSE of

simulated and observed COS only decreased from 12.23 $pmol/m^2/s$ to 12.10 $pmol/m^2/s$. In addition, the performance of the assimilation of COS at these sites was evaluated utilizing CV(RMSE). Results showed that the three experiments with the smallest CV(RMSE)s all were carried out at the FI-Hyy site, in July 2013, 2016 and 2017 respectively, with a mean value of CV(RMSE) of 0.51. While at AT-Neu and US-Wrc, the CV(RMSE)s were much larger, with 0.90 and 0.85 respectively. For AT-Neu, in addition to the large model-observation biases during nighttime (**Figure 3a**), there were also significant deviations

between observations and simulations in the morning due to the high values of observations.

### 3.3 Multi-site assimilation

FI-Hyy and US-Wrc have different soil textures, with loamy sand and silty loam, respectively. In the multi-site assimilation experiment, NUCAS took this difference into account and successfully minimized the cost function from 703.36 to 370.44 after 146 evaluations of cost function. The cost function reduction for the experiment is very reasonable, with a value of

47.33%, comparable to the cost function reductions for corresponding single-site assimilation experiments at FI-Hyy and US-Wrc (64.92% and 44.65%). Furthermore, corresponding to these two soil textures, the texture-dependent parameters Ksat and b yielded two different posterior parameter values, respectively, so that a total of seven parameters were optimized in the multi-site experiment (**Table 4**). **Table 4** shows that with the exception of VJ_slope, the multi-site posterior parameters are all very similar to those of the single-site experiments in both the sign of the change (increase or decrease) and magnitudes of the

adjustments. Overall, both the minimization efficiencies and the parameter optimization results of the multi-site assimilation experiments were very similar to the corresponding single-site experiments, demonstrating the ability of NUCAS to correctly perform joint data assimilation from COS observations at multiple sites simultaneously.

The posterior simulations of COS flux using the multi-site posterior parameters, also demonstrated the ability of NUCAS to correctly assimilate multi-site COS fluxes simultaneously. As shown in **Figure 4**, the prior COS simulations for both the FI-





Hyy site and US-Wrc site show overestimation compared to the observations. However, after the multi-site COS assimilation, the discrepancies between COS simulations and observations were significantly reduced, with RMSE reductions of 36.86% and 9.27%, achieving similar results to the simulations using the single-site posterior parameters.

### 3.4 Parameter change

As mentioned before, there were only five parameters that have been significantly changed during the assimilation of COS flux observations by the NUCAS system, whether in twin, single-site or multi-site experiments. They are the maximum carboxylation rate at 25 °C ($V_{cmax25}$), the ratio of $V_{cmax}$ to maximum electron transport rate $J_{max}$ (VJ_slope), saturated hydraulic conductivity (Ksat), Campbell parameter (b), and the ratio of PAR to shortwave radiation (f_leaf). These parameters are strongly linked to the COS exchange processes and it is therefore reasonable that they could be optimized by the assimilation of COS flux. Furthermore, these parameters are also closely linked to processes such as photosynthesis, transpiration and soil water transport, and therefore provide an indirect constraint for improving the simulation of GPP, LE, H and soil moisture based on the assimilation of COS flux.

For both single-site and multi-site experiments, the changes of those five parameters exhibited different characteristics: The texture-dependent parameters Ksat and b had a very little relative change, while the PFT-specific parameters ($V_{cmax25}$ and VJ_slope) and f_leaf changed dramatically (**Figure 5**). In particularly, the experiment with the largest relative change of Ksat and b performed in July 2017 at FI-Hyy, showed the corresponding relative change of only 1.33% and -2.08% respectively. For other experiments, the relative changes of Ksat and b were much smaller, on average 0.09% and 0.14%, respectively of their absolute values. In contrast, the other three parameters varied considerably after the assimilations, in particular f_leaf, which decreased by 31.55% on average in the single-site experiments. However, among these posterior parameters, $V_{cmax25}$ has the greatest variability, with relative changes ranging from -60.64% to 113.45%.

Across all single-site experiments, there were significant differences in the results of parameter optimization between sites. We found that for those sites where the prior simulations of COS were already very close to COS observations, such as AT-Neu, ES-Lma and IT-Soy, there are still some parameters that varied significantly in the assimilation experiments. For example, in the experiment conducted at AT-Neu, although the cost function reduction of this experiment was only 1.64%, both $V_{cmax25}$ and VJ_slope were changed significantly, with the relative changes of 45.54% and -45.42% respectively. With the opposite directions and similar magnitudes, the relative changes in $V_{cmax25}$ and VJ_slope are very reasonable, and reflect the trade-off of the assimilation system for the parameters which ensured the posterior simulated COS fluxes are still close to the COS observations. For those sites where the prior COS simulations deviated considerably from the observations, the relative changes of the posterior parameters were relatively larger. At DK-Sor, where the prior simulations of COS were significantly underestimated by 55.72%, both $V_{cmax25}$, VJ_slope and f_leaf have been greatly increased in the assimilation. In response to the apparent overestimation in the prior simulations of COS at FI-Hyy, the posterior COS plant uptake related parameters showed an overall decrease, especially f_leaf.

In the multi-site experiment, corresponding to the different soil textures of FI-Hyy and US-Wrc, two different posterior parameter values were obtained for the texture-dependent parameters Ksat and b respectively, while only one posterior parameter value was obtained for each of other parameters. The results show that the posterior values of $V_{cmax25}$ and txt-dependent parameters obtained from the multi-site optimization are very similar to those from the single-site optimization both in terms of the sign and the magnitude of adjustments. However, with a relative change of 30.72% and -63.64% in the multi-site experiment, the posterior VJ_slope and f_leaf were significantly larger and smaller than those in the single-site experiments, respectively.



### 3.5 Parameter sensitivity

Our results suggest that $V_{cmax25}$ has a critical impact on the assimilation results, followed by f_leaf and VJ_slope, while Ksat and b do not influence the assimilation results significantly (**Figure 6**). With absolute sensitivity coefficients ranging from 89.06% to 97.39% except at IT-Soy, the mean absolute sensitivity coefficient of $V_{cmax25}$ is more than three times that of VJ_slope and f_leaf, which are 24.71% and 28.76% respectively. In contrast, for the texture-dependent parameter Ksat and b, their average absolute sensitivity coefficients were only 0.01% and 0.02%, respectively.

Unlike the great variability of the posterior COS plant uptake related parameters, the sensitivities of the cost function to those parameters are very stable (except IT-Soy), especially at the same site. At US-Ha1, for example, the difference between the sensitivity coefficients of $V_{cmax25}$, VJ_slope and f_leaf in its two experiments were all smaller than 0.57%. Among the three parameters, $V_{cmax25}$ has the smallest magnitude of variation in sensitivity coefficient (except IT-Soy), only about half that of VJ_slope and f_leaf, although its sensitivity coefficients are of a much larger order of magnitude. As for Ksat and b, despite

the small values of their sensitivity coefficients, the relative variability is large, with sensitivity coefficients ranging from -0.05% to 0.04 and from -0.03% to 0.07% respectively.

Our results also suggest that the parameters related to light reaction (VJ_slope and f_leaf), tend to play more important roles in the COS assimilation at the forest sites compared to AT-Neu and ES-Lma, while $V_{cmax25}$ does the opposite. However, the smallest absolute $\Phi_{V_{cmax25}}$ was found at the agricultural site IT-Soy with a value of only 23.76%, yet its sensitivity coefficient

of f_leaf is as high as 94.97%.

### 3.6 Comparison and evaluation of simulated GPP

For single-site experiments, both the prior and posterior GPP simulations performed very well in modelling the daily variation and diurnal cycle of GPP, with mean $R^2$ of 0.76 and 0.75, respectively. The discrepancy between simulations and observations was significantly reduced, from mean RMSE of 8.22 $\mu mol/m^2/s$ in the prior case to 6.38 $\mu mol/m^2/s$ in the posterior case

(**Figure 7**). The mean bias between the observed and simulated GPP was also corrected with the reduction in mean absolute MB from 4.82 $\mu mol/m^2/s$ to 3.14 $\mu mol/m^2/s$.

Similar to COS flux, the mean of prior simulated GPP is also generally larger than the observed. We found that the tuning directions of the GPP simulations and the COS simulations were consistent for almost all single-site experiments (12/13). The only exception occurred at AT-Neu, with the simulated COS increasing by 10.32% while the simulated GPP decreasing by

15.24%. Such results also reflect that the sensitivity of COS exchange and photosynthesis to the model parameters differs due to the different physiological mechanisms.

In general, the GPP performance was improved for most of the single-site experiments (9 of 13), with RMSE reductions ranging from 9.41% to 59.83%, while for the other 4 experiments, the posterior RMSEs were slightly higher than the prior by 0.84% to 23.96%. More specifically, across all single-site experiments performed at evergreen needleleaf forest sites, the

posterior GPP simulations were remarkably improved, with an averaged RMSE reduction of 37.92%. At the sites that were dominated by deciduous broadleaf forest, the posterior simulated GPP also achieved a better fit with observations, with an averaged RMSE reduction of 11.99%. However, for experiments conducted on other low-stature vegetation types (including grass, crop and shrub), the RMSEs of the posterior simulated GPP are slightly larger than the prior. Nevertheless, the posterior simulations of GPP for these three sites also achieved a consistent fit to the GPP observations, with their CV(RMSE)s all

smaller than the averaged CV(RMSE) of all posterior simulations in single-site experiments. Moreover, for AT-Neu and IT-Soy, the GPP observations exhibited significant fluctuations even at night, suggesting that they may have large uncertainties, which is to be considered in the evaluations of our GPP simulations.



Covering different years or months, the single-site experiments performed at FI-Hyy and US-Ha1 provided an opportunity to analyze inter-annual and seasonal variation in the simulated and observed GPP. At US-Ha1, the prior simulations in July 2012

and 2013 overestimated GPP by almost the same degree, 30.58% and 34.58% respectively, while the corresponding posterior simulated GPP differs considerably. In July 2012, the model using the posterior parameters performed very well in GPP simulations, with MB of only 0.20 $\mu mol/m^2/s$. In contrast, the posterior GPP simulations in July 2013 were significantly underestimated, with MB of -6.38 $\mu mol/m^2/s$. At FI-Hyy, a total of six single-site experiments were conducted between 2013 and 2017, five of them in July and one in August 2014. The observed GPP shows little inter-annual variation in July

from 2013 to 2017, with the mean ranging from 8.30 $\mu mol/m^2/s$ to 9.15 $\mu mol/m^2/s$, while the mean for August of 6.43 $\mu mol/m^2/s$ was noticeably lower than that in July. As for simulations, the prior simulations tend to overestimate GPP, with MBs ranging from 3.76 $\mu mol/m^2/s$ to 6.61 $\mu mol/m^2/s$. However, the posterior GPP differs considerably, in some experiments achieving excellent match with the observations and other experiments yielding very low simulated GPP. In July 2013, 2015 and 2016, the model using posterior parameters performs well in simulating GPP and achieves the smallest

CV(RMSE)s of all single-site experiments, with CV(RMSE)s ranging from 0.39 to 0.42. In contrast, as the observed COS is lower than the prior simulated COS by 39.64% and 39.32% in July and August 2014, f_leaf and $V_{cmax}$ were dramatically adjusted downwards in July and August respectively, resulting in notable underestimation in the posterior simulated GPP, with MBs of -6.27 $\mu mol/m^2/s$ and -2.57 $\mu mol/m^2/s$. In addition, a dramatic reduction of f_leaf was also reported in July 2017 and resulted in an underestimation of posterior simulated GPP.

In the multi-site experiment, the posterior model-observation differences for GPP were reduced for both FI-Hyy and US-Wrc, with RMSE reductions of 45.85% and 55.71%, respectively. These RMSE reductions are even higher than those in the corresponding single-site experiments, by 20.34% for FI-Hyy and 7.84% for US-Wrc. These results suggest that simultaneous assimilation using COS observations from multiple sites can also improve GPP simulations, and the assimilation is sometimes more effective than the single-site assimilation.

Overall, the assimilation of COS data can improve the simulation of GPP in both single-site assimilation and multi-site assimilation. However, the assimilation effects vary considerably for different sites and even for different periods within the same site. The assimilation of COS degrades the fit to observed GPP at low-stature vegetation sites (including AT-Neu, ES-Lma and IT-Soy) where the prior COS simulations perform well. By contrast, for the single-site experiments conducted at forest sites, the assimilation can always improve the simulation of GPP, although the optimizations were sometimes affected

by the over-tuning of $V_{cmax25}$ and f_leaf.

### 3.7 Comparison and evaluation of simulated H and LE

In order to verify the impact of COS assimilation on stomatal conductance and energy balance, observations of latent heat and sensible heat were compared to the prior and posterior model outputs. Due to the lack of observations at AT-Neu and IT-Soy, the validation was carried out at the remaining five sites only. Results showed that the assimilation of COS is generally able

to improve both latent and sensible heat, whether in single-site experiment or multi-site experiment. And the assimilation is more effective in improving the simulation of LE, with the average RMSE decreasing from 94.69 $W/m^2$ to 79.69 $W/m^2$, while for H, the average RMSE only decreased from 101.65 $W/m^2$ to 96.29 $W/m^2$.

Results show that the BEPS model can simulate the daily variations of H and LE as well as the diurnal cycle of LE very well, while the diurnal cycle of H is relatively poorly simulated. The prior simulation tends to overestimate LE during the daytime,

and to exhibit short-time fluctuations in H that is not present in the observations. On average across all experiments, the prior simulated LE is overestimated by 41.88 $W/m^2$ (**Figure 8** and **Figure S1**) while the prior simulated H is underestimated by 39.92 $W/m^2$ (**Figure 8** and **Figure S1**). The overestimation of LE and the underestimation of H are particularly apparent at



the evergreen needleleaf forest sites (FI-Hyy and US-Wrc). In addition, at FI-Hyy and US-Wrc, the model-observation biases are more pronounced for H, with an averaged MB of -62.13 $W/m^2$ than for LE with the averaged MB of 41.78 $W/m^2$. These

results indicate that the BEPS model may underestimate the solar radiation absorbed by the evergreen needleleaf forest ecosystem. For the deciduous broadleaf forest sites DK-Sor and US-Ha1, the prior simulations of H are very close to observations, with a maximum absolute MB of only 16.18$W/m^2$. However, similar to evergreen needleleaf forests, its prior simulations also tend to overestimate LE, with MB ranging from 17.92 $W/m^2$ to 61.34 $W/m^2$. With a shrub PFT, ES-Lma is the only site where the prior simulations overestimate both H and LE, with MB of 22.00 $W/m^2$ and 50.06 $W/m^2$

respectively, which poses a significant challenge for the simultaneous optimization of H and LE.

In general, the single-site assimilation of COS effectively corrected the biases in the prior simulations of H and LE, and the correction mainly affected the daytime. Moreover, the correction was particularly effective for the evergreen needleleaf forest sites, where the mean values of the simulations of H and LE were increased by 30.95 $W/m^2$ and decreased by 31.04 $W/m^2$ respectively. With a mean RMSE reduction of -25.56%, the improvements of LE are also larger than the improvements of H.

For the deciduous broadleaf forest sites, the optimization results for LE and H show considerable inconsistency. At US-Ha1, the model overestimated the absorbed solar radiation energy both in July 2012 and 2013. And the assimilation of COS significantly corrected the overestimation of LE, with RMSE reduction of 25.63% in July 2012 and 28.90% in July 2013. In contrast to the reduction of LE, the H was increased by 21.40 and 54.40 $W/m^2$, in the respective period. At DK-Sor, the simulations of H and LE using the default parameters of the BEPS model already performed very well, and little improvement

is needed. However, as the prior simulated COS was much lower than observed COS, parameters including $V_{cmax25}$, VJ_slope and f_leaf were increased after the assimilation. As a result, the model output using the posterior parameters overestimated LE and underestimated H. As for ES-Lma, where the prior model output overestimated both H and LE, the posterior simulated LE was overestimated yet stronger, while the overestimation of H was partially corrected.

At US-Wrc, the multi-site assimilation greatly corrected the overestimation of LE and the underestimation of H in the prior

simulations during the daytime, with RMSE reductions of 26.57% for LE and 32.99% for H, achieving almost identical effect to single-site optimization. Similar to US-Wrc, the LE simulations obtained with the multi-site posterior parameters were reduced by about one third compared to the prior simulations at FI-Hyy, which allowed the overestimation of the prior simulation during the first half of the month to be effectively corrected (**Figure 8a**). Meanwhile, the model-observation differences of H were also remarkably reduced at FI-Hyy, with MB of -63.44 $W/m^2$ for the prior case and -39.93 $W/m^2$ for

the posterior case.

Overall, the BEPS model performed well in simulating the daily variations and diurnal cycle of H and LE, while it tended to overestimate LE during the daytime and underestimate H around midday and sunset. Generally, the assimilation of COS could effectively improve the simulation of LE and H, whether the assimilation was conducted at single-site or at multiple sites simultaneously, and this improvement was particularly noticeable for the simulation of LE. We also found that the simulated

LE was always adjusted in the same direction as the COS, while H was adjusted in the opposite direction.

### 3.8 Comparison and evaluation of simulated SWC

The effectiveness of COS assimilation in improving soil moisture simulations was assessed by comparing hourly soil water content observations with hourly simulations of soil moisture using prior parameters, single-site and multi-site posterior parameters. The assessments were carried out at all sites except US-Ha1, where no soil water observations were available. We

found that COS assimilation also improved the simulation of soil moisture and this improvement was closely linked to the improved simulation of LE. However, the improvement of soil moisture was not significant in a short period of time.





Results show that the model can roughly follow the soil moisture trend (**Figure 9** and **Figure S3**). However, the simulated soil water content (SWC) exhibited a clear cycle of diurnal variation whereas the observed SWC had almost no diurnal fluctuations. Generally, in response to the overestimation of LE, the prior simulations tended to overestimate the rate of decline in SWC.

After the assimilation of COS, the overestimation of the decline rate of SWC was significantly corrected and the posterior SWC simulations were more closely aligned with observations in terms of state and trend. For example, during the first half month of August 2014 at FI-Hyy, the prior simulations greatly overestimated LE (**Figure 8a**), such that the corresponding simulated SWC dropped rapidly to the wilting point and then remained constant (**Figure 9c**). In contrast, with the simulated LE being notably corrected by the assimilation of COS, the simulated SWC was also effectively corrected to the level of the

observations.

     However, the effect of the assimilation of COS on the optimization of SWC simulations varied considerably from site to site. Little difference was found between the prior and the posterior simulations of SWC for those sites (AT-Neu, ES-Lma, IT-Soy) where there the GPP simulations also changed little after the assimilations of COS. The model significantly overestimated the rate of soil moisture decline at US-Wrc and DK-Sor, with the posterior simulated LE being about 169% and 78% larger than

the observed. In contrast, the assimilation of COS remarkably improved the SWC simulations at FI-Hyy, with an average RMSE reduction of 24.86%. Yet, at FI-Hyy site, the experiment results (**Figure 9**) also showed there is still a large mismatch of observed and simulated decline rate of SWC during inter-storm periods and of the effect of precipitation on SWC.

## 4 Discussion

### 4.1 Parameter changes

As we mentioned before, the texture-dependent parameters Ksat and b had a very small relative change in the assimilation of COS, while the parameters related to PFT ($V_{cmax25}$ and VJ_slope) and f_leaf varied dramatically. This is because COS plant fluxes are much larger than COS fluxes of soil in general (Whelan et al., 2016; Whelan et al., 2018) and the texture-dependent parameters cannot directly influence the COS plant uptake. Therefore, the assimilation of the COS flux mainly changed the parameters related to COS plant uptake rather than texture-dependent parameters that relate to soil COS flux to minimize the

cost function. Among the three COS plant uptake related parameters, it was found that the posterior $V_{cmax25}$ had the largest change relative to the prior, with the relative change ranging from -60.64% to 113.45%, followed by f_leaf and VJ_slope. Although the posterior f_leaf has significant variability, f_leaf varies little in reality and is usually between 41% and 53% on an annual mean scale (Ryu et al., 2018). Considering that f_leaf is set to 0.5 in our model, it should remain about the same or be slightly reduced after the optimization. Certainly, the relative change rate of f_leaf is very reasonable in some experiments,

such as the single-site experiments conducted at FI-Hyy in August 2014 and July 2015, with relative changes of -14.18% and -13.29% respectively. However, the posterior f_leaf was also reduced dramatically by more than 60% in some single-site experiments conducted at FI-Hyy and US-Ha1, which suggested that the assimilation of COS may lead to over-tuning of f_leaf in some cases. COS plant uptake is governed by the reaction of COS destruction (Wohlfahrt et al., 2012) by carbonic anhydrase though it can also be destroyed by other photosynthetic enzymes, e.g., RuBisCo (Lorimer and Pierce, 1989), and the reaction

is not dependent on light (Stimler et al., 2011; Whelan et al., 2018). Yet, given that stomatal conductance is simulated from net photosynthetic rate with a modified version (Woodward et al., 1995; Ju et al., 2010) of the Ball-Woodrow-Berry (BWB) model (Ball et al., 1987), the adjustment of f_leaf can therefore indirectly affect the simulation of COS plant uptake by influencing the calculation of stomatal conductance. As mentioned in Sect 3.2, the prior simulated COS fluxes were larger than the observed ones at FI-Hyy and US-Ha1. Therefore, the assimilation of COS resulted in down-regulations of f_leaf in

the single-site experiments performed at FI-Hyy and US-Ha1.





In addition to f_leaf, $V_{cmax25}$ was also over-adjusted in a few assimilation experiments, particularly at the experiments conducted in August 2014. For example, at US-Wrc, $V_{cmax25}$ was dramatically down-regulated by a similar degree in the single-site and multi-site experiment, with relative change of -50.63% and -44.64% respectively, whereas the posterior VJ_slope and f_leaf are significantly different. However, with such different posterior parameters, the posterior simulated COS
are very similar (**Figures 4b**). These results revealed the 'equifinality' (Beven, 1993) of the inversion problem at hand, i.e. the fact that different combinations of parameter values can achieve a similar fit to the COS observations. Assimilation of further observational data streams is expected to reduce the level of equifinality by differentiating between such combinations of parameter values that achieve a similar fit to COS observations.

### 4.2 Parameter sensitivity

It has been widely proved that photosynthetic capacity simulated by terrestrial ecosystem models is highly sensitive to $V_{cmax}$, $J_{max}$, and light conditions (Zaehle et al., 2005; Bonan et al., 2011; Rogers, 2014; Sargsyan et al., 2014; Koffi et al., 2015; Rogers et al., 2017). Therefore, it is expected that $V_{cmax25}$, VJ_slope, and f_leaf would significantly affect the optimization results, as these parameters ultimately have an impact on the simulation of plant COS uptake by influencing the estimation of photosynthesis capacity and stomatal conductance. Specifically, results of Wang et al. (2004), Verbeeck et al. (2006), Staudt
et al. (2010), Han et al. (2020) and Ma et al. (2022) showed that the simulated photosynthetic capacity was generally more sensitive to $J_{max}$ and light conditions than to $V_{cmax}$. However, due to the differences in the physiological mechanisms of COS plant uptake and photosynthesis, e.g., the hydrolysis reaction of COS by carbonic anhydrase is not dependent on light, the sensitivities of the two processes with respect to the model parameters may differ considerably although they are tightly coupled. Indeed, our adjoint sensitivity results suggest that $V_{cmax25}$ is capable to influence the assimilation results to a greater
extent than VJ_slope and f_leaf. This result can be attributed to the model structure that $V_{cmax25}$ not only affects the estimation of stomatal conductance through photosynthesis, but is also used to characterize mesophyll conductance and CA activity due to their linear relationships with $V_{cmax}$ (Badger and Price, 1994; Evans et al., 1994; Berry et al., 2013). In addition, such a large sensitivity of $V_{cmax25}$ also indicates the importance of accurate modelling of the apparent conductance of COS for ecosystem COS flux simulation. As for Ksat and b, they also play a role in the assimilation of COS since the SWC simulations of BEPS are sensitive to the two (Liu et al., 2011). But since the soil COS exchange is generally much smaller than COS plant uptake
(Whelan et al., 2018) and they have less impact on the simulation of GPP (Novick et al., 2022), the assimilation results are not significantly affected by these two parameters.

In Sect 3.5, we mentioned that the parameters related to light reaction (VJ_slope and f_leaf), tend to play more essential roles in the assimilation of COS at the forest sites. Actually, similar features were found in the sensitivity of photosynthesis to
radiation, i.e. the simulated GPP was more sensitive to radiation at forested vegetation types and less sensitive at low-stature vegetation types (Sun et al., 2019). Particularly, the simulated GPP was also found to be highly sensitive to variations of radiation at low radiation conditions (Koffi et al., 2015). At IT-Soy, **Figure 3j** showed that the assimilation of COS observations mainly changes the COS simulation in the early evening to minimize the cost function. Thus, it is reasonable that f_leaf is the most influential parameter for that experiment as photosynthesis is very sensitive to radiation under such low light
condition and f_leaf is an essential parameter for the calculation of PAR.

### 4.3 Impacts of COS assimilation on ecosystem carbon, energy and water cycles

Due to the physiological basis that COS is taken up by plants through the same pathway of stomatal diffusion as $CO_2$, the assimilation of COS was expected to optimize the simulation of GPP. And it was confirmed by our single-site and multi-site experiments conducted in a variety of ecosystems. However, limited by many factors, such as the observation errors of the





COS fluxes, the assimilation of COS does not always improve the simulation of GPP, especially if the prior simulations of COS are already very close to the observations. Moreover, the assimilation of COS could sometimes lead to overshooting of photosynthesis-related parameters, such as f_leaf, and thus result in considerable errors in the GPP simulations. In our experiments, those significant overshoots of f_leaf all occurred at well-vegetated forest sites (FI-Hyy and US-Ha1). This is also very reasonable as f_leaf is relevant to the calculation of PAR and light can become a limiting factor for photosynthesis,

in particular when plants grow in dense vegetation (Demarsy et al., 2018).

Similar to the photosynthesis, the transpiration is also coupled with the COS plant uptake through stomatal conductance. But the difference is that after $CO_2$ is transported to the chloroplast surface, it continues its journey inside the chloroplast, and is eventually assimilated in the Calvin cycle (Wohlfahrt et al., 2012; Kohonen et al., 2022). Based on the BWB model, photosynthesis-related parameters only indirectly influence the calculation of stomatal conductance through photosynthesis in

our model. In our experiments, posterior simulation results consistent with this mechanism were obtained in that although the posterior GPP simulations significantly deviate from reality due to parameter overshooting, the posterior LE does not. An example is the experiment conducted at FI-Hyy in July 2014, in which the posterior simulated GPP was substantially underestimated by 68.77%, while the posterior simulated LE was only 19.57% lower than the observations. Moreover, as transpiration rate and leaf temperature change show a linear relationship (Kümmerlen et al., 1999; Prytz et al., 2003) and

surface-air temperature difference is a key control factor for sensible heat fluxes (Campbell and Norman, 2000; Arya, 2001; Jiang et al., 2022), the optimization for transpiration can therefore improve the simulation of leaf temperature and consequently improve the simulation of sensible heat flux.

Driven by the difference in water potential between the atmosphere and the substomatal cavity (Manzoni et al., 2013), the water is taken up by the roots, flows through the xylem, and exits through the leaf stomata to the atmosphere in the soil-plant-

atmosphere continuum (Daly et al., 2004). Thus, when plants transpire, the water potential next to the roots decreases, driving water from bulk soil towards roots (Carminati et al., 2010) and reducing soil moisture. Certainly, soil moisture dynamics are also influenced by soil evaporation and leakage during inter-storm periods under ideal conditions (Daly et al., 2004). However, studies have shown that transpiration represents 80 to 90 percent of terrestrial evapotranspiration (Jasechko et al., 2013) and evaporation is typically a small fraction of transpiration for well-vegetated ecosystems (Scholes and Walker, 1993; Daly et al.,

2004). Based on current knowledge of leakage, for example the relationship between leakage and the behavior of hydraulic conductivity (Clapp and Hornberger, 1978), extremely small adjustments of Ksat and b, with average of the absolute values of the relative changes of 0.17% and 0.28% across all of the data assimilation experiments, hardly caused any change in leakage. Therefore, our results indicate that the assimilation of COS can significantly improve the modelling of stomatal conductance and transpiration and finally improve soil moisture. However, our results also show that there are large uncertainties in the

BEPS model for the simulation of the decline rate of SWC during inter-storm periods and of the effect of precipitation on SWC, although in some cases the model using the posterior parameters has already achieved an excellent simulation of LE. This result suggests that there may still be significant errors in the soil texture-related parameters, and that these errors cannot be effectively corrected by the assimilation of COS due to the weak connection between ecosystem COS fluxes and soil hydrological processes.

**4.4 Impacts of leaf area index data on parameter optimization**

As an essential input data of the BEPS model, LAI products have been demonstrated to be a source of uncertainty in the simulation of carbon and water fluxes (Liu et al., 2018). Therefore, it is necessary to investigate the influence of LAI on our parameter optimization results, as the LAI is directly related to the simulation of COS and the discrepancy between COS simulations and COS observations is an essential part of the cost function. Here we collected three widely used satellite-derived



LAI products (GLOBMAP, GLASS and MODIS) and the means of in situ LAI during the growing seasons or during the COS measurement periods for these sites (see **Table 2**). These in situ LAI means were used to drive the BEPS model along with the other three satellite-derived LAI products, with the assumption that they are representative of the LAI values during the assimilation periods. The configurations of those assimilation experiments were the same as those listed in **Table 2**, so that a total of 52 single-site experiments were conducted. Almost all experiments were successfully performed, with the exception of a few at the DK-Sor and IT-Soy sites, and the results were shown in in **Figure 10** and **Figure S4.**

We found that the posterior $V_{cmax25}$ correlated best with the LAI ($R^2 = 0.23$, P < 0.01), followed by VJ_slope ($R^2 = 0.14$, P < 0.05) and f_leaf ($R^2 = 0.09$, P < 0.1). Whilst there was no apparent relationship between the optimization results of the other three parameters and the LAI. As mentioned before, the LAI is directly related to the simulation of COS and thus influences the optimal values of the parameters. Therefore, to some extent, the correlations of LAI with these parameters reflects the robustness of the constraint abilities of COS assimilation with respect to them. These results suggest that the assimilation of COS is able to provide strong constraints on $V_{cmax25}$, while it constrains VJ_slope and f_leaf weakly, although the latter also considerably changed by the assimilation.

In Sect 3.4, we have noted that the posterior $V_{cmax25}$ and f_leaf were sometimes over-tuned, which significantly influenced the posterior simulation of GPP. Here, by comparing the posterior parameters obtained with different LAI data, we further found that the over-tuning of those parameters could be partly attributed to the uncertainty of the LAI. For example, in the experiment conducted at FI-Hyy in July 2017, driven by the GLOBMAP LAI which were on average 41% greater than the in situ LAI, the posterior f_leaf value was significantly reduced, with a decrease rate of 78.09%. However, when the GLASS LAI, which is only 4% larger than the in situ LAI, is used to drive the model, the percentage decrease in f_leaf is significantly reduced to only 43.12%. Such results suggest that the uncertainty in satellite-derived LAI not only can exert large impacts on the modelling of water-carbon fluxes (Wang et al., 2021), but also is an important source of the uncertainty in the parameter optimization results when performing data assimilation experiments with ecosystem models driven by LAI.

## 4.5 Caveats and implications

In general, we found that the assimilation of COS can improve the model performance for GPP, LE, H and SWC for both single-site assimilation and multi-site assimilation. Nonetheless, there are currently limitations that affect the use of COS data for the optimization of parameters, processes and variables related to water-carbon cycling and energy exchange in terrestrial ecosystem models.

The assimilation of COS fluxes relies on the availability and quality of field observations. As both COS plant uptake and COS soil exchange are modelled within NUCAS and the data assimilation was performed at the ecosystem scale, a large number of accurate measurements of both COS soil flux and COS plant flux are essential for assimilation. However, at present, we face a serious lack of ecosystem-scale field measurements (Brühl et al., 2012; Wohlfahrt et al., 2012), more laboratory and field measurements are needed for better understanding of mechanistic processes of COS. Besides, the existing COS flux data were calculated based on different measurement methods and data processing steps, which poses significant challenges for comparing COS flux measurements across sites. Standardization of measurement and processing techniques of COS (Kohonen et al., 2020) is therefore urgently needed.

In this study, the uncertainty of observation was estimated by the standard deviation of ecosystem COS fluxes within 24 hours with the assumption of a normal distribution. However, Hollinger and Richardson (2005) suggested that flux measurement error more closely follows a double exponential than a normal distribution. Furthermore, the prior uncertainty of the parameters was simply set to 25% of the prior values in this study, which could certainly be refined. In conclusion, we should be more careful in considering the distribution and the magnitude of the uncertainty of observations and parameters.



The spatial and temporal variation in atmospheric COS concentrations has a considerable influence on the COS plant uptake (Ma et al., 2021) due to the linear relationship between the two (Stimler et al., 2010). The typical seasonal amplitude of atmospheric COS concentrations is $\sim$ 100–200 parts per trillion (ppt) around an average of $\sim$ 500 ppt (Montzka et al., 2007; Kooijmans et al., 2021; Hu et al., 2021; Ma et al., 2021; Belviso et al., 2022). However, COS mole fractions in the bulk air are currently assumed to be spatially invariant over the globe and to vary annually in NUCAS, which may introduce significant

errors into the parameter calibration. Kooijmans et al. (2021) has confirmed that modifying the COS mole fractions to vary spatially and temporally significantly improved the simulation of ecosystem COS flux. Thus, we suggest to take into account the variation in COS concentration and their interaction with surface COS fluxes at high spatial and temporal resolution in order to achieve better parameter calibration.

Currently, there are still uncertainties in the simulation of COS fluxes by BEPS particularly for nighttime COS fluxes. As the

nighttime COS plant uptake is driven by stomatal conductance (Kooijmans et al., 2021), the nighttime COS fluxes can therefore be used to test the accuracy of the model settings for nighttime stomatal conductance ($g_n$). In the BEPS model, A low and constant value (1 $mmol/m^2/s$) of $g_n$ was set for all PFTs. Our simulations of nighttime COS flux indicate that $g_n$ is underestimated in different degrees in BEPS for different sites. This result is also proved by Resco De Dios et al. (2019), which found that the median $g_n$ in the global dataset was 40 $mmol/m^2/s$. In addition, soil COS exchange is an important source of

uncertainty in the use of COS as carbon-water cycle tracer since carbonic anhydrase activity occurs in the soil as well (Kesselmeier et al., 1999; Smith et al., 1999; Ogée et al., 2016; Meredith et al., 2019). Kaisermann et al. (2018) showed that COS hydrolysis rates were linked to microbial C biomass, whilst COS production rates were linked to soil N content and mean annual precipitation (MAP). Interestingly, MAP was also suggested to be the best predictor of $g_n$ in Yu et al. (2019) which found that plants in locations with lower rainfall conditions had higher $g_n$. Therefore, using the global microbial C biomass,

soil N content and MAP datasets and the relationships between these variables and the associated COS exchange processes is expected to further achieve more accurate modelling of terrestrial ecosystem COS fluxes, increase the understanding of the global COS budget and facilitate the assimilation of COS fluxes.

**5 Conclusions**

Over the past decades, considerable efforts have been made to obtain field observations of COS ecosystem fluxes and to

describe empirically or mechanistically COS plant uptake and soil exchange, which offers the possibility of investigating the ability of assimilating ecosystem COS flux to optimize parameters and variables related to the water and carbon cycles and energy exchange. In this study, we first introduced the NUCAS system, which has been developed based on the BEPS model and was designed to have the ability to assimilate ecosystem COS flux data. In NUCAS, the resistance analog model of COS plant uptake and the empirical model of soil COS flux were embedded in the BEPS model to achieve the simulation of

ecosystem COS flux, and a gradient-based 4D-Var data assimilation algorithm was implemented to optimize the internal parameters of BEPS.

Fourteen twin experiments, thirteen single-site experiments and one multi-site experiment within the period from 2012 to 2017, were conducted to investigate the data assimilation capability and the optimization effect of parameters and variables of NUCAS for COS flux observations over a range of ecosystems that contains five PFTs and five soil textures. Our results show

that NUCAS has the ability to optimize parameter vectors, and the assimilation of COS can constrain parameters affecting the simulation of carbon and water cycles and energy exchange and thus effectively improve the performance of the BEPS model. We found that there is a tight link between the assimilation of COS and the optimization of LE, which demonstrates the role of COS as an indicator of stomatal conductance and transpiration. The improvement of transpiration can further improve the



model performance for H and SWC, although the propagation of the optimization effect is subject to some limitations. These
results highlight the broad perspective of COS as a tracer for improving the simulation of variables related to stomatal
conductance. Furthermore, we demonstrated that COS can provide a strong constraint on $V_{cmax25}$, whereas the adjustment of
parameters related to the light reaction of photosynthesis appears to compensate for weaknesses in the model. We also proved
the strong impact of LAI on the parameter optimization results, emphasizing the importance of developing more accurate LAI
products for models driven by observed LAI. In addition, we made a number of recommendations for future improvement of
the assimilation of COS. Particularly, we flagged the need for more observations of COS, suggested better characterisation of
observational and prior parameter uncertainties, the use of varying COS concentrations and the refinement of the model for
COS fluxes of soil.

It should be noted that the NUCAS was designed as a platform that integrates multiple data streams to provide a consistent
map of the terrestrial carbon cycle although only ecosystem COS flux data were used to evaluate the performance of NUCAS
in this study. As shown here, the optimization of model parameters often faces the challenge of 'equifinality' due to the
complexity of the model and the limited observation data. However, the 'equifinality' can be avoided by imposing additional
observational constraints (Beven, 2006). Indeed, using several different data streams to simultaneously (Kaminski et al., 2012;
Schürmann et al., 2016; Scholze et al., 2016; Wu et al., 2018; Scholze et al., 2019) or step-wise (Peylin et al., 2016) to constrain
multiple processes in the carbon cycle is becoming a focus area in carbon cycle research. Therefore, it is necessary to combine
COS with other observations to constrain different ecosystem processes and/or exploit multiple constraints on the same
processes in order to achieve better modelling and prediction of the ecosystem water-carbon cycle and energy exchange.

*Code availability.* The source code for BEPS is publicly available at https://github.com/yongguangzhang/BEPS-SIF-model,
the adjoint code for BEPS is available upon request to the correspondence author (mousongwu@nju.edu.cn).


*Data availability.* Measured eddy covariance Carboy sulfide fluxes data can be found at https://zenodo.org/record/3406990
for AT-Neu, DK-Sor, ES-Lma and IT-Soy, https://zenodo.org/record/6940750 for FI-Hyy, and from the Harvard Forest Data
Archive under record HF214 (https://portal.edirepository.org/nis/mapbrowse?packageid=knb-lter-hfr.214.4 ) for US-Ha1.The
raw COS concentration data of US-Wrc can be obtained at https://zenodo.org/record/1422820. The meteorological data can
be obtained from the FLUXNET database (https://fluxnet.org/) for AT-Neu, DK-Sor, ES-LMa, FI-Hyy; from the AmeriFlux
database (https://ameriflux.lbl.gov/) for US-Ha1 (except shortwave radiation data) and US-Wrc; from the ERA5 dataset
(https://cds.climate.copernicus.eu/cdsapp#!/dataset/reanalysis-era5-single-levels?tab=overview) for AT-Neu, IT-Soy and US-
Ha1. The evaluation data can be obtained from the FLUXNET database for DK-Sor, ES-LMa, FI-Hyy; from the AmeriFlux
database for US-Ha1 and US-Wrc; and from https://zenodo.org/record/6940750 for AT-Neu and IT-Soy. The GLOBMAP LAI
is available at https://zenodo.org/record/4700264#.YzvSYnZBxD8%2F, the GLASS LAI is available at ftp://ftp.glcf.umd.edu/,
and the MODIS LAI product is available at https://lpdaac.usgs.gov/products/mod15a2hv006/. All datasets used in this study
and the model outputs are available upon request.

*Author contributions:* MW designed the experiments and developed the model, MV and TK developed the data assimilation
layer including the adjoint code for the ecosystem model, HZ wrote the original manuscript and made the analysis. All the
authors contributed to the writing of the manuscript.

*Competing interests:* The authors declare that they have no conflict of interest.



*Acknowledgements:* This study was supported by the National Key Research and Development Program of China (2020YFA0607504, 2016YFA0600204), the National Natural Science Foundation of China (42141005, 41901266), the Research Funds for the Frontiers Science Center for Critical Earth Material Cycling, Nanjing University (Grant No: 090414380031). MV and TK thank Laurent Hascoët for supporting this activity.

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

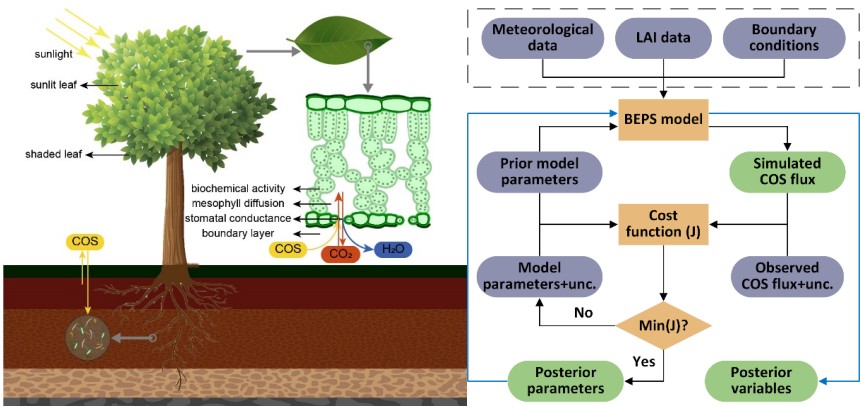

**Figure 1. Schematic of the Nanjing University Carbon Assimilation System (NUCAS). Left: illustration of a two-leaf model coupling stomatal conductance, photosynthesis, transpiration and COS uptake, and an empirical model for simulating soil COS fluxes in NUCAS. Right: data assimilation flowchart of NUCAS. Ovals represent input (blue-grey) and output data (green). Boxes and the rhombi represent the calculation and judgement steps. The solid black line represents the diagnostic process, the solid blue line represents the prognostic process, and the input datasets of BEPS are used in both diagnostic process and prognostic process.**



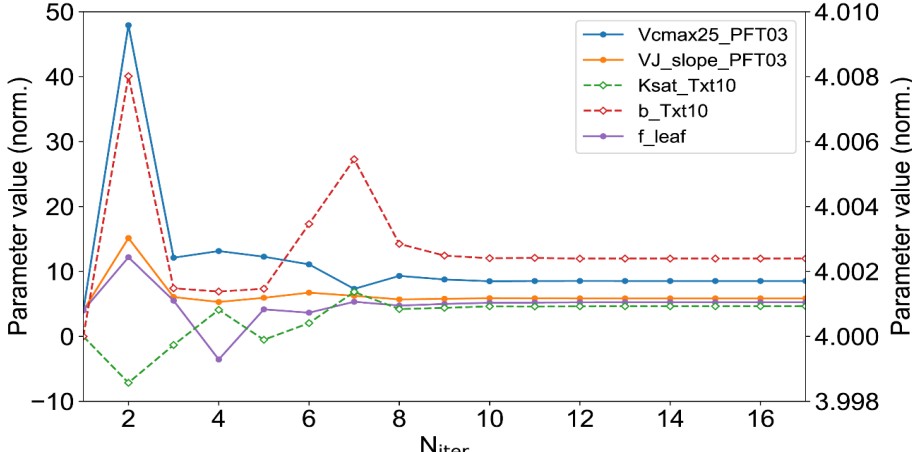

**Figure 2. The evolution of model parameters with the number of iterations of cost function ($N_{iter}$) in the normalized parameter space during the single-site experiment at the DK-Sor site in June 2016. Evolution (open carats and dashed lines) of soil texture (abbreviated as Txt) dependent parameters is plotted on the right-hand y axis, evolution (filled circles and solid lines) of PFT-dependent parameters and global parameter is plotted on the left-hand y axis.**




**Figure 3. The mean diurnal cycle of observed (blue) and simulated COS flux using prior parameters (red) and single-site posterior parameters (green). The size of the circle indicates the number of observations within each circle, and the error bars depict the standard deviations in the mean of observations from the variability within each circle if the number of corresponding observations is greater than three. Lines connect the mean values of simulations and pale bands depict the standard deviation in the mean of simulations from the variability within each bin.**

1160




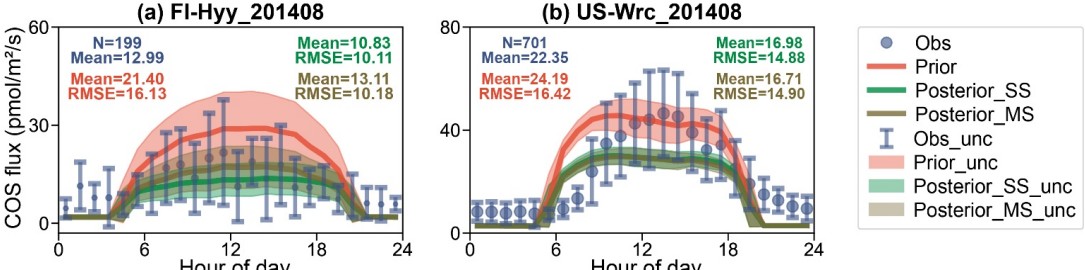

**Figure 4. The diurnal cycle of observed (blue) and simulated COS flux using prior parameters (red), single-site (green) and multi-site (brown) posterior parameters. The size of the circle indicates the number of observations within each circle, and the error bars depict the standard deviations in the mean of observations from the variability within each circle if the number of corresponding observations is greater than three. Lines connect the mean values of simulations and pale bands depict the standard deviation in the mean of simulations from the variability within each bin.**

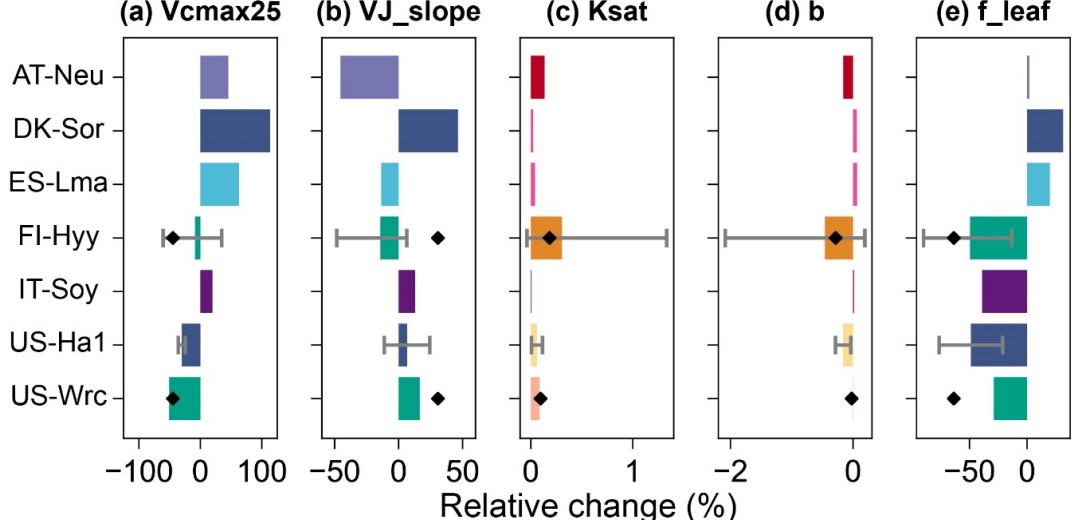

**Figure 5. Relative changes of parameters for single-site experiments (bars) and the multi-site experiment (diamond points). For sites where multiple single-site experiments were conducted, the ends of the error bars and the bar indicate the maximum, minimum and mean of the relative changes of the parameters, respectively. For sites with the same PFT or soil texture, the same colors were used for their PFT-dependent and texture-dependent parameters, and f_leaf was plotted using the same color scheme as the PFT-dependent parameters.**





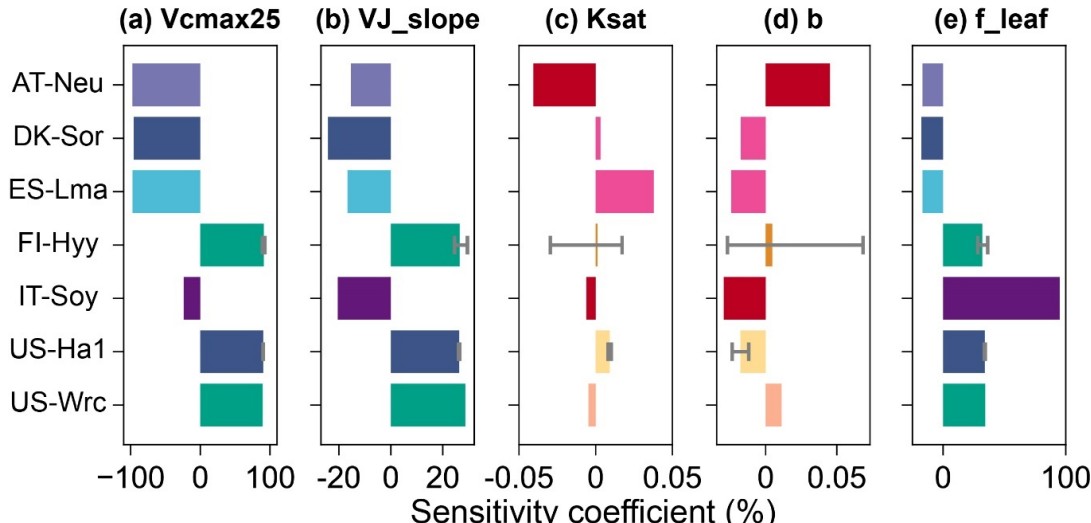

**Figure 6. Sensitivity coefficients of parameters at default values. The ends of the error bars and the bar indicate the maximum, minimum and mean of the sensitivity coefficients of the parameters, respectively. For sites with the same PFT or soil texture, the same colors were used for their PFT-dependent and texture-dependent parameters, and f_leaf was plotted using the same color scheme as the PFT-dependent parameters.**





**Figure 7.** The diurnal cycle of observed (blue) and simulated GPP using prior parameters (red), single-site (green) and multi-site (brown) posterior parameters. The size of the circle indicates the number of observations within each circle, and the error bars depict the standard deviations in the mean of observations from the variability within each circle. Lines connect the mean values of simulations and pale bands depict the standard deviation in the mean of simulations from the variability within each bin.





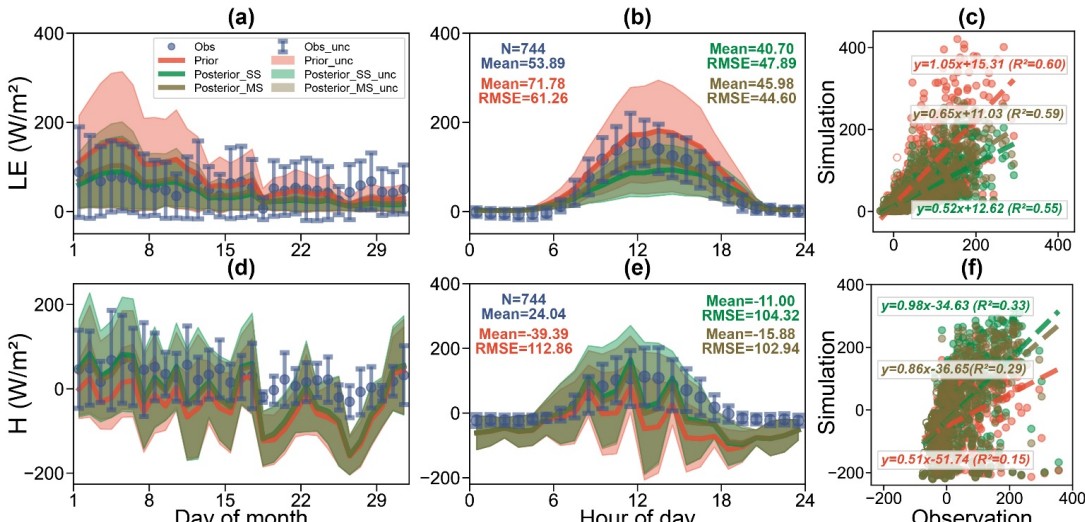

**Figure 8.** Daily variation (a and d), diurnal cycle (b and e) and scatter (c and f) plots of LE and H at FI-Hyy in August 2014. Observations (blue) are compared to simulations using prior (red) parameters, single-site (green) and multi-site (brown) posterior parameters. In the daily variation and diurnal plots, the size of the circle indicates the number of observations within each circle, and the error bars depict the standard deviations in the mean of observations from the variability within each circle if the number of corresponding observations is greater than three. Lines connect the mean values of simulations and pale bands depict the standard deviation in the mean of simulations from the variability within each bin. And in the scatter plots, the daytime data (6:00-18:00LT) and nighttime data (18:00-6:00LT) are represented as solid and hollow circles respectively.

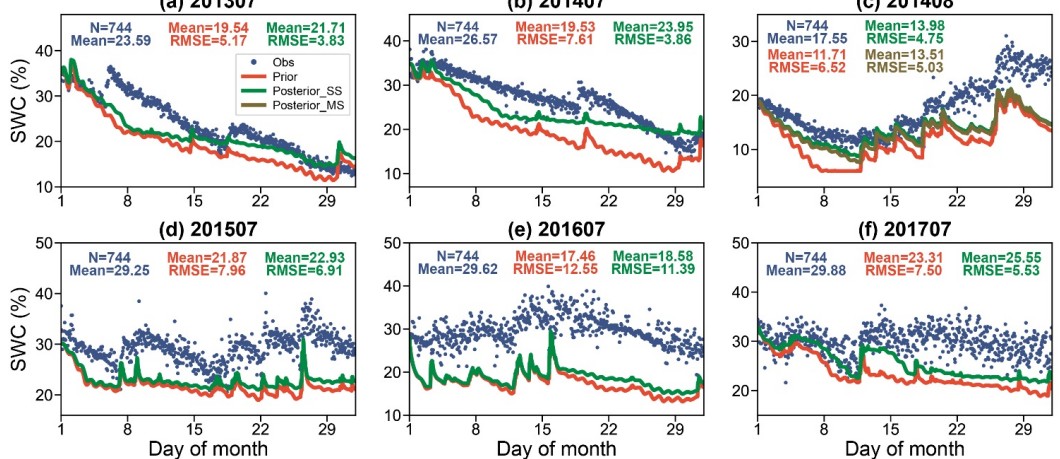

**Figure 9.** Observed (blue point) and simulated SWC (%) at FI-Hyy. Results show SWC simulated using prior parameters (red line), single-site (green line) and multi-site (brown line) posterior parameters.



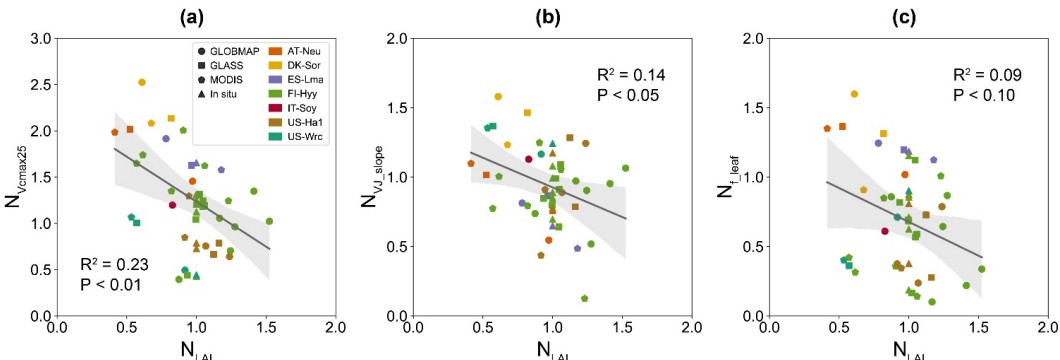

1195

**Figure 10. Influence of LAI on the posterior $V_{cmax25}$ (a), the posterior VJ_slope (b) and the posterior f_leaf (c) obtained by the single-site experiments conducted at seven sites and driven by four LAI data.** The posterior $V_{cmax25}$, the posterior VJ_slope and the posterior f_leaf and the LAI were represented by their normalized values $N_{Vcmax25}$, $N_{VJ\_slope}$, $N_{f\_leaf}$ and $N_{LAI}$, respectively. The posterior parameters

1200 were normalized by their prior values and the LAI were normalized by the in situ values. The linear regression fit lines of the posterior parameters obtained based on the satellite-derived LAI with the corresponding LAI data is shown, with 95% confidence intervals spread around the lines.

**Table 1. Site characteristics. Site identification includes the country initials and a three-letter name for each site; locations of the sites are provided by the latitude (Lat) and longitude (Lon); PFTs covered by the sites are evergreen needleleaf forest (ENF),**
1205 **deciduous broadleaf forest (DBF), grass, shrub and crop; Soil texture covered by the sites are silty clay loam, silty clay, loamy sand, loam and silty loam.**

| Site Name | Lat (°N) | Lon (°E) | PFT | Soil texture | LAI ($m^2/m^2$)* | year | References |
|---|---|---|---|---|---|---|---|
| AT-Neu | 47.12 | 11.32 | grass | sandy clay | 4.7 | 2015 | Spielmann et al. (2019) |
| DK-Sor | 55.49 | 11.64 | DBF | clay | 5.0 | 2016 | Spielmann et al. (2019) |
| ES-Lma | 39.94 | -5.77 | shrub | clay | 1.82 | 2016 | Spielmann et al. (2019) |
| FI-Hyy | 61.85 | 24.29 | ENF | sandy loam | 4.0 | 2013-2017 | Kohonen et al. (2022) |
| IT-Soy | 45.87 | 13.08 | crop | sandy clay | 2.3 | 2017 | Spielmann et al. (2019) |
| US-Ha1 | 42.54 | -72.17 | DBF | silty loam | 5.0 | 2012-2013 | Wehr et al. (2017) |
| US-Wrc | 45.82 | -121.95 | ENF | sandy clay loam | 8.7 | 2014 | Rastogi et al. (2018) |

*\* Mean LAI during the experimental period*

**Table 2. Configuration and assimilation result of each twin experiment. $J_{initial}$ and $J_{final}$ denote the initial value and the final value of the cost function $J(x)$ respectively, $G_{initial}$ and $G_{final}$ denote the initial value and the final value of the gradient respectively.**

| Site | Assimilation window | Perturbation | $J_{initial}$ | $J_{final}$ | $G_{initial}$ | $G_{final}$ |
|---|---|---|---|---|---|---|
| AT-Neu | June 2015 | 0.4 | 2.31E+04 | 2.70E-14 | 1.91E+04 | 3.14E-05 |
| DK-Sor | June 2016 | 0.4 | 3.20E+04 | 2.34E-16 | 2.54E+04 | 8.28E-05 |
| ES-Lma | May 2016 | 0.4 | 4.58E+03 | 1.63E-18 | 3.94E+03 | 1.22E-06 |
| | July 2013 | 0.2 | 1.05E+04 | 4.99E-16 | 1.66E+04 | 2.77E-05 |
| | July 2014 | 0.2 | 1.56E+04 | 1.51E-16 | 2.44E+04 | 6.41E-05 |
| | August 2014 | 0.2 | 7.76E+03 | 1.87E-18 | 1.20E+04 | 1.49E-06 |
| FI-Hyy | July 2015 | 0.2 | 7.95E+03 | 4.01E-19 | 1.33E+04 | 8.42E-07 |
| | July 2016 | 0.2 | 1.20E+04 | 1.01E-14 | 1.92E+04 | 2.18E-04 |
| | July 2017 | 0.2 | 9.27E+03 | 8.35E-16 | 1.55E+04 | 1.48E-04 |
| IT-Soy | July 2017 | 0.4 | 1.72E+04 | 3.50E-13 | 1.42E+04 | 2.79E-04 |
| US-Ha1 | July 2012 | 0.4 | 6.85E+04 | 1.61E-14 | 5.48E+04 | 8.54E-05 |
| | July 2013 | 0.4 | 7.76E+04 | 8.21E-16 | 6.23E+04 | 2.65E-05 |
| US-Wrc | August 2014 | 0.2 | 1.13E+04 | 6.90E-15 | 1.78E+04 | 6.69E-05 |
| Multi-site | August 2014 | 0.2 | 1.70E+04 | 3.17E-14 | 2.68E+04 | 1.41E-04 |

1210





**Table 3.** The configuration and the relative changes (%) of the parameters for each single-site assimilation experiment. The minimization efficiency of each experiment is indicated by the reduction rate between the initial value of cost function ($J_{initial}$) and the final value of cost function ($J_{final}$), defined as $1 - J_{final}/J_{initial}$, and $N_{cos}$ denotes the number of ecosystem COS flux observations.

| Site | Assimilation window | $N_{cos}$ | Cost function reduction (%) | Relative change (%) of parameters | | | | |
|---|---|---|---|---|---|---|---|---|
| | | | | $V_{cmax25}$ | VJ_slope | Ksat | b | f_leaf |
| AT-Neu | June 2015 | 483 | 1.64 | 45.54 | -45.42 | 0.1347 | -0.1583 | 1.77 |
| DK-Sor | June 2016 | 440 | 42.17 | 113.45 | 46.37 | 0.0233 | 0.0600 | 31.35 |
| ES-Lma | May 2016 | 278 | 10.48 | 62.60 | -13.49 | 0.0412 | 0.0669 | 19.65 |
| FI-Hyy | July 2013 | 470 | 21.43 | 2.28 | 6.48 | 0.0067 | -0.0305 | -66.26 |
| | July 2014 | 479 | 62.23 | 5.60 | -2.79 | 0.0399 | -0.0859 | -89.93 |
| | August 2014 | 199 | 64.92 | -60.64 | -26.28 | 0.2223 | -0.3704 | -14.18 |
| | July 2015 | 457 | 14.74 | -3.74 | -48.22 | -0.0374 | 0.1939 | -13.29 |
| | July 2016 | 413 | 35.02 | -29.59 | -9.65 | 0.2689 | -0.3773 | -35.65 |
| | July 2017 | 513 | 53.71 | 34.79 | -4.66 | 1.3329 | -2.0845 | -78.09 |
| IT-Soy | July 2017 | 218 | 2.08 | 19.69 | 12.81 | 0.0049 | 0.0157 | -39.00 |
| US-Ha1 | July 2012 | 335 | 27.96 | -35.92 | 24.31 | 0.0060 | -0.0358 | -21.31 |
| | July 2013 | 514 | 58.10 | -24.54 | -11.15 | 0.1137 | -0.2864 | -76.31 |
| US-Wrc | August 2014 | 701 | 44.65 | -50.63 | 16.52 | 0.0860 | 0.0060 | -28.92 |

1215

**Table 4.** The configuration and the relative changes (%) of the parameters for the multi-site assimilation experiment at FI-Hyy and US-Wrc site. $N_{cos}$ denotes the total number of ecosystem COS flux observations.

| Site | Assimilation window | $N_{cos}$ | Cost function reduction (%) | Relative change (%) of parameters | | | | |
|---|---|---|---|---|---|---|---|---|
| | | | | $V_{cmax25}$ | VJ_slope | Ksat | b | f_leaf |
| FI-Hyy | August 2014 | 900 | 47.33 | -44.64 | 30.72 | 0.1837 | -0.2841 | -63.64 |
| US-Wrc | | | | | | 0.0963 | -0.0225 | |