# Peer review of "Assimilation of Carbonyl Sulfide (COS) fluxes within the adjointbased data assimilation system—Nanjing University Carbon Assimilation System (NUCAS v1.0)"

_EGUsphere, 2023_

## Referee Comment (RC1)

The paper by Zhu et al. presents an interesting study of data assimilation of carbonyl sulfide (COS) using the BEPS model. They used adjoint method to assimilate the COS fluxes as NUCAS v1.0. This is a new model tool to the modelling science and is useful for study of carbon cycle. The novelty of the model is that it assimilates COS flux to improve the model performance of GPP and other model parameters. Therefore, the research is within the scope of GMD and could be considered as publishable. However, there are some issues the authors should address before publication.

First of all, the adjoint code used in this paper is based on the automatic differentiation tool TAPENADE (Hascoët and Pascual, 2013). Yet, the authors did not validate the adjoint method or did not write it clearly. The question is: how do you justify that the adjoint codes will produce correct optimization?

Secondly, the logic of the paper is lost in some places. Section 3.7 and 3.8 showed results of comparison and evaluation of simulated H and LE, and SWC. But it is unclear how data assimilation of COS flux can impact those parameters, and the performance is less satisfactory than evaluations of COS fluxes and GPP. The question: is there causality between assimilation of COS fluxes and H, LE, and SWC? What is your hypothesis that COS fluxes are linked to H, LE and SWC? Consider adding details in Section 2.

Another recent paper By Cho et al. is worthy of a comparison and discussion: Cho, A., Kooijmans, L. M. J., Kohonen, K.-M., Wehr, R., and Krol, M. C.: Optimizing the carbonic anhydrase temperature response and stomatal conductance of carbonyl sulfide leaf uptake in the Simple Biosphere model (SiB4), Biogeosciences, 20, 2573–2594, https://doi.org/10.5194/bg-20-2573-2023, 2023.

Other minor comments:
Line 142: "For NUCAS, we use the same soil texture" to "we used the same soil texture."
Line 185: the sites used in the study is better to be shown in a Figure to give a general idea of the locations of those sites.
Line 197: "the $CO_2$ and COS mole fractions in the bulk air were assumed to be spatially invariant." What is the value of $CO_2$ and COS mole fractions in your case?
Line 227: "in situ" to "*in situ*", and all elsewhere.
Line 284: "For all cases where the PFT is evergreen needleleaf
forest, a perturbation ratio of 0.2 was used. And for the remaining six single-site twin experiments, a perturbation rate of 0.4 was used." Please specify the reasons to those perturbation rate as 0.2 or 0.4.
Line 425: "transpiration and soil water transport, and therefore provide an indirect constraint for improving the simulation of GPP, LE, H". Please specify LE and H, at the first place giving full names.
Line 440: "very reasonable". Is there another way to say "very"?
Line 450: "very similar". The same as Line 440. And check all elsewhere.
Line 513: "assimilation using COS observations from multiple sites can also improve GPP simulations, and the assimilation is sometimes", it is vague to use sometimes to describe results.
Line 1165: Figure 4, it is not easy to see clearly the green and gray shading. Please consider better visualization.

Ling 1170 and 1175: Figure 5 and 6, why there are error bars for some sites but no error bars for other sites?

Line 1185: Figure 8. It is hard to see difference between green and gray. The dots in c and f are maybe too big.

---

## Referee Comment (RC2)

Zhu et al. present a new assimilation model NUCAS v1.0 for simulating carbonyl sulfide (COS) fluxes at ecosystem scale. The model is a good addition to the COS modeling pool, but the study requires some modifications and the paper lacks important information and is in many places too ambiguous and inconsistent.

General comments:

The paper lacks consistency on terminology used throughout the paper. Examples: in Eq. 1 observation is marked with O and model with M while in Eq. 12 they are marked with c and s and in Eqs. 14-16 they are marked obs and sim, respectively. Soil moisture is sometimes marked with SWC and sometimes as Θ. Section 2.1.3 is full of examples (listed below in more detail). This makes the paper very difficult to follow for the reader.

The authors model soil and plant COS fluxes separately but only report the total ecosystem flux. However, it would be interesting to see the simulated soil and plant fluxes separately and see how they compare with measured chamber COS fluxes from the different sites and also with e.g. other soil models.

Some coefficients and uncertainty estimates used in the paper are very poorly explained. Where does a perturbation rate of 0.4 come for some sites while for others it is 0.2? How do the authors come up with an uncertainty of 1 pmol $m^{-2}$ $s^{-1}$ for the prior simulated COS flux (L275)? Section 2.1.3 is also filled with these coefficients, listed in more detail below.

The benefit of the "multi-site" assimilation is unclear since it produces more or less similar results as the single-site assimilation. This is primarily due to using only two sites in this assimilation. The use of the word "multi" is thus exaggerated and I suggest leaving this part totally out of the paper, since it does not bring any notable improvement to the model. I understand that using only two sites is due to lack of in-situ COS flux measurements in similar ecosystems, but I don't really see a point doing a two-site assimilation since the results will be very similar to single-site assimilation.

I have several comments regarding the use of measured COS flux data:
- all sites: The authors do not specify any quality criteria used to filter the measured fluxes. Usually eddy covariance flux data are given a quality flag from 0 to 2; 2 indicating poor quality fluxes that should not be used, 1 indicating medium quality fluxes that are fine for budget calculations and 0 indicating the best quality that should be used for functional relationships and modelling. Please specify if you have used quality filtering in the data and if not, please give reasons why.
- US-Wrc: The dataset provided by Rastogi et al. 2018 does include the ready calculated gradient fluxes, and it is unclear why the authors are not using those fluxes but give a very ambiguous explanation of their own gradient flux parameter calculations. Moreover, since US-Wrc fluxes were calculated partly from the simulated COS fluxes in this study, this introduces a huge bias to these fluxes which gives even more reason not use this site in the "multi"-site assimilation.
- FI-Hyy: The dataset provided in Vesala et al. 2022 and Kohonen et al. 2022 already include storage corrected COS fluxes and it is not clear why the author have decided to do another storage correction for this site but not to other sites. In addition, this dataset includes gap-filled COS fluxes and it is not clear if the authors have used the gap-filled fluxes or the direct measured fluxes since the authors have not given any information on quality filtering.

Simulation of sensible and latent heat fluxes as well as SWC seems quite out of place. Can you explain how COS fluxes should be related to sensible heat flux, and why assimilating COS fluxes should

improve simulated sensible heat flux and soil moisture? Simulated sensible heat flux has even a different direction than the measured one. I suggest to leave this part out of the paper.

The abstract is too ambiguous and no concrete results are given. The authors use expressions "various processes" and "various ecosystems" without providing any details that would be useful for the reader.

The authors need to mention in the method section if they use one-sided or all-sided LAI data, and if that applies everywhere in the paper or not. Also specify if negative fluxes mean uptake or emission. The word "significantly" is thrown around a lot, without any relation to statistical significance, it seems.

Section 2.1.3 needs to be rewritten, especially regarding the equations that are inconsistent and lacking information. Specifically:

- Where is $F_{cos,leaf}$ used in the model? It is not present in any other equations after Eq. 3
- The authors need to explain where the different coefficients (e.g., 1.94 and 1.56 in Eq. 3; 1.4, 1.0, 5.33, -0.45 in Eq. 4; 0.437 and 0.0984 in Eq. 6; -0.00986, 0.197, -9.31 in Eq. 9; -0.119, 0.110, -1.18 in Eq. 10 , and 0.28 and 14.5 in Eq. 11) come from; what they represent and what is the reference.
- What is $f_{sw}$ (how it is defined, is there an equation, what unit does it have and what kind of variation does it have) exactly.
- $V_{cmax}$; what is the unit and how do you get values (and which values) for it?
- $F_{cos,biotic}$ suddenly changes to $F_{\Theta g}$ in the switch from Eq. 7 to Eq. 8, if I got it right. Be consistent with the terms, as this is impossible to follow as a reader!! Also, where does $\Theta_i$ go in between these equations?? Is it switched to $\Theta_g$?
- How is "optimum soil moisture" defined? Optimum in terms of what?

In general, there is lot of repetition throughout the paper and the text could certainly be condensed.

Finally, I would like to see scatter plots in addition to the diurnal variation comparison, to better see how the model is able to simulate the COS fluxes and GPP.

Specific comments:

L19: "various processes" is too ambiguous

L25: "various ecosystems"; please specify which ecosystems

L26: "can significantly improve"; how much did it improve, which timescale, which ecosystem(s) etc?

L34: "carbon dioxide ($CO_2$)" since this is the first time

L 47-49: I don't really see a point in repeating the same references twice in the same sentence

L55: Wohlfahrt et al 2012 and Kooijmans et al 2019 present an empirical model for leaf relative uptake (the uptake ratio of COS and $CO_2$ at the leaf scale) but do not model COS flux itself

L58-60: This sentence is very unclear and I am not sure what the authors want to emphasize here. Please rephrase

L71-75: Please rephrase this sentence and preferably split it in two. At the moment it reads like Liu et al 1997 developed a model for simulating COS fluxes (which is not the case).

L78: Since you do not assimilate COS fluxes in all ecosystems existing, please specify which ecosystems you are talking about here

L79: Controlling factors in which time scale of variability? E.g., in yearly scale temperature and radiation are for sure the most important drivers for carbon fluxes since they drive the seasonality, but this might not be the case in sub-daily time scales.

L81: List the ecosystems

L96: all-sided or one-sided LAI?

L98: "phenology is driven by LAI" but isn't it the other way around?

L103: remove one "of the"

L148: "pmol/m2/s" -> should be pmol $m^{-2}$ $s^{-1}$ and units are not supposed to be written in italic. Check this everywhere in the paper, also with other units like $m^2$ $m^{-2}$

L153: "And the leaf-level" -> "The leaf-level"

L156: Are the conductances different for shaded and sunlit leaves?

L179: Do you perhaps mean Table S2?

L187-195: It is quite strange to cite here not the papers whose data you use but other papers from those same sites. Please cite the papers whose data you are using.

L189: ICOS is not defined (Integrated Carbon Observation System)

L199-201: Specify that you use ecosystem scale eddy covariance (or gradient) flux measurements.

Sect. 2.4.1: I don't understand how the authors decided that the GLOBMAP LAI product was too low for the DK-Sor site but not for other sites. I did not find this information from Spielmann et al. 2019, as the authors claim. Please elaborate.

L224-226: I am sure US-Ha1 site has some radiation data, at least PPFD data if not shortwave radiation, as well as air temperature and relative humidity. In-situ data is for sure better than the ERA5 data.

L235: Table 1 does not list soil measurement information (and not the references either)

L248-250: Now this is very confusing. In Kohonen et al 2020 the uncertainty is high **with low absolute fluxes**, the fact there is a stronger peak in negative fluxes is simply due to lack of observations of positive fluxes. In Kohonen et al. 2020 the negative fluxes are defined as uptake by the biosphere. In any case there should be no reason to remove either positive or negative fluxes, unless the quality criteria are not filled!

L254: "gross primary productivity" -> "GPP"; "sensible heat" -> "H"; "latent heat" -> "LE"

L257: Cite Reichstein 2005 for the nighttime partitioning method

L260: How is nighttime defined?

L280: "And as a…" -> "As a.."

L296: Do you really mean that only one set of model parameters is required, independent of the ecosystem type? I would assume e.g. Vcmax to be quite different for different ecosystems and PFTs.

L307: I don't understand where the number 76 comes from. In table S2 there are 11 different parameters and their values are repeated as a constant value to get to 76, but there are certainly not 76 different parameters?

L310: "correlation" -> "coefficient"

L330: "dozens" -> please give an exact number

L335-337: This sentence is too vague. Please be more specific.

L337-339: Where are these parameters used? Not in the COS model presented earlier

L353: 1.64% is very low, how do you explain that?

L357: Figure 3 comes in the text before Figure 2 is presented

L360: Could this have something to do with the dry conditions and stomatal limitations, discussed in Vesala et al. 2022 regarding the low COS fluxes at FI-Hyy in July and August 2014?

L378: "for all experiments" -> not true for IT-Soy and US-Ha1!

L385: Can this even be called an increase? In any case very low correlation coefficient.

L387: Why are the simulated nighttime fluxes unchanged?

L400: "due to high value of observation" or rather underestimation by simulation?

L412: I would not call two sites multiple sites….

L422: Can the ratio between PAR and SW really change that much? Why is it allowed to change so much?

L429: Either "In particular," or "Particularly"

L444: "underestimated (by 55.72%), …"

L444: "greatly increased"; how much?

L445: "…simulations of COS **flux** at FI-Hyy.."

L468: "forest sites (DK-Sor, FI-Hyy, US-Ha1, US-Wrc) compared to grassland and savanna (AT-Neu and ES-Lma)"

L489-491: GPP cannot be observed directly, it is always a model!!

L502: "excellent match" needs quantification

L513-515: Not a very convincing result with the multi-site assimilation though

L515-520: How would the results be without COS assimilation?

L523: It is not possible that there would not be sensible heat flux measured at a site where other eddy fluxes are measured, since it comes directly from the sonic anemometer used for wind measurements. If the authors have not published their sensible heat flux data, you can ask for it from the authors.

L525: "And the assimilation.." -> "The assimilation.."

L536 & L554-556: Refer to the supplement figs

L571: "not significant" by what metric? What is a "short period of time"?

L573: "almost no diurnal…" very vague, be more specific

L578-580: This is not really true, especially in the end of August (but other months are also underestimated)

L583-585: Refer to the supplement figs

L592: "COS fluxes of soil" -> "soil COS fluxes" or "COS fluxes from soil"

Sect 4.1: Would it make sense to limit f_leaf and Vcmax25 variability to reasonable scales?

L635: But since soil COS fluxes are low, wouldn't that lead to higher change in the parameters, to compensate for low fluxes..?

L652-655: Already mentioned in the previous section¨

L662: Could this be due to drought/ drier than normal conditions at FI-Hyy reported in Vesala et al. 2022?

L691: Table 1 perhaps?

L706: Which in-situ LAI data was used for FI-Hyy? Maybe the other one is all-sided and the other one-sided LAI?

L720: Start a new sentence "More laboratory…"

L728: Why are the authors not already refining the uncertainty of prior values in this study?

L735-738: Given that this is already known, why is the COS concentration variation not already taken into account in this model?

L749: Plants in lower rainfall conditions could also be e.g. CAM plants?

Data availability section: Please include also citations to all datasets used

Figure 1: How about mesophyll conductance? What does the dashed box represent?

Figure 2: Are there any boundary values given to the parameters? How are these normalized? Add a similar plot from each site to same figure (as subplots) and put the figure to the supplementary material.

Figure 3: I don't think these colors are color-blind friendly. Fig. 3 m: How is the RMSE in posterior lower, even though it looks worse than prior? Are the times presented here local time? For FI-Hyy the dataset is in local winter time (UTC +2). Please include the variability of the circle size (and what it means) to the figure legend. Why are you using mean instead of median diurnal variability?

Figure 4: I suggest to remove this fig with the whole "multi-site" analysis

Figure 5: Add in legend what the different colors mean. It is not clear from the caption what do the thick bars and the errorbars represent.

Figure 6: Same comments as for Fig. 5; you could combine these two figs in one as two different rows

Figure 7: same comments as for Figure 3.

Figure 8: Very weird pattern in simulated H. Solid and hollow circles are not distinguishable. I suggest to remove this fig with the analysis of H and LE.

Figure 9: Suggest to remove or move to supplement.

Figure 10: Not cited in the results section. What are "four LAI data"?

Table 1: Better reference to FI-Hyy would in this case be Vesala et al. 2022, since that paper presents the COS fluxes while Kohonen et al 2022 is about GPP.

Table 4: Suggest to remove.

Table S2: Not clear why the constant parameter values are repeated so many times

---

## Author Comment (AC1)

We would like to thank both reviewers for their detailed and insightful comments. These comments have helped improve and clarify the submitted manuscript. Below we reply to each comment point by point, showing the reviewers' comments in black and our responses in blue. Changes to the original manuscript are highlighted in **bold blue**. Note that the line numbers in the response are updated based on the revised manuscript, which we provide with our response..

We note already here that we reran all our numerical experiments, in response to two comments of Reviewer #2, one on the processing of COS flux observations and one on the prior uncertainty specified for the parameter f_leaf and to one comment by reviewer # 1 on the size of the perturbation for the starting point of the twin experiments.

Reviewer #1

The paper by Zhu et al. presents an interesting study of data assimilation of carbonyl sulfide (COS) using the BEPS model. They used adjoint method to assimilate the COS fluxes as NUCAS v1.0. This is a new model tool to the modelling science and is useful for study of carbon cycle. The novelty of the model is that it assimilates COS flux to improve the model performance of GPP and other model parameters. Therefore, the research is within the scope of GMD and could be considered as publishable. However, there are some issues the authors should address before publication.

Response: We thank the reviewer for this comment. We will address these issues in order to make this paper publishable in GMD.

First of all, the adjoint code used in this paper is based on the automatic differentiation tool TAPENADE (Hascoët and Pascual, 2013). Yet, the authors did not validate the adjoint method or did not write it clearly. The question is: how do you justify that the adjoint codes will produce correct optimization?

Response: We extended the text as follows "In this study, all derivative code is generated from the model code by the automatic differentiation tool TAPENADE (Hascoët and Pascual, 2013). **The derivative with respect to each parameter was validated against finite differences of model simulations, which showed agreement within the accuracy of the finite difference approximation.**" (Line 125-127)

Secondly, the logic of the paper is lost in some places. Section 3.7 and 3.8 showed results of comparison and evaluation of simulated H and LE, and SWC. But it is unclear how data assimilation of COS flux can impact those parameters, and the performance is less satisfactory than evaluations of COS fluxes and GPP. The question: is there causality between assimilation of COS fluxes and H, LE, and SWC? What is your hypothesis that COS fluxes are linked to H, LE and SWC? Consider adding details in Section 2.

Response: Since the leaf exchange of COS, carbon dioxide ($CO_2$) and water vapor are tightly coupled though stomata, COS has been proved as a useful tracer of photosynthesis, stomatal conductance and transpiration (Sandoval-Soto et al., 2005; Wohlfahrt et al., 2012). Transpiration is closely linked to soil moisture because the water it dissipates originates from the soil (Berry et al., 2006). This process of water turning from liquid to vapor requires energy,

and that energy is a crucial part of the ecosystem latent heat (LE) (Gupta et al., 2018). The energy is obtained from the surrounding leaf cells, leading to a decrease in temperature within the leaf (so called "cooling effect") (Gates, 1968; Gupta et al., 2018). Thus, the sensible heat (H) can be linked to transpiration since the leaf-to-air temperature gradient is a key control factor of it (Monteith and Unsworth, 2013; Dong et al., 2017). Therefore, our hypothesis is that the assimilation of COS is expected to improve the modelling of LE, H and SWC due to the ability of COS to indicate transpiration and the mechanism of transpiration (i.e. the corresponding energy transfer, cooling effect and water source).

We have added detailed in Section 2.3: "**Due to the coupling between leaf exchange of COS, $CO_2$ and $H_2O$, GPP and LE data are selected to evaluate the model performance of COS assimilation in this study. In addition, we further explored the ability of COS to constrain SWC as well as H simulations since the water dissipated in transpiration originates from the soil (Berry et al., 2006) and the transpiration contribute to a decrease in temperature within the leaf (so called "cooling effect") (Gates, 1968; Konarska et al., 2016).**" (Line 276-279)

Another recent paper By Cho et al. is worthy of a comparison and discussion: Cho, A., Kooijmans, L. M. J., Kohonen, K.-M., Wehr, R., and Krol, M. C.: Optimizing the carbonic anhydrase temperature response and stomatal conductance of carbonyl sulfide leaf uptake in the Simple Biosphere model (SiB4), Biogeosciences, 20, 2573–2594, https://doi.org/10.5194/bg-20-2573-2023, 2023

Response: Based on previous studies on the temperature response of carbonic anhydrase (CA), Rubisco enzyme and LRU, Cho et al. (2023) proposed a new COS plant uptake scheme for CA with the argument that different enzymes have different physiological characteristics. Through data assimilation, they combined COS and GPP observations with the Simple Biosphere model (SiB4) simulations to optimize stomatal conductance parameters b0 and b1, empirical parameter a, and CA enzyme optimum temperature, and thus improved the model performance of stomatal conductance, 'interior' conductance, and COS leaf uptake. This study provides new insights into achieving accurate modeling of COS plant uptake, which is worthy of comparison and discussion.

Firstly, precise modeling of carbonyl sulfide (COS) is fundamental for the utility of COS observations in optimizing model parameters associated with COS. The remarkable contribution of Cho et al. (2023) to COS modeling would undoubtedly benefit the work in utilizing COS as a probe to explore the ecological processes such as water-carbon exchange and energy flow within ecosystems.

Secondly, while the study by Cho et al. (2023) focused on optimizing COS-related and stomatal-related parameters, our investigation concentrates on refining parameters associated with photosynthesis and soil hydrology. Although the parameters optimized in our study influence stomatal modeling, our results reveal that the optimization of transpiration-related variables (LE, H, SWC) is comparatively less successful than that of COS and GPP. The insights gained from Cho et al. (2023)'s work underscore the potential for achieving improved optimization of transpiration-related variables by utilizing COS to directly constrain parameters associated with stomatal conductance.

Thus, we extended the text as follows: "This result is also proved by Resco De Dios et al. (2019), which found that the median $g_n$ in the global dataset was 40 mmol m$^{-2}$ s$^{-1}$. **Therefore, utilizing COS to directly optimize stomatal related parameters should be perused. Cho et al. (2023) has proven the effectiveness of optimizing the minimum stomatal conductance as well as other parameters by the assimilation of COS. Besides, with the argument that different enzymes have different physiological characteristics, Cho et al. (2023) proposed a new temperature function for the CA enzyme and showcase the considerate difference in temperature response of enzymatic activities of CA and RuBisCo enzyme, which also provided valuable insights into the modelling and assimilation of COS.**" (Line 701-706)

Other minor comments:

Line 142: "For NUCAS, we use the same soil texture" to "we used the same soil texture."

Response: Corrected.

Line 185: the sites used in the study is better to be shown in a Figure to give a general idea of the locations of those sites.

Response: Thanks for your suggestion, we have added such a figure to our manuscript, as shown below.

[Figure]

**Figure 2.** Locations of the 7 studied sites. Sites sharing the same plant function type are represented with consistent colors. The background map corresponds to the "Nature color I" map (https://www.naturalearthdata.com). ENF and DBF denote evergreen needleleaf forest and deciduous broadleaf forest, respectively.

Line 197: "the $CO_2$ and COS mole fractions in the bulk air were assumed to be spatially invariant." What is the value of $CO_2$ and COS mole fractions in your case?

Response: Thanks for your comment. we extended the text as follows: "The $CO_2$ and COS mole fractions in the bulk air were assumed to be spatially invariant over the globe and to vary annually. **The CO2 mole fraction data utilized in this study are taken from the Global Monitoring Laboratory (https://gml.noaa.gov/ccgg/trends/global.html). For the COS mole fraction, the average of the COS mole fraction observations from sites SPO (South Pole) and MLO (Mauna Loa, United States) was utilized to drive the model, the data are publicly available on line at: https://gml.noaa.gov/hats/gases/OCS.html.**" (Line 219-223)

Line 227: "in situ" to "*in situ*", and all elsewhere.

Response: Thanks for your reminder, we have changed to "*in situ*" throughout the manuscript.

Line 284: "For all cases where the PFT is evergreen needleleaf forest, a perturbation ratio of 0.2 was used. And for the remaining six single-site twin experiments, a perturbation rate of 0.4 was used." Please specify the reasons to those perturbation rate as 0.2 or 0.4.

Response: Thanks for your comment. The settings of the prior parameter uncertainties in this study refer to previous studies, e.g., Chen et al. (2022), Ryu et al. (2018). Now, the prior uncertainty of most model parameters was set to 25% of the prior value, while the prior uncertainty of f_leaf was estimated using the datasets provided by Ryu et al. (2018) and was about 7 % of the prior value. These studies also provide us with reference for understanding the degree of parameter variability and choosing the perturbation rate. Now, we chose a perturbation ratio (0.2) that falls between these two values (7 % and 25 %), but is closer to the prior uncertainty with most of the parameters, and reran all the twin experiments.

Line 440: "very reasonable". Is there another way to say "very"?

Response: Thank you for the suggestion. The relevant parts have been re-written in the revised manuscript.

Line 450: "very similar". The same as Line 440. And check all elsewhere.

Response: Thanks for your comment. The relevant parts have been re-written in the revised manuscript and we have checked all elsewhere.

Line 513: "assimilation using COS observations from multiple sites can also improve GPP simulations, and the assimilation is sometimes", it is vague to use sometimes to describe results.

Response: Thanks for your comment. We have reorganized the sentences to avoid vagueness.

Line 1165: Figure 4, it is not easy to see clearly the green and gray shading. Please consider better visualization.

Response: Thanks for your comment. We have remade our figures so that the results can be easily distinguished.

Line 1170 and 1175: Figure 5 and 6, why there are error bars for some sites but no error bars for other sites?

Response: Thanks for your comment. In this study, FI-Hyy and US-Ha1 are the only two sites with multi-year COS observations, which provides an opportunity to investigate the optimization results of COS-related parameters and the effectiveness of COS assimilation in different years. For these two sites, error bars were plotted to represent the maximum and minimum of the posterior parameter values. In contrast, no error bars were plotted for the other sites due to the lack of multi-year COS observations. We have described in the manuscript that we plotted error bars for sites with multiple years of COS observations. In response to your question, we have added a note to the figure legend of the revised manuscript: "**For those sites lacking multi-year COS observations, no error bars were plotted.**" (Line 1177-1178)

Line 1185: Figure 8. It is hard to see difference between green and gray. The dots in c and f are maybe too big.

Response: Thanks for your comment. We have reorganized the figure using smaller dots and

changed the colours for better visualization.

Review #2

Zhu et al. present a new assimilation model NUCAS v1.0 for simulating carbonyl sulfide (COS) fluxes at ecosystem scale. The model is a good addition to the COS modeling pool, but the study requires some modifications and the paper lacks important information and is in many places too ambiguous and inconsistent.

Response: We thank the reviewer for this comment. In response to this comment, we have refined the manuscript to enhance clarity and ensure consistency. The necessary information has been incorporated, rendering the manuscript comprehensive and informative.

General comments: The paper lacks consistency on terminology used throughout the paper. Examples: in Eq. 1 observation is marked with O and model with M while in Eq. 12 they are marked with c and s and in Eqs. 14-16 they are marked obs and sim, respectively. Soil moisture is sometimes marked with SWC and sometimes as $\Theta$. Section 2.1.3 is full of examples (listed below in more detail). This makes the paper very difficult to follow for the reader.

Response: We thank the reviewer for this comment. To enhance readability, we have revised the manuscript to ensure consistency in terminology. In the revised manuscript, we have designated observations as 'O' and the model as 'M.' Soil moisture is identified by 'SWC.' Furthermore, to mitigate ambiguity with 'C' in Eq.1, we now use 'F' to represent the corrected COS fluxes. Additional details regarding the rationale for utilizing corrected COS data from the US-Wrc site have been elaborated below.

The authors model soil and plant COS fluxes separately but only report the total ecosystem flux. However, it would be interesting to see the simulated soil and plant fluxes separately and see how they compare with measured chamber COS fluxes from the different sites and also with e.g. other soil models.

Response: Thanks for this valuable comment. Actually, there are many difficulties in evaluating COS soil and plant fluxes separately for the sites used in this study. The five-year COS ecosystem flux data at FI-Hyy provided us an opportunity to investigate the difference of assimilation performance of COS. However, the soil COS flux data at FI-Hyy are only available in 2015, which makes it impossible for us to separately evaluate COS plant flux and soil flux for the vast majority of experiments conducted at FI-Hyy. In addition, Whelan et al. (2022)have evaluated the model performance at FI-Hyy in 2015 and US-Ha1 using a similar soil model. At US-Wrc, only the raw COS concentration data at different altitudes are provided in Rastogi et al. (2018), while the values of the parameters needed to calculate the COS fluxes by the aerodynamic gradient method are not provided. Thus, there may be significant biases in our estimates of both plant and soil fluxes at US-Wrc. As for DK-Sor, ES-Lma and IT-Soy, a random forest regression model was trained for each site in order to simulate the soil COS exchange, and only the modelled COS soil fluxes are provided in Spielmann et al. (2019) while the observational data for COS soil flux is lacking. Overall, given the insufficient and inconsistent availability of separate COS soil and plant data, we face considerable obstacles in separately

assessing simulated COS soil and plant fluxes.

Additionally, in NUCAS, the resistance analog model of COS plant uptake and the empirical model of soil COS flux were embedded in the BEPS model, and the model performance of these COS models have been evaluated in numerous previous studies (Berry et al., 2013; Whelan et al., 2016; Kooijmans et al., 2021; Maignan et al., 2021; Whelan et al., 2022; Chen et al., 2023; Cho et al., 2023). These studies have demonstrated the usefulness and robustness of these models to simulate COS plant and soil fluxes, thus founded the basis for us to assimilate COS ecosystem flux in this study.

Last but not least, we do agree with your opinion and we also believe that assimilating the component fluxes of COS individually should be pursued in the future as this assimilation approach would provide separate constraints on different parts of the model. We expect the observational information on the partitioning between the two flux component to provide a stronger constraint than using just their sum.

Therefore, we extended the text in the conclusion: "**Specifically, with the lack of separate COS plant and soil flux data, the ecosystem-scale COS flux observations were utilized in this study. However, we believe that assimilating the component fluxes of COS individually should be pursued in the future as this assimilation approach would provide separate constraints on different parts of the model. We expect the observational information on the partitioning between the two flux components to provide a stronger constraint than using just their sum.**" (Line 739-743)

Some coefficients and uncertainty estimates used in the paper are very poorly explained. Where does a perturbation rate of 0.4 come for some sites while for others it is 0.2? How do the authors come up with an uncertainty of 1 pmol m$^{-2}$ s$^{-1}$ for the prior simulated COS flux (L275)? Section 2.1.3 is also filled with these coefficients, listed in more detail below.

Response: Thanks for your comment. Reviewer #1 asked a similar question about the choice of the perturbation size, please refer to our previous answer. Besides, we have changed the uncertainty of the prior simulated COS flux in twin experiments, and reperformed the experiments. Now, the uncertainty of the prior simulated COS flux was estimated as the standard deviation of the prior simulated COS fluxes within 24 hours around each simulation.

The benefit of the "multi-site" assimilation is unclear since it produces more or less similar results as the single-site assimilation. This is primarily due to using only two sites in this assimilation. The use of the word "multi" is thus exaggerated and I suggest leaving this part totally out of the paper, since it does not bring any notable improvement to the model. I understand that using only two sites is due to lack of in-situ COS flux measurements in similar ecosystems, but I don't really see a point doing a two-site assimilation since the results will be very similar to single-site assimilation.

Response: We appreciate the reviewer's understanding of the lack of *in situ* COS flux measurements in similar ecosystems. Therefore, we only performed a "multi-site" or "two-site" assimilation experiment at evergreen forest sites FI-Hyy and US-Wrc. Our two-site setup constitutes a challenge for the assimilation system, the model and the observations. In this setup the assimilation system has to determine a parameter set that achieves a fit to the observations

at both sites, and NUCAS passes this important test. NUCAS was designed as a platform that integrates multiple data streams to provide a consistent map of the terrestrial carbon cycle, although only ecosystem COS flux data were used to evaluate the performance of NUCAS in this study. The "two-site" assimilation experiment conducted in this study gives us more confidence that the calibrated model will provide a reasonable parameter set and posterior simulation throughout the plant functional type. In other words, what we present here is a pre-requisite for applying the model and assimilation system at regional to global scales. We did, however, replace the formulation "multi-site" by "two-site".

Also, we have extended the text in the conclusion: "**Our two-site setup constitutes a challenge for the assimilation system, the model and the observations. In this setup, the assimilation system has to determine a parameter set that achieves a fit to the observations at both sites, and NUCAS passes this important test.** It should be noted that the NUCAS was designed as a platform that integrates multiple data streams to provide a consistent map of the terrestrial carbon cycle although only ecosystem COS flux data were used to evaluate the performance of NUCAS in this study. **The "two-site" assimilation experiment conducted in this study gives us more confidence that the calibrated model will provide a reasonable parameter set and posterior simulation throughout the plant functional type. In other words, what we present here is a pre-requisite for applying the model and assimilation system at regional to global scales.**" (Line 744-751)

I have several comments regarding the use of measured COS flux data:

- all sites: The authors do not specify any quality criteria used to filter the measured fluxes. Usually eddy covariance flux data are given a quality flag from 0 to 2; 2 indicating poor quality fluxes that should not be used, 1 indicating medium quality fluxes that are fine for budget calculations and 0 indicating the best quality that should be used for functional relationships and modelling. Please specify if you have used quality filtering in the data and if not, please give reasons why.

Response: Thanks for this comment. In the dataset for FI-Hyy (Vesala et al., 2022), No quality flags are provided, but measured COS fluxes as well as gap-filled COS fluxes are provided. In this study, only the measured COS fluxes are utilized and we have provided additional clarification on this (Line 260-261). For US-Ha1 and US-Wrc, no quality flag or gap-filled data is provided. At the remaining four sites, "COS filter" flag was provided to mark whether the COS observations are without flux detection limits. In this study, we do not use the detection limits to filter the COS flux data because such filtering would cause us to lose all values close to zero.

- US-Wrc: The dataset provided by Rastogi et al. 2018 does include the ready calculated gradient fluxes, and it is unclear why the authors are not using those fluxes but give a very ambiguous explanation of their own gradient flux parameter calculations. Moreover, since US-Wrc fluxes were calculated partly from the simulated COS fluxes in this study, this introduces a huge bias to these fluxes which gives even more reason not use this site in the "multi"-site assimilation.

Response: Thanks for this comment. The dataset (https://zenodo.org/records/1422820)

provided by Rastogi et al. (2018) **does lack** readily available gradient fluxes. Consequently, we implemented a bias correction to align the simulated and estimated COS fluxes for the US-Wrc site, drawing upon methodologies outlined in previous studies (Leung et al., 1999; Scholze et al., 2016). In addition, we have reached out to the corresponding authors via email to kindly request assistance in obtaining their readily-calculated flux data. Unfortunately, as of now, we have not received a response.

We acknowledge that the absence of precise COS flux data at US-Wrc poses challenges to our two-site assimilation experiments. Nevertheless, we maintain the importance of conducting two-site experiment, as detailed before.

- FI-Hyy: The dataset provided in Vesala et al. 2022 and Kohonen et al. 2022 already include storage corrected COS fluxes and it is not clear why the author have decided to do another storage correction for this site but not to other sites. In addition, this dataset includes gap-filled COS fluxes and it is not clear if the authors have used the gap-filled fluxes or the direct measured fluxes since the authors have not given any information on quality filtering.

Response: Thank you for pointing this out. We deleted the sentence: "We then corrected the COS fluxes from FI-Hyy using the storage-correction method (Kooijmans et al., 2017)." At FI-Hyy, only the direct measured COS flux data were utilized in the assimilation experiments, and we have clarified this (Line 260-261).

Simulation of sensible and latent heat fluxes as well as SWC seems quite out of place. Can you explain how COS fluxes should be related to sensible heat flux, and why assimilating COS fluxes should improve simulated sensible heat flux and soil moisture? Simulated sensible heat flux has even a different direction than the measured one. I suggest to leave this part out of the paper.

Response: Thanks for this comment. Reviewer #1 asked a similar question, please refer to our previous answer.

In this study, the diurnal variability of the simulated sensible heat fluxes using the BEPS model exhibited misalignment with observations, mainly at FI-Hyy. However, the simulated sensible heat showed good agreement with observations at the remaining sites. Moreover, the optimization of H was demonstrated successfully at FI-Hyy, despite the different direction of the simulated sensible heat and the measured one.

The abstract is too ambiguous and no concrete results are given. The authors use expressions "various processes" and "various ecosystems" without providing any details that would be useful for the reader.

Response: Thanks for your comment. We have deleted the expression "variable ecosystems" and listed the corresponding ecosystems of our study site in detail.

The authors need to mention in the method section if they use one-sided or all-sided LAI data, and if that applies everywhere in the paper or not. Also specify if negative fluxes mean uptake or emission. The word "significantly" is thrown around a lot, without any relation to statistical significance, it seems.

Response: Thanks for this comment. The leaf area index is commonly defined as half the total all-sided developed area of green leaves per unit ground surface area (Chen and Black, 1992; Liu et al., 2012; Xiao et al., 2016). In the publications listed in **Table 1**, only Kohonen et al. (2022) specified that the all-sided leaf area index (LAI) of FI-Hyy was ca. 8 $m^2$ $m^{-2}$ during the measurement period (2013–2017). In this study, we followed the convention of using one-sided LAI (for broadleaves). We now have added "one-sided" (Line 99 and Line 1994) to account for this. In Sect. 2.4.3, we have specified positive values indicate COS uptake. Furthermore, we have corrected the inappropriate use of "significantly".

Section 2.1.3 needs to be rewritten, especially regarding the equations that are inconsistent and lacking information. Specifically:

- Where is Fcos,leaf used in the model? It is not present in any other equations after Eq. 3

Response: In eq.3, $F_{cos,leaf}$ represents the leaf-level COS uptake rate. For COS simulations,

BEPS uses the leaf-level resistance analog model of COS (Berry et al., 2013) with a two-leaf upscaling scheme (Chen et al., 1999) from leaf to canopy.

   - The authors need to explain where the different coefficients (e.g., 1.94 and 1.56 in Eq. 3; 1.4, 1.0, 5.33, -0.45 in Eq. 4; 0.437 and 0.0984 in Eq. 6; -0.00986, 0.197, -9.31 in Eq. 9; -0.119, 0.110, -1.18 in Eq. 10 , and 0.28 and 14.5 in Eq. 11) come from; what they represent and what is the reference.

Thanks for your comment, we have detailed the coefficients relevant to COS plant flux modeling (Eq. 3-6). For the COS soil model, we have updated them and detailed the coefficients currently used (please see Table S2 and Table S3 for details).

In NUCAS, the resistance analog model of COS plant uptake (Berry et al., 2013) were used. Such a model utilizes the COS mole fraction in the bulk air and the series conductance (conductance = 1/resistance) of the leaf system for COS (the terms in parentheses in Eq. 3) to calculate the flux of COS uptake. In the series conductance of the leaf system for COS, the stomatal conductance and laminar boundary layer conductance of COS are framed in reference to that of $H_2O$ vapor. The greater mass and larger cross section of COS restricts its diffusion relative to $H_2O$ in the stomatal pore by a factor of 1.94 and in the laminar boundary layer by 1.56 (Seibt et al., 2010; Stimler et al., 2010).

As for Eq. 5, we followed the modelling scheme of COS in the SiB (version 4.2) (Haynes et al., 2020), and we have provided additional clarification on this.

   - What is fsw (how it is defined, is there an equation, what unit does it have and what kind of variation does it have) exactly.

Response: Thanks for your comment, we renamed it to $f_w$. In sect. 2.1.3, we mentioned $f_w$ is a soil moisture stress factor describing the sensitivity of $g_{sw}$ to soil water availability. We have added the definition of $f_w$ to the appendix and also citations to the relevant literature, i.e. Ju et al. (2006).

- Vcmax; what is the unit and how do you get values (and which values) for it?

Response: The unit of $V_{cmax}$ is µmol m$^{-2}$ s$^{-1}$, we now added the detail calculation of $V_{cmax}$ in the appendix.

   - Fcos,biotic suddenly changes to FΘg in the switch from Eq. 7 to Eq. 8, if I got it right. Be consistent with the terms, as this is impossible to follow as a reader!! Also, where does Θi go in between these equations?? Is it switched to Θg?

Response: Thanks for this comment. To enhance readability, we have revised the manuscript to ensure consistency in terminology. In the soil COS model proposed by Whelan et al. (2016), The soil abiotic COS flux corresponding to a soil moisture of $SWC_i$ can be calculated by Eq. 7 (Eq. 9 in the revised manuscript). In Eq. 7, $SWC_{opt}$ denote the optimum soil moisture, at which soil abiotic COS flux reaches a maximum ($F_{opt}$), $SWC_g$ denote a certain soil moisture, which is greater than $SWC_{opt}$ and whose corresponding soil abiotic emissions are known. The last constant (a) that needs to be known in Eq. 7 can be calculated by Eq. 8 (Eq. 10 in the revised manuscript).

   - How is "optimum soil moisture" defined? Optimum in terms of what?

Response: According to Whelan et al. (2016) and Whelan et al. (2022), there exists an optimum soil moisture at which the simulated biotic COS flux is maximized, i.e. optimum in terms of COS soil biotic uptake.

In general, there is lot of repetition throughout the paper and the text could certainly be condensed.

Response: Thank for your suggestion. We have thoroughly reviewed our manuscript and made refinements to the text.

Finally, I would like to see scatter plots in addition to the diurnal variation comparison, to better see how the model is able to simulate the COS fluxes and GPP.

Response: Thank for your suggestion. We now plotted the corresponding scatterplots and added them to the supplement.

Specific comments:

L19: "various processes" is too ambiguous

Response: Thanks for this comment. We have deleted the expression "variable ecosystems".

L25: "various ecosystems"; please specify which ecosystems

Response: we now specified the ecosystems, including evergreen needleleaf forest, deciduous broadleaf forest, C3 grass and C3 crop, respectively.

L26: "can significantly improve"; how much did it improve, which timescale, which ecosystem(s) etc?

Response: Thanks for this comment. Now we rewrite this sentence.

**Comparing prior simulations with validation datasets, we found that the assimilation of COS can significantly improve the model performance in gross primary productivity,**

**sensible heat, latent heat and even soil moisture.** (L26-L27)

L34: "carbon dioxide (CO2)" since this is the first time

Response: Corrected.

L 47-49: I don't really see a point in repeating the same references twice in the same sentence

Response: Thanks for this comment. We have revised the references in the manuscript.

**Recently, carbonyl sulfide (COS) has emerged as a promising proxy for understanding terrestrial carbon uptake and plant physiology (Montzka et al., 2007; Campbell et al., 2008) since it is taken up by plants through the same pathway of stomatal diffusion as $CO_2$ (Goldan et al., 1988; Sandoval-Soto et al., 2005; Seibt et al., 2010) and completely removed by hydrolysis without any back-flux in leaves under normal conditions (Protoschill-Krebs et al., 1996; Stimler et al., 2010).** (Line 47-51)

L55: Wohlfahrt et al 2012 and Kooijmans et al 2019 present an empirical model for leaf relative

uptake (the uptake ratio of COS and CO2 at the leaf scale) but do not model COS flux itself

Response: Thanks for this comment. We now deleted these two references.

L58-60: This sentence is very unclear and I am not sure what the authors want to emphasize here.

Please rephrase

Response: Thanks for this comment. As mentioned earlier, a crucial hypothesis in this study is that the assimilation of COS is expected to improve the modelling of LE, H and SWC due to the ability of COS to indicate transpiration and the mechanism of transpiration. Therefore, here we would like to emphasize the second half of the sentence, i.e., only few experiments were conducted to systematically assessed the ability of COS to simultaneously constrain photosynthesis, transpiration and other related processes in ecosystem models. Of course, We also mentioned COS observations here (in the first half of the sentence). That is because the lack of COS measurements is for sure an essential limiting factor in examining the ability of COS to constrain ecosystem processes, such as photosynthesis and transpiration. At the same time, we also believe that the mention of observations here can also serve to pave the way for the introduction of data assimilation below. Therefore, we have rewritten the sentence while retaining the main content. The revised sentence now reads as: **However, with the lack of ecosystem-scale measurements of the COS flux (Brühl et al., 2012; Wohlfahrt et al., 2012; Kooijmans et al., 2021), only few studies were conducted to systematically assess the ability of COS to simultaneously constrain photosynthesis, transpiration and other related processes in ecosystem models.** (Line 58-61)

L71-75: Please rephrase this sentence and preferably split it in two. At the moment it reads like Liu et al 1997 developed a model for simulating COS fluxes (which is not the case).

Response: Thank for this suggestion. We have split it in two:

**In this study, we present the newly developed adjoint-based Nanjing University Carbon**

**Assimilation System (NUCAS) v1.0. NUCAS v1.0 is designed to assimilate multiple observational data streams including COS flux data to improve the process-based Biosphere-atmosphere Exchange Process Simulator (BEPS) (Liu et al., 1997), which has been specifically extended for simulating the ecosystem COS flux with the advanced two-leaf model that is driven by satellite observations of leaf area index (LAI).** (Line 72-76)

L78: Since you do not assimilate COS fluxes in all ecosystems existing, please specify which ecosystems you are talking about here

Response: Corrected.

L79: Controlling factors in which time scale of variability? E.g., in yearly scale temperature and radiation are for sure the most important drivers for carbon fluxes since they drive the seasonality, but this might not be the case in sub-daily time scales.

Response: Thanks for your comment. We have reorganized and revised that question and question one " What are the main changes in the parameters through the assimilation of COS flux and which processes are constrained?" The revised sentence reads as follows: **What parameters are the COS simulation sensitive to and how do these parameters change in the assimilation of ecosystem scale COS flux data?** (Line 78-79) **Which processes are constrained by the assimilation of COS and what are the mechanisms leading to adjustments of the corresponding process parameters?** (Line 82-83)

Response: Thanks for your comment.

L81: List the ecosystems

Response: Corrected.

**To achieve these objectives, COS observations across a wide range of ecosystems (including evergreen needleleaf forest, deciduous broadleaf forest, C3 grass and C3 crop) are assimilated into NUCAS to optimize the model parameters using the four-dimensional variational (4D-Var) data assimilation approach, and the optimization results are evaluated against *in situ* observations.** (Line 85-88)

L96: all-sided or one-sided LAI?

Response: one-sided LAI.

L98: "phenology is driven by LAI" but isn't it the other way around?

Response: The BEPS model (Liu et al., 1997; Chen et al., 1999) used in this study is a process-based diagnostic model driven by remotely sensed leaf area index (Chen et al., 2019). In BEPS, LAI is used as an indicator of the current state of vegetation within an ecosystem, and the plant phenology is driven by LAI. In contrast, in prognostic models, LAI is used as a dynamic variable that evolves over time, and the prognostic models allow researchers to make predictions about how LAI will change in response to varying environmental conditions and disturbances.

L103: remove one "of the"

Response: Corrected.

L148: "pmol/m2/s" -> should be pmol m$^{-2}$ s$^{-1}$ and units are not supposed to be written in italic. Check this everywhere in the paper, also with other units like m$^2$ m$^{-2}$

Response: Thank for this comment. we have corrected the units in this manuscript.

L153: "And the leaf-level" -> "The leaf-level"

Response: Corrected.

L156: Are the conductances different for shaded and sunlit leaves?

Uniform leaf laminar boundary layer conductance was applied to both shaded and sunlit leaves. However, BEPS takes into account radiation transmission processes (e.g., direction and scattering) within the canopy and calculates the amount of radiation received by the sunlit and shade leaves accordingly. Thus, the sunlit and shade leaves have different photosynthesis rates in theory due to the different radiation they receive, and in turn have different stomatal conductance (Ball et al., 1987; Ju et al., 2010).

L179: Do you perhaps mean Table S2?

Response: Yes, we have corrected the clerical error here.

L187-195: It is quite strange to cite here not the papers whose data you use but other papers from those same sites. Please cite the papers whose data you are using.

Response: Thanks for this comment. This arose from the fact that certain literature corresponding to the sites from which we obtained data lacked detailed site descriptions. We have addressed this by including references to the papers from which we sourced the data.

L189: ICOS is not defined (Integrated Carbon Observation System)

Response: Corrected.

L199-201: Specify that you use ecosystem scale eddy covariance (or gradient) flux measurements.

Response: Corrected.

Sect. 2.4.1: I don't understand how the authors decided that the GLOBMAP LAI product was too low for the DK-Sor site but not for other sites. I did not find this information from Spielmann et al. 2019, as the authors claim. Please elaborate.

Response: Thanks for this comment. Mean LAI during the campaign of DK-Sor (referred to DBL in Spielmann et al. (2019)) was presented in Table S1 of the supplement in Spielmann et al. (2019).

L224-226: I am sure US-Ha1 site has some radiation data, at least PPFD data if not shortwave radiation, as well as air temperature and relative humidity. In-situ data is for sure better than the ERA5 data.

Thank you for your comment. We re-examined and collected the meteorological data of the

US-Ha1 site. As a FLUXNET site and an Ameriflux site, the meteorological data for the US-Ha1 can be found in both the Ameriflux and FLUXNET datasets, and both datasets does include some radiation data. However, the shortwave radiative data required by the BEPS model of US-Ha1 are only available at FLUXNET while only net radiation and PPFD data are available at Ameriflux. Considering the meteorological data of US-Ha1 provided by FLUXNET are only available in 1991-2012, we currently use FLUXNET data at US-Ha1 in 2012 and ERA5 shortwave radiation data with Ameriflux data in 2013 to drive the BEPS model.

L235: Table 1 does not list soil measurement information (and not the references either)

Thanks for your comment. Measurement information on COS soil fluxes already included in the literature we listed in Table 1 except for FI-Hyy. The reason we did not cite literature on soil COS flux observations at FI-Hyy (Sun et al., 2018) is that we assimilated ecosystem scale COS fluxes (Vesala et al., 2022) in this study. However, soil texture derived from the harmonized world soil database (Wieder et al., 2014) was used before. Now, we have updated the soil texture with *in situ* data and added relevant references (including Sun et al. (2018)).

L248-250: Now this is very confusing. In Kohonen et al 2020 the uncertainty is high with low absolute fluxes, the fact there is a stronger peak in negative fluxes is simply due to lack of observations of positive fluxes. In Kohonen et al. 2020 the negative fluxes are defined as uptake by the biosphere. In any case there should be no reason to remove either positive or negative fluxes, unless the quality criteria are not filled!

Thanks for your comment. Currently, we kept both positive and negative values of COS fluxes and re-ran the assimilation experiments.

L254: "gross primary productivity" -> "GPP"; "sensible heat" -> "H"; "latent heat" -> "LE"

Response: Corrected.

L257: Cite Reichstein 2005 for the nighttime partitioning method

Response: Corrected.

L260: How is nighttime defined?

Response: In light of the extended daylight hours during the Northern Hemisphere summer and to prevent misclassification of actual daytime hours as nighttime due to discrepancies in local longitude and locally adopted time, we fit the equation for the relationship between respiration and temperature based only on data from 21:00 local time to 3:00 the following day.

L280: "And as a…" -> "As a.."

Response: Corrected.

L296: Do you really mean that only one set of model parameters is required, independent of the ecosystem type? I would assume e.g., Vcmax to be quite different for different ecosystems and PFTs.

Response: Thanks for your comment. We absolutely recognize that e.g., $V_{cmax}$ varies greatly from ecosystem to ecosystem. In this study, we take the PFT- and texture-dependence of

parameters into consideration, thus the parameter number of one set of accurate and generalized model parameters is 76. In other words, the only one set of model parameters mentioned here, includes parameters that are specific to a PFT or texture but not to the point on the global that is populated by this PFT and characterized by this texture.

L307: I don't understand where the number 76 comes from. In table S2 there are 11 different parameters and their values are repeated as a constant value to get to 76, but there are certainly not 76 different parameters?

Response: The interdependence of parameters was considered in this study. Therefore, when counting the PFT-dependent parameters as well as the texture-dependent parameters, we multiply the number of PFTs and the number of textures considered in the BEPS model. This is how the number 76 is obtained.

L310: "correlation" -> "coefficient"

Response: Corrected.

L330: "dozens" -> please give an exact number

Response: We have modified this sentence with specific instructions.

L335-337: This sentence is too vague. Please be more specific.

Response: Thanks for your comment. We have reorganized the sentence: "**Corresponding to the PFT and soil texture of the experimental site, some PFT-dependent and texture-dependent parameters as well as global parameters showed different adjustments from others as they can affect the simulation of COS to different degrees.**"

L337-339: Where are these parameters used? Not in the COS model presented earlier

Response: We detailed how these parameters affect the simulation of COS in the appendix.

L353: 1.64% is very low, how do you explain that?

Response: As shown in the Figure 3j of the original manuscript, it is because the prior simulated COS at IT-Soy is already very close to the corresponding observations.

L357: Figure 3 comes in the text before Figure 2 is presented

Response: Corrected.

L360: Could this have something to do with the dry conditions and stomatal limitations, discussed in Vesala et al. 2022 regarding the low COS fluxes at FI-Hyy in July and August 2014?

Thanks for your comment. But according to Vesala et al. (2022), these months were not considered to be drought because the SWC remained at a normal level (well above $0.1 \text{ m}^3 \text{ m}^{-3}$). However, the SWC observations as well as simulations in August 2014 are indeed noticeably lower than the other months, and are close to the optimum soil moisture for the COS abiotic flux modelling (see Figure S9 for details). As a result, the prior simulated COS for that month were significantly overestimated by 41.06 %, resulting in $V_{cmax25}$ and VJ_slope being

considerable downward adjustments by -42.44 % and -41.03 % in the single-site experiments. Thus, the simulated GPP were also markedly downgraded by 53.54 % in August 2014, ultimately resulting in the underestimation of the single-site posterior simulated GPP. Regarding this, we have added the text in the manuscript: "**However, with a low SWC in August 2014, the prior simulated COS were obviously overestimated by 41.06 %, which led to remarkable downward adjustments of $V_{cmax25}$ as well as VJ_slope. Thus, the simulated GPP were also markedly downgraded by 53.54 % in August 2014, ultimately resulting in the underestimation of the single-site posterior simulated GPP.**" (Line 478-481)

L378: "for all experiments" -> not true for IT-Soy and US-Ha1!

Response: Corrected.

L385: Can this even be called an increase? In any case very low correlation coefficient.

Response: Yes, thus we say "$R^2$ remained almost unchanged by the optimizations".

L387: Why are the simulated nighttime fluxes unchanged?

Response: In the BEPS model, stomatal conductance was set to a constant value at night. Meanwhile, soil fluxes were small and less variable relative to the magnitude of plant COS flux.

L400: "due to high value of observation" or rather underestimation by simulation?

Response: Could, of course, be either, but according to Kooijmans et al. (2021), the air depleted in COS can then suddenly be captured by the EC system when turbulence is enhanced in the morning.

L412: I would not call two sites multiple sites….

Response: Now we changed our expression from 'multi-site' to 'two-site'.

L422: Can the ratio between PAR and SW really change that much? Why is it allowed to change so much?

Thanks for your comment. According to Ryu et al. (2018), the default f_leaf value in the BEPS model and the prior uncertainty of f_leaf in this study is overestimated. Thus, it tends to overshoot in the previous assimilation experiments. Now, we have computed the mean value of f_leaf with its standard deviation as an estimate of the error based on the MODIS PAR and SW data from 2012-2017 (Ryu et al., 2018) and re-ran the assimilation experiments.

L429: Either "In particular," or "Particularly"

Response: Corrected.

L444: "underestimated (by 55.72%), …"

Response: Corrected.

L444: "greatly increased"; how much?

Response: We have provided a quantitative description.

L445: "…simulations of COS flux at FI-Hyy.."

Response: Corrected.

L468: "forest sites (DK-Sor, FI-Hyy, US-Ha1, US-Wrc) compared to grassland and savanna (AT-Neu and ES-Lma)"

Response: Corrected.

L489-491: GPP cannot be observed directly, it is always a model!!

Response: Thanks for your comment. We know that GPP cannot be measured directly. In order to distinguish it from the modeled GPP of BEPS, we rephrase it to **GPP derived from EC measurements**.

L502: "excellent match" needs quantification

Response: Corrected.

L513-515: Not a very convincing result with the multi-site assimilation though

Thanks for your comment. Due to the lack of *in situ* COS observation data of the same PFT, we only conducted a two-site assimilation experiment. Therefore, we admit that the results of our experiments are not very convincing. More multi-site or two-site assimilation experiments would have helped us to get more statistically significant and plausible results, however we are faced with the challenge of lack of COS data.

L515-520: How would the results be without COS assimilation?

Response: the results be without COS assimilation, i.e., the prior simulation result can be found in Figure 4 and Figure 5 in the revised manuscript.

L523: It is not possible that there would not be sensible heat flux measured at a site where other eddy fluxes are measured, since it comes directly from the sonic anemometer used for wind measurements. If the authors have not published their sensible heat flux data, you can ask for it from the authors.

Response: Thanks for your suggestion. We have reached out to the corresponding authors via email to kindly request assistance in obtaining the sensible and latent heat flux data. With their assistance, we have conducted a thorough comparison and evaluation of H and LE simulations at the AT-Neu and IT-Soy sites. For the help they provided, we have added a note in the acknowledgements.

L525: "And the assimilation.." -> "The assimilation.."

Response: Corrected.

L536 & L554-556: Refer to the supplement figs

Response: Corrected.

L571: "not significant" by what metric? What is a "short period of time"?

Response: Thanks for your comment. Actually, this sentence is not necessary. We have therefore deleted it to avoid confusion.

L573: "almost no diurnal…" very vague, be more specific

Response: Thanks for your comment. We rewrite the sentence.

**However, the simulated SWC exhibited a clear diurnal cycle whereas the observed SWC had almost no diurnal fluctuations.** (Line 534-535)

L578-580: This is not really true, especially in the end of August (but other months are also underestimated)

Response: Thanks for your comment. We rewrote the sentence.

L583-585: Refer to the supplement figs

Response: Corrected.

L592: "COS fluxes of soil" -> "soil COS fluxes" or "COS fluxes from soil"

Response: Corrected.

Sect 4.1: Would it make sense to limit f_leaf and Vcmax25 variability to reasonable scales?

Response: Thanks for the comment. Since $V_{cmax25}$ and f_leaf have their physical significance, the optimized values of both should be within certain ranges, e.g., greater than zero. Currently, both are within their physical significance, despite the huge relative change of them. The magnitude of the adjustment of f_leaf is expected to be limited by improving the estimation of its prior uncertainty. However, the prior uncertainty we set of the parameter $V_{cmax25}$ is comparable to the existing dataset Chen et al. (2022). Furthermore, we have indeed refined the prior uncertainty of f_leaf and re-run the assimilation experiments.

L635: But since soil COS fluxes are low, wouldn't that lead to higher change in the parameters, to compensate for low fluxes?

Response: Thanks for the comment. The optimized parameter values are the result of the trade-off between the two parts of the cost function. When the reduction in the discrepancy between observation and simulation resulting from the adjustment of the parameters is not sufficient to offset the increase in the discrepancy between the current and prior parameter values, the adjustment is not continued.

L652-655: Already mentioned in the previous section¨

Response: Removed.

L662: Could this be due to drought/ drier than normal conditions at FI-Hyy reported in Vesala et al. 2022?

Thanks for your comment. As shown in Table 3 of the original manuscript, f_leaf has been greatly downregulated after the assimilation of COS. We believe that this inappropriate parameter value is the main reason for the underestimation of posterior simulation. Now, we have refined the prior parameter uncertainty and re-ran the assimilation experiment.

L691: Table 1 perhaps?

Response: Yes, now we corrected this error.

L706: Which in-situ LAI data was used for FI-Hyy? Maybe the other one is all-sided and the other one-sided LAI?

According to Kohonen et al. (2022), the all-sided leaf area index (LAI) of FI-Hyy was ca. 8 m2 m−2 during the measurement period (2013–2017). In this study, we followed the convention of using one-sided LAI, so the LAI at FI-Hyy is 4 $m^2$ $m^{-2}$, as listed in **Table 1**.

L720: Start a new sentence "More laboratory…"

Response: Corrected.

L728: Why are the authors not already refining the uncertainty of prior values in this study?

Thanks for your comment. We have currently referred to the relevant literature and refined the prior uncertainty of the parameters (as mentioned before). Specifically, as the COS data utilized in this study range from 2012-2017, only the Moderate Resolution Imaging Spectroradiometer (MODIS) PAR and shortwave radiation (SW) data ranging from 2012-2017 was used to calculated the mean and standard deviation of f_leaf, and the prior uncertainty of f_leaf was estimated as the calculated standard deviation. The MODIS PAR and SW datasets are publicly available at: http://environment.snu.ac.kr.

L735-738: Given that this is already known, why is the COS concentration variation not already taken into account in this model?

Response: Continuous COS concentration data are a pre-condition for continuous COS flux simulations based on COS concentrations due to the linear relationship between the two (Stimler et al., 2011; Berry et al., 2013). However, similar to COS flux data, the *in situ* observed COS concentrations are not continuous in the whole assimilation windows. Therefore, in order to perform continuous simulations of COS flux based on a variable COS concentration, Kooijmans et al. (2021) used the surface COS mole fraction fields retrieved from an atmospheric transport inversion performed with TM5-4DVAR. We also think that modelling and assimilation of COS fluxes based on spatially and temporally varying COS concentrations is an aspect of the NUCAS system that can be further enhanced, and we will strive to combine the ecosystem model with atmospheric transport model to address this issue in our next steps. However, **with the lack of *in situ* COS mole fraction data**, COS mole fractions in the bulk air are currently assumed to be spatially invariant over the globe and to vary annually in NUCAS, which may introduce significant errors into the parameter calibration.

L749: Plants in lower rainfall conditions could also be e.g. CAM plants?

Response: Thanks for your comment. According to the summary of species information used in Yu et al. (2019), they do not include the crassulacean acid metabolism (CAM) plants in the study. However, the CAM plants are indeed commonly found in harsh environments such as arid and semi-arid regions (Amin et al., 2019), and the main feature of stomatal conductance patterns in CAM plants is nocturnal opening (Males and Griffiths, 2017).

Data availability section: Please include also citations to all datasets used

Response: Done.

Figure 1: How about mesophyll conductance? What does the dashed box represent?

Response: Thanks for your comment. In the resistance analog model of COS plant uptake (Berry et al., 2013), the apparent conductance for COS uptake from the intercellular airspaces (include the mesophyll conductance and the biochemical reaction rate of COS and carbonic anhydrase) is represented by $g_{cos}$. The dashed box includes the driver data of BEPS, and those data were utilized in both diagnostic process and prognostic process.

Figure 2: Are there any boundary values given to the parameters? How are these normalized? Add a similar plot from each site to same figure (as subplots) and put the figure to the supplementary material.

Response: We didn't set any boundary values for the parameters. Currently, they are normalized by their prior values. We have carefully considered showing the convergence trajectory through the parameter space from the starting point of the iterative procedure to the final point. In fact, this trajectory is to a large extent arbitrary, because branches depend on specifics of the floating-point arithmetic/rounding, which depend in turn on aspects like computing platform, compiler, or even compiler flags. What both technically and scientifically matters are the values of parameters, cost function and its gradient at the starting and end points of the minimization. These are now provided in Tables S5 for the twin experiments and S4 and 2 for the experiments with real data. We thus refrain from including the trajectory plots into the manuscript or its supplement, but provide the corresponding graphs and their presentation (requested by the reviewer) here:

[Figure]

**Figure 1**. The evolution of model parameters with the number of iterations of cost function ($J_{iter}$) during the single-site experiments. Evolution (open carats and dashed lines) of soil texture dependent parameters is plotted on the right-hand y axis, evolution (filled circles and solid lines) of PFT-dependent parameters and global parameter is plotted on the left-hand y axis. Parameters are normalized by their prior values.

[Figure]

**Figure 2.** The evolution of model parameters with the number of iterations of cost function ($J_{iter}$) during the two-site experiment. Evolution (open carats and dashed lines) of soil texture (abbreviated as Txt) dependent parameters is plotted on the right-hand y axis, evolution (filled circles and solid lines) of PFT-dependent parameters and global parameter is plotted on the left-hand y axis. The texture-dependent parameters for FI-Hyy are denoted by "Txt3" and that of US-Wrc are denoted by "Txt4". Parameters are normalized by their prior values.

Corresponding to the PFT and soil texture of the experimental site, some PFT-dependent and texture-dependent parameters as well as global parameters showed different adjustments from others as they can affect the simulation of COS to different degrees. Those parameters are the maximum carboxylation rate at 25 °C ($V_{cmax25}$), the ratio of $V_{cmax}$ to maximum electron transport rate $J_{max}$ (VJ_slope), the scaling factors ($Ksat_{scalar}$ and ($b_{scalar}$) of saturated hydraulic conductivity (Ksat) and Campbell parameter (b), and the ratio of photosynthetically active radiation (PAR) to shortwave radiation (f_leaf). Particularly, as the soil textures at the FI-Hyy and US-Wrc are different, $Ksat_{scalar}$ and $b_{scalar}$ corresponding to these two soil textures were both optimized in the two-site twin experiment.

Figure 3: I don't think these colors are color-blind friendly. Fig. 3 m: How is the RMSE in posterior lower, even though it looks worse than prior? Are the times presented here local time? For FI-Hyy the dataset is in local winter time (UTC +2). Please include the variability of the circle size (and what it means) to the figure legend. Why are you using mean instead of median diurnal variability?

Response: Thanks for your suggestion. We have modified the color scheme of our figures to make them easier to read for the color-blind. Certainly, the times presented here are local time. We have included the variability of the circle size in the legend in the revised manuscript. We use the mean because it is sensitive to all values.

Figure 4: I suggest to remove this fig with the whole "multi-site" analysis

Response: Thanks for your suggestion. For a detailed explanation of the need for two-site experiments we as well, refer to the previous section. Therefore, we've left the experiment in the main manuscript but changed to "two-site". Additionally, we also added the explanation of the need for two-site experiment in the revised manuscript. (Line 744-751)

Figure 5: Add in legend what the different colors mean. It is not clear from the caption what do the thick bars and the error bars represent.

Response: Corrected.

Figure 6: Same comments as for Fig. 5; you could combine these two figs in one as two different rows

Response: Thanks for your suggestion. We have combined these two figures in one as two different rows.

Figure 7: same comments as for Figure 3.

Response: Thanks for your suggestion. We will modify the color scheme of our figures to make them easier to read for the color-blind. Certainly, the times presented here are local time. We will include the variability of the circle size. We use the mean because it is sensitive to all values

Figure 8: Very weird pattern in simulated H. Solid and hollow circles are not distinguishable. I suggest to remove this fig with the analysis of H and LE.

Response: Thanks for this comment. The less effective simulation of H by the BEPS model compared to other variables, i.e. LE has been confirmed in previous studies (Ju et al., 2006). We acknowledge that the different direction of the simulated sensible heat and the measured one was observed at FI-Hyy. However, the optimization of H was demonstrated successfully, including at the FI-Hyy site. The connection between COS and latent and sensible heat, and the hypotheses of this paper have already been explained in the previous section and we have put the corresponding figures in the supplement.

Figure 9: Suggest to remove or move to supplement.

Response: Thanks for this comment. The connection between COS and SWC, and the hypotheses of this paper have already been carefully explained in the previous section, and we have put the corresponding figures in the supplement.

Figure 10: Not cited in the results section. What are "four LAI data"?

Response: Thanks for this comment. We have cited this figure in the results section and specified these four types of LAI data.

Table 1: Better reference to FI-Hyy would in this case be Vesala et al. 2022, since that paper presents the COS fluxes while Kohonen et al 2022 is about GPP.

Response: Thanks for this comment. We've changed the reference.

Table 4: Suggest to remove.

Response: Thanks for this suggestion. The necessity of conducting two-site experiment, we have already explained in detail above in this response and now also provide the explanation in the revised manuscript on lines 744-751.

Table S2: Not clear why the constant parameter values are repeated so many time

Response: Thanks for your comment. This is due to the fact that we take into account the interdependence of parameters, and we actually optimize the scaling factor of Ksat and b in this study. Regarding this, we have modified the table (**Table S4** in the revised supplement) and restated the description of the parameters.

[revised manuscript text omitted]

*Mean one-sided LAI* (m² m⁻²) during the experimental period

**Table 2. Configuration and assimilation result of each twin experiment.** $J_{initial}$ and $J_{final}$ denote the initial value and the final value of the cost function $J(x)$ respectively, $G_{initial}$ and $G_{final}$ denote the initial value and the final value of the gradient respectively.

| Site | Assimilation window | Perturbation | $J_{initial}$ | $J_{final}$ | $G_{initial}$ | $G_{final}$ |
|---|---|---|---|---|---|---|
| AT-Neu | June 2015 | 0.4 | 2.31E+04 | 2.70E-14 | 1.91E+04 | 3.14E-05 |
| DK-Sor | June 2016 | 0.4 | 3.20E+04 | 2.34E-16 | 2.54E+04 | 8.28E-05 |
| ES-Lma | May 2016 | 0.4 | 4.58E+03 | 1.63E-18 | 3.94E+03 | 1.22E-06 |
| FI-Hyy | July 2013 | 0.2 | 1.05E+04 | 4.99E-16 | 1.66E+04 | 2.77E-05 |
| | July 2014 | 0.2 | 1.56E+04 | 1.51E-16 | 2.44E+04 | 6.41E-05 |
| | August 2014 | 0.2 | 7.76E+03 | 1.87E-18 | 1.20E+04 | 1.49E-06 |
| | July 2015 | 0.2 | 7.95E+03 | 4.01E-19 | 1.33E+04 | 8.42E-07 |
| | July 2016 | 0.2 | 1.20E+04 | 1.01E-14 | 1.92E+04 | 2.18E-04 |
| | July 2017 | 0.2 | 9.27E+03 | 8.35E-16 | 1.55E+04 | 1.48E-04 |
| IT-Soy | July 2017 | 0.4 | 1.72E+04 | 3.50E-13 | 1.42E+04 | 2.79E-04 |
| US-Ha1 | July 2012 | 0.4 | 6.85E+04 | 1.61E-14 | 5.48E+04 | 8.54E-05 |
| | July 2013 | 0.4 | 7.76E+04 | 8.21E-16 | 6.23E+04 | 2.65E-05 |
| US-Wre | August 2014 | 0.2 | 1.13E+04 | 6.90E-15 | 1.78E+04 | 6.69E-05 |
| Multi-site | August 2014 | 0.2 | 1.70E+04 | 3.17E-14 | 2.68E+04 | 1.41E-04 |

1495 **Table 2.** The configuration and the relative changes (%) of the parameters for each single-site assimilation experiment. The cost function reduction of each experiment is indicated by the reduction rate between the initial value of cost function ($J_{initial}$) and the final value of cost function ($J_{final}$), defined as $1 - J_{final}/J_{initial}$, and $N_{COS}$ denotes the number of ecosystem COS flux observations.

[revised manuscript text omitted]

The copyright of individual parts of the supplement might differ from the article licence.

[Figure]

**Figure S1.** Scatterplots of observed versus simulated hourly COS flux using prior (red) and single-site posterior (blue) parameters.

[Figure]

25  **Figure S2.** Hourly scatterplots of observed versus simulated hourly COS flux using prior (red), single-site (blue) and two-site (green) posterior parameters.

[Figure]

**Figure S3.** Hourly scatterplots of observed versus simulated hourly GPP using prior (red), single-site (blue) and two-site (green) posterior parameters.

[Figure]

**Figure S4.** The diurnal cycle of observed (black) and simulated LE using prior parameters (red), single-site (blue) and two-site (green) posterior parameters. The size of the circle indicates the number of observations within each circle (ranging from 1 to 31), and the error bars depict the standard deviations in the mean of observations from the variability within each circle. Lines connect the mean values of simulations and pale bands depict the standard deviation in the mean of simulations from the variability within each bin.

[Figure]

**Figure S5.** Scatterplots of observed versus simulated hourly LE using prior (red), single-site (blue) and two-site (green) posterior parameters.

[Figure]

**Figure S6.** The diurnal cycle of observed (black) and simulated H using prior parameters (red), single-site (blue) and two-site (green) posterior parameters. The size of the circle indicates the number of observations within each circle (ranging from 1 to 31), and the error bars depict the standard deviations in the mean of observations from the variability within each circle. Lines connect the mean values of simulations and pale bands depict the standard deviation in the mean of simulations from the variability within each bin.

[Figure]

**Figure S7.** Hourly scatterplots of observed versus simulated hourly H using prior (red), single-site (blue) and two-site (green) posterior
parameters.

[Figure]

**Figure S8.** Observed (black point) and simulated SWC (%). Results show SWC simulated using prior parameters (red line), single-site (blue line) and two-site (green line) posterior parameters.

[Figure]

**Figure S9.** Influence of LAI on the posterior VJ_slope, $Ksat_{scalar}$, $b_{scalar}$ and f_leaf obtained by the single-site experiments conducted at seven sites and driven by four LAI data (GLOBMAP, GLASS, MODIS and *in situ*). The posterior VJ_slope, $Ksat_{scalar}$, $b_{scalar}$, f_leaf and the LAI were represented by their normalized values $N_{VJ\_slope}$, $N_{Ksat_{scalar}}$, $N_{b_{scalar}}$, $N_{f\_leaf}$ and $N_{LAI}$, respectively. The posterior parameters were normalized by their prior values and the LAI were normalized by the *in situ* values. The linear regression fit line of the posterior parameters obtained based on the satellite-derived LAI (GLOBMAP, GLASS and MODIS) with the corresponding LAI data is shown, with 95% confidence interval spread around the line.

**Table S1. PFT and Soil Texture descriptions in BEPS model.**

| PFT No. | Descriptions |
|---|---|
| 1 | Evergreen needleleaf forest |
| 2 | Deciduous needleleaf forest |

| 3 | Deciduous broadleaf forest |
| 4 | Evergreen broadleaf forest |
| 5 | Mixed forest |
| 6 | Shrub |
| 7 | C3 grass |
| 8 | C3 crop |
| 9 | C4 grass |
| 10 | C4 crop |

| Soil texture No. | Description |
| --- | --- |
| 1 | Sand |
| 2 | Loamy sand |
| 3 | Sandy loam |
| 4 | Loam |
| 5 | Silt loam |
| 6 | Sandy clay loam |
| 7 | Clay loam |
| 8 | Silty clay loam |
| 9 | Sandy clay |
| 10 | Silty clay |
| 11 | Clay |

**Table S2.** $alpha$ and $beta$ parameters for COS production term.

| Site name | PFT in BEPS | PFT in Whelan et al. (2016) | $alpha$ (unitless) | $beta$ ($°C^{-1}$) |
| --- | --- | --- | --- | --- |
| AT-Neu | C3 grass | Savanna | -9.54 | 0.108 |
| ES-Lma | C3 grass | Savanna | -9.54 | 0.108 |
| DK-Sor | Deciduous broadleaf forest | Temperate forest | -7.77 | 0.119 |
| US-Ha1 | Deciduous broadleaf forest | Temperate forest | -7.77 | 0.119 |
| FI-Hyy | Evergreen needleleaf forest | Temperate forest | -7.77 | 0.119 |
| US-Wrc | Evergreen needleleaf forest | Temperate forest | -7.77 | 0.119 |
| IT-Soy | C3 crop | Soy field | -6.12 | 0.096 |

**Table S3. Parameters for COS uptake term.**

| PFT in BEPS | PFT in Whelan et al. (2022) | $SWC_{opt}$ (%) | $F_{opt}$ ( pmol m$^{-2}$ s$^{-1}$ ) with temperature (°C) at $SWC_{opt}$ | $SWC_g$ (%) | $F_{opt}$ ( pmol m$^{-2}$ s$^{-1}$ ) with temperature (°C) at $SWC_g$ |
| --- | --- | --- | --- | --- | --- |
| C3 grass | Grassland | 12.5 | $F_{opt}$: -4.5
$F_{T_g}$: -1.5
$T_{opt}$: -10.9 | 26.9 | $F_{opt}$: -2.3
$F_{T_g}$: -1.3
$T_{opt}$: -14.8 |

| | | | $T_g$: -25 | | $T_g$: -25 |
|---|---|---|---|---|---|
| Deciduous broadleaf forest | Forest - Temperate or broadleaf | 24.6 | 12.6 | 51 | -0.18$T$+0.48 |
| Evergreen needleleaf forest | Forest – Boreal or needleleaf | 12.5 | $F_{opt}$: -18 $F_{T_g}$: -12 $T_{opt}$: 28 $T_g$: 35 | 19.3 | $F_{opt}$: -5.9 $F_{T_g}$: -3.8 $T_{opt}$: 28 $T_g$: 35 |
| C3 crop | Agricultural | 17.7 | -9.7 | 22 | -5.36 |

**Table S4.** Description of parameters used for optimizations within the Nanjing University Carbon Assimilation System (NUCAS). Parameters are either specified per PFT, per soil texture, or globally, i.e., all PFTs and textures share one value, as indicated in column 3.

| No. | Parameter | Dependent | Unit | Description | Prior Value | Prior Uncertainty |
|---|---|---|---|---|---|---|
| 1 | | | | | 62.5 | 15.625 |
| 2 | | | | | 39.1 | 9.775 |
| 3 | | | | | 57.7 | 14.425 |
| 4 | | | | | 29 | 7.25 |
| 5 | $V_{cmax25}$ | PFT | $\mu$mol m$^{-2}$ s$^{-1}$ | maximum carboxylation rate at 25°C | 66 | 16.5 |
| 6 | | | | | 57.85 | 14.4625 |
| 7 | | | | | 48 | 12 |
| 8 | | | | | 84.5 | 21.125 |
| 9 | | | | | 30 | 7.5 |
| 10 | | | | | 30 | 7.5 |
| 11 | | | | | 2.39 | 0.5975 |
| 12 | | | | | 2.39 | 0.5975 |
| 13 | | | | | 2.39 | 0.5975 |
| 14 | | | | | 2.39 | 0.5975 |
| 15 | VJ_slope | PFT | unitless | Slope of the $V_{cmax}$ and $J_{max}$ (maximum electron transport rate) relationship | 2.39 | 0.5975 |
| 16 | | | | | 2.39 | 0.5975 |
| 17 | | | | | 2.39 | 0.5975 |
| 18 | | | | | 2.39 | 0.5975 |
| 19 | | | | | 2.39 | 0.5975 |
| 20 | | | | | 2.39 | 0.5975 |
| 21 | | | | | 0.046 | 0.0115 |
| 22 | | | | | 0.046 | 0.0115 |
| 23 | | | | | 0.046 | 0.0115 |
| 24 | Q10 | PFT | unitless | Soil respiration temperature factor | 0.046 | 0.0115 |
| 25 | | | | | 0.046 | 0.0115 |
| 26 | | | | | 0.046 | 0.0115 |
| 27 | | | | | 0.046 | 0.0115 |

| | | | | | | |
|---|---|---|---|---|---|---|
| 28 | | | | | 0.046 | 0.0115 |
| 29 | | | | | 0.046 | 0.0115 |
| 30 | | | | | 0.046 | 0.0115 |
| 31 | | | | | 6.2473 | 1.561825 |
| 32 | | | | | 6.2473 | 1.561825 |
| 33 | | | | | 6.2473 | 1.561825 |
| 34 | | | | | 6.2473 | 1.561825 |
| 35 | SIF_alpha | PFT | W m$^{-2}$ | Quadratic term coefficient for the relationship between additional heat dissipation under light adapted conditions and relative reduction of photochemical yield | 6.2473 | 1.561825 |
| 36 | | | | | 6.2473 | 1.561825 |
| 37 | | | | | 6.2473 | 1.561825 |
| 38 | | | | | 6.2473 | 1.561825 |
| 39 | | | | | 6.2473 | 1.561825 |
| 40 | | | | | 6.2473 | 1.561825 |
| 41 | | | | | 0.5994 | 0.14985 |
| 42 | | | | | 0.5994 | 0.14985 |
| 43 | | | | | 0.5994 | 0.14985 |
| 44 | | | | | 0.5994 | 0.14985 |
| 45 | SIF_beta | PFT | W m$^{-2}$ | Primary term coefficient for the relationship between additional heat dissipation under light adapted conditions and relative reduction of photochemical yield | 0.5994 | 0.14985 |
| 46 | | | | | 0.5994 | 0.14985 |
| 47 | | | | | 0.5994 | 0.14985 |
| 48 | | | | | 0.5994 | 0.14985 |
| 49 | | | | | 0.5994 | 0.14985 |
| 50 | | | | | 0.5994 | 0.14985 |
| 51 | | | | | 1 | 0.25 |
| 52 | | | | | 1 | 0.25 |
| 53 | | | | | 1 | 0.25 |
| 54 | | | | | 1 | 0.25 |
| 55 | | | | | 1 | 0.25 |
| 56 | $Ksat_{scalar}$ | texture | unitless | Scaling factor of saturated hydraulic conductivity (Ksat) | 1 | 0.25 |
| 57 | | | | | 1 | 0.25 |
| 58 | | | | | 1 | 0.25 |
| 59 | | | | | 1 | 0.25 |
| 60 | | | | | 1 | 0.25 |
| 61 | | | | | 1 | 0.25 |
| 62 | | | | | 1 | 0.25 |
| 63 | | | | | 1 | 0.25 |
| 64 | | | | | 1 | 0.25 |
| 65 | $b_{scalar}$ | texture | unitless | Scaling factor of Campbell parameter b (the exponential parameter of Campbell's soil moisture retention model) | 1 | 0.25 |
| 66 | | | | | 1 | 0.25 |
| 67 | | | | | 1 | 0.25 |
| 68 | | | | | 1 | 0.25 |

| | | | | | | |
|---|---|---|---|---|---|---|
| 69 | | | | | 1 | 0.25 |
| 70 | | | | | 1 | 0.25 |
| 71 | | | | | 1 | 0.25 |
| 72 | | | | | 1 | 0.25 |
| 73 | f_leaf | global | unitless | The ratio of photosynthetically active radiation to shortwave radiation | 0.5 | 0.125 |
| 74 | kc25 | global | μbar | Michaelis–Menten constants for $CO_2$ in 25°C | 274.6 | 68.65 |
| 75 | ko25 | global | mbar | Michaelis–Menten constants for $O_2$ in 25°C | 419.8 | 104.95 |
| 76 | tau25 | global | unitless | The $CO_2/O2$ specificity factor, which reflects the carbon assimilation efficiency of Rubisco | 2904.12 | 726.03 |

**Table S5.** Summary of configurations of twin experiments. $J_{initial}$ and $J_{final}$ denote the initial value and the final value of the cost function $J(x)$ respectively; $G_{initial}$ and $G_{final}$ denote the initial value and the final value of the gradient respectively; $D_{initial}$ and $D_{final}$ denote the initial value and the final value of the respectively. $D_{final}$ denote the final value of the distance ($D_x$) between the parameter vector and the prior parameter vector. The initial value ($D_{initial}$) of $D_x$ for all twin experiments is 7.48, due to an identical perturbation size (0.2) being applied.

| Site name | Data duration | $J_{initial}$ | $J_{final}$ | $G_{initial}$ | $G_{final}$ | $D_{final}$ | Relative changes of parameters (%) | | | | |
|---|---|---|---|---|---|---|---|---|---|---|---|
| | | | | | | | $V_{cmax25}$ | VJ_slope | $Ksat_{scalar}$ | $b_{scalar}$ | f_leaf |
| AT-Neu | June 2015 | 55.08 | 6.52E-16 | 48.09 | 6.65E-07 | 1.48E-07 | -8.13E-10 | -3.16E-09 | -6.88E-10 | -1.68E-09 | 1.24E-09 |
| DK-Sor | June 2016 | 77.13 | 7.45E-16 | 77.01 | 1.30E-06 | 1.70E-08 | 1.55E-09 | -8.85E-10 | -2.82E-09 | -1.08E-09 | -1.80E-09 |
| ES-Lma | May 2016 | 53.01 | 3.34E-15 | 51.59 | 1.55E-06 | 8.80E-10 | -1.06E-09 | 1.88E-09 | 8.54E-09 | 7.58E-09 | 4.26E-11 |
| FI-Hyy | July 2013 | 73.44 | 2.02E-17 | 70.43 | 1.10E-06 | 2.57E-08 | 1.29E-10 | 3.66E-10 | -9.30E-11 | 4.46E-10 | -2.01E-10 |
| | July 2014 | 77.59 | 1.06E-17 | 76.83 | 2.97E-07 | 4.74E-09 | 3.18E-10 | -6.80E-10 | -2.08E-11 | -1.96E-10 | -1.56E-10 |
| | August 2014 | 74.09 | 9.27E-18 | 70.00 | 4.63E-07 | 1.02E-09 | -7.33E-11 | 1.22E-10 | 5.99E-10 | 4.59E-10 | 2.20E-10 |
| | July 2015 | 72.76 | 1.19E-16 | 70.07 | 7.93E-07 | 7.58E-10 | -1.16E-10 | -4.87E-10 | 1.14E-11 | 7.20E-10 | 1.07E-09 |
| | July 2016 | 75.89 | 1.13E-18 | 73.35 | 2.12E-07 | 4.53E-08 | -9.64E-11 | 1.08E-10 | 3.16E-11 | 3.95E-11 | -5.55E-12 |
| | July 2017 | 73.94 | 8.47E-17 | 73.64 | 7.18E-07 | 2.45E-08 | 8.68E-11 | 7.31E-10 | 3.69E-12 | 2.01E-10 | 8.47E-10 |
| IT-Soy | July 2017 | 50.75 | 5.09E-13 | 38.82 | 4.94E-07 | 6.98E-08 | 2.86E-09 | -7.41E-09 | 2.74E-09 | -5.89E-09 | -5.70E-10 |
| US-Ha1 | July 2012 | 66.15 | 1.93E-19 | 59.66 | 2.05E-07 | 1.63E-07 | -6.01E-12 | 7.29E-11 | 1.35E-11 | 7.87E-11 | -5.81E-12 |
| | July 2013 | 66.50 | 1.61E-17 | 60.25 | 9.99E-07 | 2.36E-08 | 4.42E-09 | 7.44E-10 | -9.77E-11 | 4.07E-10 | -3.52E-11 |
| US-Wrc | August 2014 | 58.97 | 3.28E-18 | 46.87 | 1.45E-07 | 2.84E-08 | -1.16E-10 | 4.40E-10 | 1.22E-10 | -7.50E-11 | 6.04E-11 |
| FI-Hyy* | August 2014 | 108.04 | 3.95E-15 | 119.27 | 1.28E-06 | 2.01E-08 | -1.16E-10 | 4.40E-10 | 1.22E-10 | -7.50E-11 | 6.04E-11 |
| US-Wrc* | | | | | | | | | -3.41E-10 | 4.63E-10 | |

---

## Referee Report (RR1)

Zhu et al. have greatly improved the clarity of their manuscript and mostly made adequate changes and responses to the reviewer comments on the first round. However, I still have two major concerns regarding the paper.

Continuing the discussion related to the US-Wrc site gradient fluxes. Note that there are two papers by Rastogi et al. in 2018 using gradient flux data from the same site; one has data from 2014 and COS fluxes are (for whatever reason) not published (https://doi.org/10.1029/2018JG004430 ) and the other reports fluxes from year 2015 (https://doi.org/10.5194/bg-15-7127-2018 ). Dataset for the latter (including gradient fluxes of COS) can be found from: https://zenodo.org/records/1516332
To reduce the considerable bias the authors currently have regarding the calculation of US-Wrc fluxes, I highly recommend to use the published COS gradient fluxes from this site, from year 2015, and to rerun the analysis once more using this dataset. This would considerably reduce bias and improve the analysis. These data can then also be used in the two-site assimilation, which, I still in its current state (when gradient fluxes are first calculated using simulations, which then are again used to simulate fluxes) I do not approve of. If proper gradient fluxes provided by Rastogi et al (in the link above) are used, only then the two-site assimilation is possible. Note that Hyytiälä forest also has flux measurements in 2015.

The authors argue that sensible heat flux (H) and latent heat flux (LE) as well as soil water content (SWC) are related to COS fluxes because COS fluxes are related to transpiration. However, transpiration is only one part of ET (evapotranspiration, highly related to LE) and the other part is evaporation, which has no relation to COS fluxes. Evaporation and SWC are also highly related to water availability (precipitation) as well as other environmental variables (radiation, temperature). In addition, it is definitely not only the leaf-scale energy demand that controls the sensible heat flux at ecosystem scale. You forget soil, atmospheric turbulence, input energy from the sun, ground heat flux, evaporation, precipitation, saturation of SWC... Yes, COS fluxes could be used to estimate transpiration, but anything further is overinterpretation. Thus, I still very highly recommend completely leaving out the LE, H and SWC simulations.

From the response document:
*L400: "due to high value of observation" or rather underestimation by simulation?*
*Response: Could, of course, be either, but according to Kooijmans et al. (2021), the air depleted in COS can then suddenly be captured by the EC system when turbulence is enhanced in the morning.*

➔ This is why we do storage correction to EC fluxes! Storage corrected fluxes do **not** have this problem. I am not saying that observations would be perfect, but they are "the best guess" we have. Thus, I suggest to reformulate accordingly.

Specific comments:

How is this manuscript related to a preprint that is simultaneously in review (Zhu et al., 2024)? The other study seems very much related, and should be cited in this study as well.

The abstract and conclusions are still missing concrete results. The authors use descriptive words such as "improved"-> improved by how much or by what metric? Describe in detail (using numbers) what were the most important results of your study (e.g. how much (in %) did the assimilation improve the prior simulation etc).

Merge Figs 3 and 4 in a similar way as Fig. 6.

Eq. 9-10: $F_{cos,biotic}$ is switched to $F_{SWCg}$ and SWC to $SWC_g$ between the equations. Please check that is consistent.

References:

Rastogi, B., Berkelhammer, M., Wharton,S., Whelan, M. E., Itter, M. S., Leen, J. B.,et al. (2018). Large uptake ofatmospheric OCS observed at a moistold growth forest: Controls andimplications for carbon cycleapplications.Journal of GeophysicalResearch: Biogeosciences,123,3424–3438. https://doi.org/10.1029/2018JG004430

Rastogi, B., Berkelhammer, M., Wharton, S., Whelan, M. E., Meinzer, F. C., Noone, D., and Still, C. J.: Ecosystem fluxes of carbonyl sulfide in an old-growth forest: temporal dynamics and responses to diffuse radiation and heat waves, Biogeosciences, 15, 7127–7139, https://doi.org/10.5194/bg-15-7127-2018, 2018.

Zhu, H., Xing, X., Wu, M., Ju, W., and Jiang, F.: Optimizing the terrestrial ecosystem gross primary productivity using carbonyl sulfide (COS) within a "two-leaf" modeling framework, EGUsphere [preprint], https://doi.org/10.5194/egusphere-2023-3032, 2024.

---

## Referee Report (RR2)

Zhu et al. present a novel study that utilises the Nanjing University Carbon Assimilation System (NUCAS) v1.0 data assimilation framework and the process-based terrestrial ecosystem model Boreal Ecosystem Productivity Simulator (BEPS) is used as the adjoint model. The authors focus on simulating carbonyl sulfide (COS) fluxes at an ecosystem scale. Additionally, prior and posterior estimates of gross primary productivity (GPP), soil water content (SWC), sensible heat (H) and latent heat (LE) are presented. Assimilating COS flux measurements is a novel approach to better our understanding of COS processes, with regards to the carbon and water cycles, and the energy budget. This research is certainly within the scope of GMD and should be considered for publication following some minor corrections.

Having read the author's response to referee comments, Zhu et al. have clearly made a significant effort to improve their work. This is evidenced in the improved narrative of the paper and clarification on some of the methodology and data inputs. However, there is room for a bit more improvement in the readability, which my recommendations are focused on.

General Comments:

- The abstract would benefit from having at least one specific quantifiable metric that shows the improvements in COS flux or GPP made by NUCAS.

- There is use phrases like 'various constants' and other ambiguous phrases. I understand that elaborating in every instance could lead to excessive description of basic processes or repetition. But there is flexibility for some expansion on the use of these phrases. They often offer context and provide insight to readers who may not be familiar with this field, especially that of carbonyl sulfide which is rapidly becoming a field of its own within the world of carbon cycle science. An example on Lines 129-130: "The model parameters are the various constants that are not influenced by the model state."

- Sections 1 and 2 provide a thorough introduction to the topic and the methodology used in this work. However, the description of variables used in the modelling and assimilation is a bit light in places, leaving a lot of work for the reader to piece together. This harms to reproducibility of this work and makes it relatively inaccessible to a reader who is not familiar with matrix inversion, Bayesian statistics or data assimilation.
    o A sentence distinguishing biotic and abiotic soil processes would be helpful in Section 2.1.3.
    o Elaborate a bit on the parameters being referred to in Section 2.2.
    o Expand on the addition made in this iteration in Section 2.4.3, regarding coupling of COS with LE, H and SWC. It's still a bit light. At the very least include some additional references. Alternatively this could be elaborated on in the Discussion (Section 4), to which you could point the reader to from Section 2.4.3.
        ▪ After giving this point some further thought, perhaps elaboration would be best suited to when the main scientific questions are presented in Section 1. I will leave this up to the discrepancy of the author(s).

- Section 3 is a thorough summary of the results. However, there are 2 key points that could be improved, firstly some more summary of the implication of what the parameters mean, i.e. what does it mean if the relative change in VJ_slope is large for example. Does it mean the assimilation has changed the posterior results significantly with regards to the prior and if so, what does that actually tell us? The summary at the end of 3.6 is a good example of where this

has been done well. A few more sentences like that would be good! Secondly and a much more minor point, there could be more specific reference to which figure and sub-figure is being referred to. It doesn't need to be excessive but try and help the reader from having to constantly check tables to cross-reference the PFT and soil texture between sites when this is referenced, for example. A good example of this occurring is at the start of Section 3.7, where you introduce the topic of this section and then just point the reader to Figure S4-S7. Provide a bit more guidance and refer to them as they are discussed in turn – I have included some specifics in the minor comments.

- Generally, I would say that the results for H, LE and SWC don't bring a huge amount to the paper. And large portions could be condensed or moved to supplementary material. But it certainly **does not** detract from the paper. However, it somewhat distracts the reader from what I would consider very good results regarding posterior COS fluxes and GPP. This is meant as an opinion and how to deal with this can be left to the discretion of the authors.

- There seems to be interchangeable use of "COS flux data", "COS flux measurements", "COS data" and "COS measurements". There is a big difference between a measurement of COS fluxes and ambient COS concentration for example. A good example is the first sentence of section 2.5.2: "After the ability of NUCAS to assimilate COS flux data was confirmed by twin experiments, we could then use the system was then utilised to conduct data assimilation experiments with real COS observations under single-site and multi-site conditions" – Is "real COS observations" referring to measurements of the flux? Please be diligent in distinguishing this throughout the paper. From what I can tell, the only use of ambient COS concentrations is to drive the estimation of prior COS fluxes.

- Some sentences are unclear. I have highlighted potential improvements in the readability of these in the Minor Comments. Please read them careful to ensure there has not been a misinterpretation.

Minor Comments:
- L22-24: improve readability. Example: "Data assimilation experiments were conducted to investigate the robustness of NUCAS, and to test the feasibility and applicability of assimilating carbonyl sulfide (COS) fluxes from seven surface sites, in order to better our understanding of stomatal conductance and photosynthesis."
- L26: I assume you mean COS fluxes: "assimilation of COS fluxes can".
- L28: Be consistent in referring to NUCAS as "the NUCAS" or simply "NUCAS". Also see note for Line 73.
- L28: "to the" -> "with"
- L36: "of earth system" -> "of the Earth system"
- L37: the biosphere 'significantly mitigating climate change' is a bit speculative in my opinion. It has certainly reduced the full potential of climatic changes since 1850. But the reasoning is partly through feedbacks, such as longer growing seasons in the Northern Hemisphere, as a result of climate change. I would just end the sentence at 1850 (and of course keep the reference).
- L38-39: "of terrestrial biosphere have changed" -> "of the terrestrial biosphere has changed".
- L41: "important tool to investigate" -> "important tool used to investigate".
- L44: "data" -> "datasets"

- L45: The reference the Scholze et al. (2017) feels a bit random. Be more specific about the 'various observations' you're referring to at the beginning of the sentence or just remove the reference. I would just start the sentence: "Observations such as sun-induced.."
- L47: Not sure this is a 'recent' finding any more. Remove.
- L52-54: Move these two sentences (starting 'Plants' and 'As' to the paragraph above), as they are still discussing the relationship between CO2 and COS. The sentence afterwards moves into discussing COS and GPP modelling.
- L58: What other 'key ecosystem variables' are you referring to? Are there any others that COS can estimate better than direct measurement? The reason it's useful to estimate GPP is because it is impossible to measure GPP directly at large spatial scales. Perhaps be specific about what you are referring to, the reader might be interested.
- L58: "However, with the lack of" -> "However, due to the lack of".
- L58-61: Move the references to Wohlfahrt et al. (2012) and Kooijmans et al. (2021) to the end of the sentence. Remove Bruhl et al. (2012), I don't understand the context of that reference here.
- L62: "behavior" -> "behaviour".
- L65: remove "various".
- L68-71: I think it's clear you're referring to data assimilation techniques. This sentence could be reduced and read better, example: "More specifically, the observed dynamics of ecosystems can be more accurately portrayed, additionally, our understanding of ecosystem processes can be deepened, with respect to their responses to climatic changes."
- L73: Include a note that NUCAS v1.0 will be referred to as NUCAS for the remainder of the paper: "NUCAS v1.0, hereafter referred to as NUCAS, is designed"
- L73-74: Improved punctuation. "to assimilate multiple observational data streams including COS flux data to improve the process based" -> "to assimilate multiple observational data streams, including COS flux data, to improve the process-based".
- L77-78: An example of non-line-breaking hyphen. This will be the only in-text mention I refer to. **Please see Technical Note on this.**
- L82: Clarify "COS fluxes"
- L85: Again, are you referring to COS observations or COS flux observations?
- L88: Combine and clean these sentences: "Materials and methods used in our study are described in Sect. 2, such as the BEPS model and NUCAS, are introduced".
- L89: What data do you mean? Just be a bit more specific: "along with the data used to drive BEPS and assimilated into NUCAS."
- L104: "if" -> "is" and "First" -> "first".
- L119: Remove BFGS acronym definition. It isn't used again.
- L128-132: Keep this paragraph with the previous one. The discussion is still relevant, i.e. regarding cost function.
- L140: Change "new" to "updated". I wouldn't refer to literature written in 1999 as new.
- L153-155: Be more specific about what is being referenced here: "The canopy-level COS plant uptake $Fcos,plant$ (pmol m$^{-2}$ s$^{-1}$) was calculated by upscaling the resistance analog model of COS uptake, as presented by Berry et al. (2013), with the upscaling scheme recommended by Chen et al. (1999)."
- L161: No need to end the sentence. Also include another reference to Berry et al. (2013), as you are directly quoting the calculation: "where $COS_a$ is the COS mole fraction in the bulk air and $g_{sw}$ and $g_{bw}$ are the stomatal conductance and leaf laminar boundary layer conductance to water vapor (H$_2$O), respectively (Berry et al., 2013)."

- L164-165: Try to avoid starting a sentence with a lower case, even if it is a relevant parameter. Perhaps: "The apparent conductance for COS uptake from the intercellular airspaces is denoted by $g_{cos}$ and combines the mesophyll conductance and the biochemical reaction rate of COS and carbonic anhydrase (CA)."
- L166-169: Move the references to the end and lead the sentence into equation 4. Perhaps: "rate of Rubisco at 25℃ (Badger and Price, 1994; Evans et al., 1994), such that:
  *Eq. (4)*
  where α is a scaling parameter that is calibrated"
- L170: These values of α need a reference. If it's from Stimler et al. (2012), then move the reference in the previous sentence to here.
- L170-171: "With reference to the COS modelling scheme: Simple" -> "According to the COS modelling scheme: Simple"
- L178: End this sentence with : rather than . As you are referring to Eq. 7 anyway.
- L180: Same as above. : rather than .
- L182: Now start sentence with where, instead of Where.
- L187: "Here" -> "In Eq. 10," Also it is not clear what a is used for. Maybe add some context?
- L187: "maximum" -> "optimal".
- L190: "reference of" -> "reference to"
- L191: This is the first use of SWC as an abbreviation, soil define it, i.e. soil water content (SWC).
- L195: use the variable names, "COS plant uptake" -> "$F_{cos,plant}$" and COS soil fluxes -> "$F_{cos,soil}$". You can probably just remove this sentence to be honest. But up to the author.
- L198: "tuned previous model in development" -> "tuned in past model development"
- L200: Specifically refer to the research: "The prior uncertainty of parameters is set based on previous studies by Chen et al. (2022) and Ryu et al. (2018)."
- L222: "was" -> "were"
- L223-224: Sentence about LAI, meteorology and soil datasets need a reference. Or as they were mentioned in 2.1.2, perhaps direct the reader there.
- L226: "soil water content (SWC) at these sites collected at the sites were used" -> "SWC collected at the sites were used".
- L231: "LAI product represents Lead area index at a" -> "LAI product quantifies leaf area index".
- L231-233: spatial resolution is traditionally presented as 8 × 8 km for example. As it is a 2-D shape. Please clarify.
- L235: You may as well be specific in the products you are referring to. "The other two LAI products were used to investigate the effect of the LAI products" -> "The GLASS and MODIS LAI products were used to investigate the effect of different LAI products"
- L238-239: This last sentence is a bit wordy, but I understand what is to be communicated. Perhaps: "In addition, the 8-day temporal resolution of the LAI data was interpolated into daily values using the nearest neighbour method."
- L241-245: These few sentences could be a bit neater and consistent. Perhaps: "Standard hourly meteorological data was inputted in BEPS, including air temperature at 2 m, shortwave radiation, precipitation, relative humidity and wind speed, taken from the FLUXNET database (for sites: AT-Neu, DK-Sor, ES-Lma, FI-Hyy and US-Ha1 see https://fluxnet.org), the AmeriFlux database (for sites: US-Ha1, US-Wrc, see https://ameriflux.lbl.gov) and the ERA5 dataset (for sites: AT-Neu, IT-Soy, US-Ha1 see https://cds.climate.copernicus.eu/cdsapp#!/dataset/reanalysis-era5-single-levels?tab=overview)."
- L247: Move the reference to just before the comma. Also remove "Particularly" at the start of the next sentence.

- L257-258: I think the eddy-covariance and gradient-based approach both require a reference.
- 258-259: "The COS soil measurements were collected using soil chamber, except at US-Ha1, where a sub-canopy flux-gradient approach was used to calculate the soil COS flux." -> "The COS soil flux measurements were collected using soil chambers, except at US-Ha1, where a sub-canopy flux-gradient approach was used."
- L260: "COS flux measurements"
- L262-264: The first sentence of this paragraph needs to be reworded. There is very little context, and the ordering is confusing. Also, what is the aerodynamic gradient method? IS this the same as the gradient-based approach? Perhaps: "US-Wrc utilises the gradient-based approach to measure COS ecosystem flux (reference), however available data is limited to only COS concentration measurements and lacking other parameters required, therefore this site risks introducing biases." Or similar..
- L268-273: summary of Equation 11 could be much more concise. Perhaps: "This was done using the mean, $\overline{M}$, and standard deviation, $\sigma_m$, of the simulated COS flux to correct the COS flux observations ($O$):
  Eq. 11"
  where $\overline{O}$ and $\sigma_O$ are mean and standard deviation of the observed COS flux series. F is the corrected observed COS flux and the COS simulations were calculated using the prior parameters for the time period corresponding to the COS flux observations.
- L273: No need to start new paragraph. Move up to previous.
- L273: "as an estimate"
- L276: "LE data are selected" -> "LE, data were selected"
- L277: This needs a reference.
- L283: Move this reference to the end of the sentence.
- L306-307: "With reference" -> "Regarding"
- L309: "After the ability of NUCAS to assimilate COS flux data was confirmed by twin experiments, we could then use the system" -> "After the ability of NUCAS to assimilate COS flux data was confirmed by twin experiments, the system was then utilised"
- L312:
- L317-318: put comma (,) after experiment and simultaneously.
- L325: "in twin experiment" -> "in a twin experiment".
- L326: was calculated "using Eq. 12".
- L328: "where" -> "Where".
- L352: I only see $D_{final}$ in Table S5.
- L355: "nearly zero with the maximum value below" -> "nearly zero, where the maximum value was below".
- L356: pseudo-observations
- L358: .
- L364:  I read this as 6.35% in Table 2?
- L365: "with the cost function reduction of 16.39% and 15.70%." -> " of 16.39% and 15.70% respectively."
- L366: Include the percentage values for these 2 sites: "FI-Hyy (21.47%)" and "US-Wrc (27.71%)".
- L367-368: I believe you mean July of 2015. This sentence needs to be reworded, perhaps: "In August 2014 and July 2015, the cost function reduction was between 40.59 % and 50.94 %, while in July of all other years, the cost function reduction was much lower, ranging from 4.87 % to 18.94 %."

- L371-372: This is a bit of a throwaway sentence. If prior simulations were that good, we wouldn't need an inversion scheme right. I don't think sentence is necessary and almost devalues the posterior results.
- L377: I found this to be a particularly interesting finding. In that the way the COS fluxes are being calculated in the posterior are clearly missing one or more processes to exactly replicate measurements. Certainly something to investigate in future research and perhaps highlight in your conclusions.
- L382: You can remove the sentence starting "Similar to". I believe you raised this in the previous sentence.
- L384: 6.94  to 3.09 pmol m$^{-2}$ s$^{-1}$
- L387: missed a t: nighttime.
- L390: "FI-Hyy and US-Wrc have different soil textures, with sandy loam and loam, respectively." -> "FI-Hyy and US-Wrc have different soil textures; sandy loam and loam respectively."
- L391: "took this difference into account and" -> "accounted for this difference appropriately and".
- L392: I calculate this to be 26.28% but perhaps I have mixed up numbers. Please check.
- L408: If the two-site assimilation method achieved similar results to the single-site, why do we need the two-site? Was is it more of proof-of-concept? A sentence summarising why it was useful would be helpful.
- L410: regarding 'as mentioned before', it looks like this material has been moved to the appendix. Please check and update accordingly.
- Section 3.4: It isn't really clear if a positive or negative change is a good thing. Especially as the majority of the summary refers to absolute differences. Could this be elaborated on?
- Line 419: 45.09% surely this value is hugely skewed by ES-Lma? What is the value excluding this site?
- L425-427: my interpretation of IT-Soy is that on paper the RMSE is ok and improved in the posterior due to improvements during the daytime. However, it's minimal change from the prior suggests it is not particularly sensitive to assimilation of COS flux data.
- L430: remove capital T.
- L432: include at the end of this sentence "(note the difference in x-axis scales)". By eye it is misleading initially.
- L435-438: These values are very different. Is it appropriate to be comparing variables like-for-like in this way? A bit more explanation of the implications of the results would be helpful.
- L442: Unless you have specifically excluded DK-Sor, I would remove this.
- L444-445: [at end of sentence] respectively.
- L447: Maybe this is clearer: "Our results also suggest that f_leaf tends to play a more important role in the COS assimilation at the forest sites (DK-Sor, FI-Hyy, US-Ha1 and US-Wrc) compared to the low-stature vegetation type sites (AT-Neu, ES-Lma and IT-Soy), with the mean absolute SIs about two times than that of the latter, with the exception of DK-Sor." Optional.
- L445: Does a lower R$^2$ value not suggest that the assimilation has worsened the result? Also do you mean Figure S3?
- Section 3.6: Lots of plots being referred to. Include 'see Figure 6c' etc.. where necessary. Help the reader.
- L465: I calculate the 3.81% to be 8.61%. Please check.
- L471: GPP? Not COS.
- L472: Drop line after 'underestimated.'
- L484: struggling to get where these 2 percentages have come from.

- L495: Please refer to figures S4-S7 as they are discussed. Rather than just listing them at the start of a section. Also as a note, if you are having to discuss and refer to figures in supplementary material, it's probably a sign that you are trying to present too much. As mentioned earlier, you could probably remove sections 3.7 and 3.8.
- L551: Requires more references, bottom-up or top-down. Kooijmans et al. (2021), Ma et al. (2021), Maignon et al. (2021) and Remaud et al. (2022). For example.
- L567-569: "COS plant uptake is governed by the hydrolysis reaction of COS (Wohlfahrt et al., 2012), catalysed by CA, though it can also be degraded by other photosynthetic enzymes, e.g., RuBisCo (Lorimer and Pierce, 1989), and the reaction is not dependent on light (Stimler et al., 2011; Whelan et al., 2018)." I think reads a bit better. Optional.
- L586: Proven.
- L593: CA, not carbonic anhydrase.
- L595: "capable to influence" -> "capable of influencing"
- L599: "sensitivity of $V_{cmax25}$" -> "sensitivity in $V_{cmax25}$"
- L608-611: It's not clear if you're saying your work also found this. Please clarify and amend accordingly.
- L623-624: "In comparison, the RMSEs of GPP simulations were reduced by an average of 25.37% within the assimilation of COS, while that of LE were reduced by 16.27 %." -> "In comparison, the RMSEs of GPP simulations were reduced by an average of 25.37 % as a result of assimilating COS, but reducing LE by only 16.27 %."
- L630: at the end of sentence: "via evapotranspiration".
- L635: behaviour
- L639: 'remarkable differences' is an odd phrase. Be specific. Also if you mean large, this is a different narrative to Section 3.
- L641: BEPS
- L679: move reference to end of sentence.
- Some of the discussion in Section 4.5 is a bit wordy (mainly last paragraph). Below are a few instances of trying to improve readability and flow.
    - Final paragraph, sentence 2: "As the nighttime COS plant uptake is driven by stomatal conductance (Kooijmans et al., 2021), $g_n$, nighttime COS fluxes can therefore be used to test the capability of BEPS to model $g_n$.
    - L699: space between 1 and mm.
    - Final paragraph sentence 5: "Similar findings by Resco De Dios et al. (2019), showed that the median $g_n$ in the global dataset was 40 mmol m$^{-2}$ s$^{-1}$."
    - Final paragraph sentence 8: "As different enzymes have different physiological characteristics, Cho et al. (2023) proposed a new temperature function for the CA enzyme and showcased the considerable difference in temperature response of enzymatic activities of CA and RuBisCo, which provided valuable insights into the modelling and assimilation of COS."
    - L707: CA
    - L709: N = nitrogen?
    - L710: in -> by
    - Final paragraph final sentence: "Therefore, using the global microbial C biomass, soil N content and MAP datasets, the relationships between these variables, and the associated COS exchange processes, it is to be expected that a more accurate modelling of terrestrial ecosystem COS fluxes could be achieved, further increasing our understanding of the global COS budget and facilitate the assimilation of COS fluxes."

- L724-726: Perhaps: "Fourteen twin experiments, thirteen single-site experiments and one two-site experiment covering the period from 2012 to 2017, were conducted to investigate the capability of NUCAS to assimilate COS fluxes and optimize output parameters and variables. COS flux observations from a range of ecosystems were used, including four PFTs and three soil textures."
- L729: COS fluxes
- L749: "throughout the plant function type" -> "for different PFTs".
- L753-754: "However, the 'equifinality' can be avoided by imposing additional observational constraints (Beven, 2006)." Such as? i .e., 'in this instance, we refer to the calculation and assimilation of multiple datasets, other than just COS fluxes.'.

Technical Notes:

- Be consistent with the following phrases single-site, two-site, AmeriFlux, process-based, etc.. in terms of the use of hyphen and capitalisation. Please check other potential sources of inconsistency.

- Use non-line-breaking hyphen where possible. This way be resolved in the editing by EGU, but for future reference to avoid a hyphenated phrase, or unit, breaking a line, use ctrl+shit+-.

- % symbols should be immediately adjacent to values, not with a space.

---

## Author Response (AR2)

We would like to thank both reviewers for their detailed and insightful comments. These comments have helped improve and clarify the submitted manuscript. Below we reply to each comment point by point, showing the reviewers' comments in black and our responses in blue. Changes to the original manuscript are highlighted in **bold blue**. Note that the line numbers in the response are updated based on the revised manuscript, which we provide with our response.

We note already here that we reran the numerical experiments related to FI-Hyy and US-Ha1, as we detect an error related to multi-year LAI read.

**Reviewer #2**

Zhu et al. have greatly improved the clarity of their manuscript and mostly made adequate changes and responses to the reviewer comments on the first round. However, I still have two major concerns regarding the paper.

**Response:** We thank the reviewers for recognizing our efforts in the first round of reviews. After carefully considering your feedback, we fully understand your two major concerns and appreciate the opportunity to address them. Your insights are invaluable in helping us refine our work, and we have answered both questions in detail.

Continuing the discussion related to the US-Wrc site gradient fluxes. Note that there are two papers by Rastogi et al. in 2018 using gradient flux data from the same site; one has data from 2014 and COS fluxes are (for whatever reason) not published (https://doi.org/10.1029/2018JG004430) and the other reports fluxes from year 2015 (https://doi.org/10.5194/bg-15-7127-2018). Dataset for the latter (including gradient fluxes of COS) can be found from: https://zenodo.org/records/1516332. To reduce the considerable bias the authors currently have regarding the calculation of US-Wrc fluxes, I highly recommend to use the published COS gradient fluxes from this site, from year 2015, and to rerun the analysis once more using this dataset. This would considerably reduce bias and improve the analysis. These data can then also be used in the two-site assimilation, which, I still in its current state (when gradient fluxes are first calculated using simulations, which then are again used to simulate fluxes) I do not approve of. If proper gradient fluxes provided by Rastogi et al (in the link above) are used, only then the two-site assimilation is possible. Note that Hyytiälä forest also has flux measurements in 2015.

**Response:** Thank you for your investigation of the COS data at the US-Wrc site, and the advice to us in conducting assimilation experiments. COS flux measurements reported in Rastogi et al. (2018a) are from between 18 April and 31 December 2015. However, only the COS flux observations in October 2015 are available via https://zenodo.org/records/1516332, and we found that this dataset contains only 93 COS flux observations. Since our study focused on the hourly data combining the hourly-scale BEPS model to improve the model's performance at hourly scale, we conducted our current work in the growing-season and also the period when there are enough hourly data for assimilation. After serious evaluation of the datasets, we determined that the 2014 dataset is more suitable for our study and can be compared with other sites which are also focusing on the growing season. However, we highly appreciate the

reviewer for suggestions of alternative datasets that can potentially improve our study in the next step, by assimilating datasets from different periods.

Secondly, as you mentioned below: "observations would not be perfect, but they are 'the best guess' we have", the lack of necessary parameters makes the COS fluxes estimated from COS concertation observations unreliable at US-Wrc. Therefore, we employed a correction method proposed by Leung et al. (1999) to estimate COS flux by combining COS observation and simulation. The effectiveness of this method has been validated by previous studies and is widely used (Scholze et al., 2016; Wu et al., 2020). In fact, we have also provided the diurnal variation of the corrected COS fluxes in the manuscript (Figure 3m). While the diurnal variation of the corrected COS fluxes still has some differences compared to that in Figure 4 in Rastogi et al. (2018b), they are indeed very similar, which gives us more confidence in utilizing these data for conducting assimilation experiments.

Last but not least, we do recognize that there will be errors in the measurement and processing of the data. Recognizing the imperfections and inconsistencies between theory (usually in the form of a numerical model) and observations, and combining the two to achieve the optimal estimation of the target variable, is precisely the core concept of data assimilation. The variational method employed in NUCAS takes full account of uncertainties of the model and observation, and utilized the optimization methods to minimize the difference between the model and observation (Which will not be zero, due to uncertainties in both the model and the observations). In our study, we performed a two-site assimilation experiment at FI-Hyy and US-Wrc. This experimental scheme is, in fact, an exploration of the solution to the problem of the large uncertainty in single-site COS data. With the availability of more COS data in the future, we will continue to explore solutions to the problem of large uncertainties in single-site data like this.

In order to further remind readers of the current inconsistencies in COS data processing methods, we specifically emphasize the US-Wrc site in **Section 4.5 Caveats and implications**: "Besides, the existing COS flux data were calculated based on different measurement methods and data processing steps, which poses considerable challenges for comparing COS flux measurements across sites, "**Particularly, as only raw COS concentrations was provided and a correction approach was employed, the estimated COS fluxes at US-Wrc may subject to considerable uncertainties.**" (Line 632-633)

The authors argue that sensible heat flux (H) and latent heat flux (LE) as well as soil water content (SWC) are related to COS fluxes because COS fluxes are related to transpiration. However, transpiration is only one part of ET (evapotranspiration, highly related to LE) and the other part is evaporation, which has no relation to COS fluxes. Evaporation and SWC are also highly related to water availability (precipitation) as well as other environmental variables (radiation, temperature). In addition, it is definitely not only the leaf-scale energy demand that controls the sensible heat flux at ecosystem scale. You forget soil, atmospheric turbulence, input energy from the sun, ground heat flux, evaporation, precipitation, saturation of SWC... Yes, COS fluxes could be used to estimate transpiration, but anything further is overinterpretation. Thus, I still very highly recommend completely leaving out the LE, H and SWC simulations.

**Response:** Thanks for your detail comment. Carbonyl sulfide (COS) has been proven as a promising tracer for photosynthesis and transpiration due to the coupling of leaf exchange of COS, $CO_2$ and $H_2O$ through stomach, and the photosynthesis and transpiration play a significant role in leaf energy balance. The relationship between leaf energy balance (latent heat (LE), sensible heat (H)) and photosynthesis, and transpiration has been described in detail (e.g., in Leuning et al. (1995)). Taking into account the coupling of photosynthesis, stomatal conductance, transpiration, leaf energy balance, current ecosystem models (e.g., BEPS) have also realized the coupled simulation of them through the Ball-Berry model (Ball et al., 1987), Penman-Monteith equation (Penman, 1948; Monteith, 1965), etc. Therefore, we collected H, LE as well as SWC observations for investigating the optimization ability of COS for the model because they are related with transpiration. We note here that ecosystem-scale LE observations have been used to investigate the ability of COS to constrain transpiration (Abadie et al., 2023). Following Luo et al. (2018), we output each component of evapotranspiration (ET) and H separately, and conformed that changes in simulated ET and H are dominated by transpiration and canopy sensible heat.

Following reviewer #3's comment: "Generally, I would say that the results for H, LE and SWC don't bring a huge amount to the paper. And large portions could be condensed or moved to supplementary material. But it certainly **does not** detract from the paper. However, it somewhat distracts the reader…", we therefore removed this section from the manuscript and present them only in the supplement.

From the response document:

L400: "due to high value of observation" or rather underestimation by simulation?

Response: Could, of course, but either, according to Kooijmans et al. (2021), the air depleted in COS can then suddenly be captured by the EC system when turbulence is enhanced in the morning.

» This is why we do storage correction to EC fluxes! Storage corrected fluxes do **not** have this problem. I am not saying that observations would be perfect, but they are "the best guess" we have. Thus, I suggest to reformulate accordingly.

**Response:** Thank you for your comment. But, I apologize that we have removed that sentence in the first round of review.

Specific comments:

How is this manuscript related to a preprint that is simultaneously in review (Zhu et al., 2024)? The other study seems very much related, and should be cited in this study as well.

**Response:** Thanks for your comment. In another manuscript, a Monte Carlo based approach was utilized to optimize GPP using COS. This topic is indeed relevant to this manuscript and we have included a reference to it. The revised sentence reads as: "**As an important probe for characterizing stomatal conductance, COS has shown great potential to constrain plant photosynthesis and transpiration and to improve understanding of the water-carbon coupling (Wohlfahrt et al., 2012; Asaf et al., 2013; Wehr et al., 2017; Kooijmans et al.,**

**2019; Sun et al., 2022; Zhu et al., 2024).**" (Line 59-56)

The abstract and conclusions are still missing concrete results. The authors use descriptive words such as "improved"-> improved by how much or by what metric? Describe in detail (using numbers) what were the most important results of your study (e.g. how much (in %) did the assimilation improve the prior simulation etc).

**Response:** Thank you for your valuable comment. Our results suggest that the assimilating of COS fluxes can notably improves the model performance in GPP and ET, with average root mean square error (RMSE) reductions of 23.54% and 16.96%, respectively. For H, the RMSE of the simulations and observations exhibited little change before and after assimilation, while the $R^2$ increased, on average, by 0.07. In response to your comment, we have modified the abstract to provide a quantitative description of the assimilation results: "**Comparing model simulations with validation datasets, we found that assimilating COS fluxes notably improves the model performance in gross primary productivity and evapotranspiration, with average root mean square error (RMSE) reductions of 23.54% and 16.96%, respectively.**" (Line 27-29)

Merge Figs 3 and 4 in a similar way as Fig. 6.

**Response:** Thanks for your comment. We have merged this two Figures as you suggested.

Eq. 9-10: Fcos,biotic is switched to FSWCg and SWC to SWCg between the equations. Please check that is consistent.

**Response:** Thank you for your comment. We have checked the manuscript and confirmed that there are no problems with the model description. According to Whelan et al. (2016) and Whelan et al. (2022), the model assumes that COS soil biotic uptake changes with SWC, and there exists an optimum SWC ($SWC_{opt}$) at which the simulated biotic COS flux is maximized, i.e. optimum in terms of COS soil biotic uptake ($F_{opt}$). Base on $SWC_{opt}$, $F_{opt}$, and the COS flux ($F_g$) under another soil moisture condition ($SWC_g$, and $SWC_g > SWC_{opt}$), it is thus possible to determine the unique shape parameter $a$ for which the model varies with SWC and, in turn, to calculate soil biotic COS fluxes ($F_{COS,biotic}$) for any SWC condition.

$$a = ln\left(\frac{F_{opt}}{F_{SWC_g}}\right) * \left(ln\left(\frac{SWC_{opt}}{SWC_g}\right) + \left(\frac{SWC_g}{SWC_{opt}} - 1\right)\right)^{-1} \tag{1}$$

$$F_{COS,biotic} = F_{opt}\left(\frac{SWC}{SWC_{opt}}\right) * e^{-a\left(\frac{SWC}{SWC_{opt}}-1\right)} \tag{2}$$

**Reviewer #3**

Zhu et al. present a novel study that utilises the Nanjing University Carbon Assimilation System (NUCAS) v1.0 data assimilation framework and the process-based terrestrial ecosystem model Boreal Ecosystem Productivity Simulator (BEPS) is used as the adjoint model. The authors focus on simulating carbonyl sulfide (COS) fluxes at an ecosystem scale. Additionally, prior and posterior estimates of gross primary productivity (GPP), soil water content (SWC), sensible heat (H) and latent heat (LE) are presented. Assimilating COS flux measurements is a novel approach to better our understanding of COS processes, with regards to the carbon and water cycles, and the energy budget. This research is certainly within the scope of GMD and should be considered for publication following some minor corrections.

Having read the author's response to referee comments, Zhu et al. have clearly made a significant effort to improve their work. This is evidenced in the improved narrative of the paper and clarification on some of the methodology and data inputs. However, there is room for a bit more improvement in the readability, which my recommendations are focused on.

Response: Thank you for recognizing our efforts in the first round of reviews. We sincerely appreciate your detailed comments on improving the readability of the manuscript and the opportunity to address them. Your insights are invaluable in helping us refine our work, and we have answered your comments in detail, as shown below.

**General Comments:**

The abstract would benefit from having at least one specific quantifiable metric that shows the improvements in COS flux or GPP made by NUCAS.

Response: Thank you for your valuable comment. We have changed the relevant sentence in the abstract to provide a quantitative description: "**Comparing model simulations with validation datasets, we found that assimilating COS fluxes notably improves the model performance in gross primary productivity and evapotranspiration, with average root mean square error (RMSE) reductions of 23.54% and 16.96%, respectively.**" (Line 27-29)

There is use phrases like 'various constants' and other ambiguous phrases. I understand that elaborating in every instance could lead to excessive description of basic processes or repetition. But there is flexibility for some expansion on the use of these phrases. They often offer context and provide insight to readers who may not be familiar with this field, especially that of carbonyl sulfide which is rapidly becoming a field of its own within the world of carbon cycle science. An example on Lines 129-130: "The model parameters are the various constants that are not influenced by the model state."

Response: Thank you for your comment. We have revisited this and elaborated in more detail.

Sections 1 and 2 provide a thorough introduction to the topic and the methodology used in this work. However, the description of variables used in the modelling and assimilation is a bit light in places, leaving a lot of work for the reader to piece together. This harms to reproducibility of this work and makes it relatively inaccessible to a reader who is not familiar with matrix inversion, Bayesian statistics or data assimilation.

○ A sentence distinguishing biotic and abiotic soil processes would be helpful in Section 2.1.3.

Response: Thank you for your comment. In response to your comments, we have provided additional clarification on soil biotic and abiotic processes as detailed below:

"$F_{COS,abiotic}$ **is controlled by abiotic degradation of soil organic matter (Whelan and Rhew, 2015), can be described as an exponential function of the temperature of soil** $T_{soil}$ **(°C).**" (Line 182-183)

"$F_{COS,biotic}$ **is attributed to CA in microbial communities (Sauze et al., 2017), calculated according to Behrendt et al. (2014) and Whelan et al. (2016):** " (Line 186-187)

○ Elaborate a bit on the parameters being referred to in Section 2.2.

Response: Thanks for your comment. In response to your comment, we provide a more detailed description of the model parameters in Section 2.2, especially the five parameters optimized in this study. as shown below: "**NUCAS v1.0 can optimize 76 parameters belonging to BEPS. Of these parameters, some are global (i.e., the ratio of photosynthetically active radiation to shortwave radiation (f_leaf)), and others differentiated by PFT (i.e., maximum carboxylation rate of Rubisco at 25°C ($V_{cmax25}$)) or soil texture class (i.e., $Ksat_{scalar}$, the scaling factor of saturated hydraulic conductivity (Ksat)).**" (Line 200-203)

○ Expand on the addition made in this iteration in Section 2.4.3, regarding coupling of COS with LE, H and SWC. It's still a bit light. At the very least include some additional references. Alternatively, this could be elaborated on in the Discussion (Section 4), to which you could point the reader to from Section 2.4.3.

  ○ After giving this point some further thought, perhaps elaboration would be best suited to when the main scientific questions are presented in Section 1. I will leave this up to the discrepancy of the author(s).

Thank you for your valuable comments. We have changed the sentence as you suggested, i.e., while adding more references, we have clarified to the reader that a more detailed elaboration will be provided in the Discussion (Section 4). The revised sentences read as: "**Due to the coupling between leaf exchange of COS, CO₂ and H₂O, GPP, and ET data are selected to evaluate the model performance of COS assimilation in this study. In addition, we further explored the ability of COS to constrain H simulations, since the transpiration contribute to a decrease in temperature within the leaf (Gates, 1968; Konarska et al., 2016), and the leaf-air temperature gradient is a key control factor of H (Monteith and Unsworth, 2013; Dong et al., 2017). Moreover, SWC is used in model evaluation as the key role of SWC in modelling** $F_{COS,biotic}$ **(as shown in Eq. (9)) and that the water dissipated in transpiration originates from soil (Berry et al., 2006). A more detailed elaboration will be provided in the discussion.**" (Line 280-285)

– Section 3 is a thorough summary of the results. However, there are 2 key points that could

be improved, firstly some more summary of the implication of what the parameters mean, i.e. what does it mean if the relative change in VJ_slope is large for example. Does it mean the assimilation has changed the posterior results significantly with regards to the prior and if so, what does that actually tell us? The summary at the end of 3.6 is a good example of where this has been done well. A few more sentences like that would be good! Secondly and a much more minor point, there could be more specific reference to which figure and sub-figure is being referred to. It doesn't need to be excessive but try and help the reader from having to constantly check tables to cross-reference the PFT and soil texture between sites when this is referenced, for example. A good example of this occurring is at the start of Section 3.7, where you introduce the topic of this section and then just point the reader to Figure S4-S7. Provide a bit more guidance and refer to them as they are discussed in turn – I have included some specifics in the minor comments.

Response: Thanks for your kindly comment. In response to your another comment, "Elaborate a bit on the parameters being referred to in Section 2.2.", we have we provide a more detailed description of the model parameters in Section 2.2. For VJ_slope, the slope of the $V_{cmax}$ and $J_{max}$ (maximum electron transport rate) relationship, a low value of it indicates that photosynthesis at this site is more likely to be limited by the rate of the light reaction.

Thank you for your comment regarding to providing more specific reference to figure and sub-figure, we have made improvements in this regard to enhance the readability of the manuscript.

‒ Generally, I would say that the results for H, LE and SWC don't bring a huge amount to the paper. And large portions could be condensed or moved to supplementary material. But it certainly **does not** detract from the paper. However, it somewhat distracts the reader from what I would consider very good results regarding posterior COS fluxes and GPP. This is meant as an opinion and how to deal with this can be left to the discretion of the authors.

Response: Thank you for your valuable comments. We agree with your statement that these variables are not as well optimized as GPP, and that the presentation of these results distracts the readers. We have therefore removed this section.

‒ There seems to be interchangeable use of "COS flux data", "COS flux measurements", "COS data" and "COS measurements". There is a big difference between a measurement of COS fluxes and ambient COS concentration for example. A good example is the first sentence of section 2.5.2: "After the ability of NUCAS to assimilate COS flux data was confirmed by twin experiments, we could then use the system was then utilised to conduct data assimilation experiments with real COS observations under single-site and multi-site conditions" – Is "real COS observations" referring to measurements of the flux? Please be diligent in distinguishing this throughout the paper. From what I can tell, the only use of ambient COS concentrations is to drive the estimation of prior COS fluxes.

Response: Thank you for your detailed comments. We acknowledge that the there is a big difference between a measurement of COS fluxes and ambient COS concentration.

However, since these flux data are actually derived from concentrations, we have difficulty in describing them specifically in some places. However, we did make changes in this area to provide a more consistent description, such as changing "COS flux data" to "COS fluxes".

"real COS observations" does refer to measurements of the flux. The reason for this designation is to distinguish it from the "pseudo-observations" of the twin experiment.

As you said, the only use of ambient COS concentrations is to drive the estimation of COS plant uptake.

Some sentences are unclear. I have highlighted potential improvements in the readability of these in the Minor Comments. Please read them careful to ensure there has not been a misinterpretation.

Response: Thank you for your detailed comments. Your revisions have greatly improved the readability of this manuscript, for which we are deeply grateful. However, for a very few cases, we have not made the changes exactly as you suggested. We have explained this point by point, please see below for more details.

**Minor Comments:**

- L22-24: improve readability. Example: "Data assimilation experiments were conducted to investigate the robustness of NUCAS, and to test the feasibility and applicability of assimilating carbonyl sulfide (COS) fluxes from seven surface sites, in order to better our understanding of stomatal conductance and photosynthesis."

Response. Thank you for your valuable comment. We have carefully reviewed the manuscript and made revisions to improve its readability.

- L26: I assume you mean COS fluxes: "assimilation of COS fluxes can".

Response: Thanks for your comment. As you mentioned, we assimilated COS fluxes in this study, therefore we have modified it accordingly.

- L28: Be consistent in referring to NUCAS as "the NUCAS" or simply "NUCAS". Also see note for Line 73.

Response: Thank you for your comment. Now, we have referred to NUCAS consistently as "NUCAS".

- L28: "to the" -> "with"

Response: Thanks for your comment. We have modified the sentence accordingly.

- L36: "of earth system" -> "of the Earth system"

Response: Corrected.

- L37: the biosphere 'significantly mitigating climate change' is a bit speculative in my opinion. It has certainly reduced the full potential of climatic changes since 1850. But the reasoning is

partly through feedbacks, such as longer growing seasons in the Northern Hemisphere, as a result of climate change. I would just end the sentence at 1850 (and of course keep the reference).

Response: Thank you for your valuable comment. We have modified the sentence accordingly.

- L38-39: "of terrestrial biosphere have changed" -> "of the terrestrial biosphere has changed".

Response: Corrected.

- L41: "important tool to investigate" -> "important tool used to investigate".

Response: Corrected.

- L44: "data" -> "datasets"

Response: Corrected.

- L45: The reference the Scholze et al. (2017) feels a bit random. Be more specific about the 'various observations' you're referring to at the beginning of the sentence or just remove the reference. I would just start the sentence: "Observations such as sun-induced.."

Response: Thanks for your detailed comment. This is because Scholze et al. (2017) is a review article on datasets used in terrestrial carbon cycle data assimilation. We believe it would be better to keep this article because it would facilitate readers to learn more about the earth observations used in terrestrial carbon cycle data assimilation. According to your suggestion, we have modified the sentence: "**Observations such as sun-induced chlorophyll fluorescence (Schimel et al., 2015) and soil moisture (Wu et al., 2018), have been used to estimate or constrain carbon fluxes in terrestrial ecosystems (Scholze et al., 2017).**" (Line 47-49)

- L47: Not sure this is a 'recent' finding any more. Remove.

Response: Thanks for your comment. We have removed "Recently" accordingly.

- L52-54: Move these two sentences (starting 'Plants' and 'As' to the paragraph above), as they are still discussing the relationship between CO2 and COS. The sentence afterwards moves into discussing COS and GPP modelling.

Response: Thanks for your comment. In fact, as you say, these two sentences are still discussing COS and $CO_2$. but our idea is to have a second paragraph that gives a preliminary introduction to terrestrial ecosystem model and the datasets used to constrain the model, and lead into COS. In the third paragraph, COS is then described specifically, such as the role of COS in indicating not only GPP but also stomatal conductance and transpiration, the development of COS modeling, COS observations, etc.

- L58: What other 'key ecosystem variables' are you referring to? Are there any others that COS can estimate better than direct measurement? The reason it's useful to estimate GPP is because it is impossible to measure GPP directly at large spatial scales. Perhaps be specific about what you are referring to, the reader might be interested.

Response: Thanks for your comment. As you mentioned, GPP cannot be directly measured duo to the hindrance of respiration. Similarly, the measurements of transpiration are confounded by

evaporation. Due to the coupling of leaf exchange of COS, $CO_2$ and water vapor through stomach, COS exchange measurement can provide independent and direct way for estimating transpiration and stomatal conductance. Currently, some studies have combined COS models with COS observations to estimate not only GPP but also stomatal conductance and transpiration (Wohlfahrt et al., 2012; Wehr et al., 2017; Sun et al., 2022). Therefore, we have modified the sentences to be more specific: **"A number of empirical or mechanistic COS plant uptake models (Campbell et al., 2008; Wohlfahrt et al., 2012; Berry et al., 2013) and soil exchange models (Kesselmeier et al., 1999; Berry et al., 2013; Launois et al., 2015; Sun et al., 2015; Whelan et al., 2016; Ogée et al., 2016; Whelan et al., 2022) have been developed to simulate COS fluxes in order to more accurately estimate gross primary productivity (GPP), stomatal conductance as well as transpiration."** (Line 56-60)

- L58: "However, with the lack of" -> "However, due to the lack of".

Response: Corrected.

- L58-61: Move the references to Wohlfahrt et al. (2012) and Kooijmans et al. (2021) to the end of the sentence. Remove Bruhl et al. (2012), I don't understand the context of that reference here.

Response: Thanks for your comment. We have relocated the references of Wohlfahrt et al. (2012) and Kooijmans et al. (2021) accordingly. As done in Wohlfahrt et al. (2012), Brühl et al. (2012) was cited here for arguing that there is a lack of ecosystem-scale field COS measurements.

- L62: "behavior" -> "behaviour".

Response: Corrected.

- L65: remove "various".

Response: Corrected.

- L68-71: I think it's clear you're referring to data assimilation techniques. This sentence could be reduced and read better, example: "More specifically, the observed dynamics of ecosystems can be more accurately portrayed, additionally, our understanding of ecosystem processes can be deepened, with respect to their responses to climatic changes."

Response: Thanks for your valuable comment. We think it is important to mention "process-based model" here. Thus, we have revised the sentences as: **"More specifically, by applying data assimilation methods to process-based models, not only can the observed dynamics of ecosystems be more accurately portrayed, but also our understanding of ecosystem processes can be deepened, with respect to their responses to climate changes (Luo et al., 2011; Keenan et al., 2012; Niu et al., 2014)."** (Line 70-73)

- L73: Include a note that NUCAS v1.0 will be referred to as NUCAS for the remainder of the paper: "NUCAS v1.0, hereafter referred to as NUCAS, is designed"

Response: Thank you for your valuable comment. We have modified the sentence accordingly.

- L73-74: Improved punctuation. "to assimilate multiple observational data streams including COS flux data to improve the process based" -> "to assimilate multiple observational data

streams, including COS flux data, to improve the process-based".

Response: Thank you for your detailed comment. We have modified the sentence accordingly, and revisited the manuscript to improve punctuation.

- L77-78: An example of non-line-breaking hyphen. This will be the only in-text mention I refer to. **Please see Technical Note on this.**

Response: Response: Thank you for your detailed comment and the useful notes. We have thoroughly examined the use of non-line-breaking hyphen and made modifications.

- L82: Clarify "COS fluxes"

Response: Thanks for your comment. We have revised the sentence accordingly.

- L85: Again, are you referring to COS observations or COS flux observations?

Response: Thank you. Here we want to refer to COS flux observations, for which we have modified accordingly.

- L88: Combine and clean these sentences: "Materials and methods used in our study are described in Sect. 2, such as the BEPS model and NUCAS, are introduced".

Response: Thanks for your valuable comment. We have revised the sentence accordingly.

- L89: What data do you mean? Just be a bit more specific: "along with the data used to drive BEPS and assimilated into NUCAS."

Response: Thanks for your valuable comment. We have revised the sentence accordingly.

- L104: "if" -> "is" and "First" -> "first".

Response: Thanks for your detail comment. We've corrected the typos.

- L119: Remove BFGS acronym definition. It isn't used again.

Response: Thanks for your detail comment. We have removed it accordingly.

- L128-132: Keep this paragraph with the previous one. The discussion is still relevant, i.e. regarding cost function.

Response: Thank you. We have made the changes you suggested.

- L140: Change "new" to "updated". I wouldn't refer to literature written in 1999 as new.

Response: Corrected.

- L153-155: Be more specific about what is being referenced here: "The canopy-level COS plant uptake $F_{cos,plant}$ (pmol m−2s−1) was calculated by upscaling the resistance analog model of COS uptake, as presented by Berry et al. (2013), with the upscaling scheme recommended by Chen et al. (1999)."

Response: Thanks for your valuable comment. We have revised the sentence accordingly.

- L161: No need to end the sentence. Also include another reference to Berry et al. (2013), as

you are directly quoting the calculation: "where $COSa$ is the COS mole fraction in the bulk air and $gsw$ and $gbw$ are the stomatal conductance and leaf laminar boundary layer conductance to water vapor (H2O), respectively (Berry et al., 2013)."

Response: Thank you for your valuable comment. We have revised the sentence accordingly.

- L164-165: Try to avoid starting a sentence with a lower case, even if it is a relevant parameter. Perhaps: "The apparent conductance for COS uptake from the intercellular airspaces is denoted by gcos and combines the mesophyll conductance and the biochemical reaction rate of COS and carbonic anhydrase (CA)."

Response: Thanks for your detailed comment. We have revised the sentence accordingly.

- L166-169: Move the references to the end and lead the sentence into equation 4. Perhaps: "rate of Rubisco at 25℃ (Badger and Price, 1994; Evans et al., 1994), such that: Eq. (4) where α is a scaling parameter that is calibrated"

Response: Corrected.

- L170: These values of α need a reference. If it's from Stimler et al. (2012), then move the reference in the previous sentence to here.

Response: Thank you. We have added a reference of the values of α (Haynes et al., 2020).

- L170-171: "With reference to the COS modelling scheme: Simple" -> "According to the COS modelling scheme: Simple"

Response: Corrected.

- L178: End this sentence with : rather than . As you are referring to Eq. 7 anyway.

Response: Thanks for your detailed comment. We have modified the punctuation accordingly.

- L180: Same as above. : rather than .

Response: Corrected.

- L182: Now start sentence with where, instead of Where.

Response: Corrected.

- L187: "Here" -> "In Eq. 10," Also it is not clear what a is used for. Maybe add some context?

Response: Thanks for your comment. According to Whelan et al. (2016) and Whelan et al. (2022), the model assumes that COS soil biotic uptake changes with SWC, and there exists an optimum SWC ($SWC_{opt}$) at which the simulated biotic COS flux is maximized, i.e. optimum in terms of COS soil biotic uptake ($F_{opt}$). Base on $SWC_{opt}$, $F_{opt}$, and COS flux ($F_g$) under another soil moisture condition ($SWC_g$, and $SWC_g > SWC_{opt}$), it is thus possible to determine the unique shape parameter $a$ for which the model varies with SWC and, in turn, to calculate soil biotic COS fluxes ($F_{COS,biotic}$) for any SWC condition. In summary, parameter a is a curve shape constant of the empirical model, which can be determined by SWC and the corresponding COS abiotic flux observations.

- L187: "maximum" -> "optimal".

Response: Corrected.

- L190: "reference of" -> "reference to"

Response: Corrected.

- L191: This is the first use of SWC as an abbreviation, so define it, i.e. soil water content (SWC).

Response: Thank for your comment. We have revised the sentence to include a definition of SWC accordingly.

- L195: use the variable names, "COS plant uptake" -> "Fcos,plant" and COS soil fluxes -> "Fcos,soil". You can probably just remove this sentence to be honest. But up to the author.

Response: Thanks for your comment. This sentence is not necessary and we have deleted it.

- L198: "tuned previous model in development" -> "tuned in past model development"

Response: Corrected.

- L200: Specifically refer to the research: "The prior uncertainty of parameters is set based on previous studies by Chen et al. (2022) and Ryu et al. (2018)."

Response: Thank you for your valuable comment. Due to our current lack of understanding of model parameters, and the fact that not all parameter uncertainties have available references (or perhaps we have not found them), we have made the following modifications to the sentence: "**The prior uncertainty of parameters is set based on previous research, i.e., Ryu et al. (2018) and Chen et al. (2022).**"(Line 204-205) In addition, we have also explained this in section 4.5: "**Furthermore, there is a lack of understanding of the prior uncertainty for certain model parameters, such as VJ_slope, which makes the uncertainty estimates subject to potentially large errors.**" (Line 640-641)

- L222: "was" -> "were"

Response: Corrected.

- L223-224: Sentence about LAI, meteorology and soil datasets need a reference. Or as they were mentioned in 2.1.2, perhaps direct the reader there.

Response: Thanks for your comment. We have now added the necessary references here, i.e., Liu et al. (1997) and Chen et al. (1999).

- L226: "soil water content (SWC) at these sites collected at the sites were used" -> "SWC collected at the sites were used".

Response: Corrected.

- L231: "LAI product represents Lead area index at a" -> "LAI product quantifies leaf area index".

Response: Corrected.

- L231-233: spatial resolution is traditionally presented as 8 × 8 km for example. As it is a 2-D shape. Please clarify.

Response: Corrected.

- L235: You may as well be specific in the products you are referring to. "The other two LAI products were used to investigate the effect of the LAI products" -> "The GLASS and MODIS LAI products were used to investigate the effect of different LAI products"

Response: Thanks for your detailed comment. We have revised the sentence accordingly.

- L238-239: This last sentence is a bit wordy, but I understand what is to be communicated. Perhaps: "In addition, the 8-day temporal resolution of the LAI data was interpolated into daily values using the nearest neighbour method."

Response: Thanks for your understanding and suggestion. We have revised the sentence accordingly.

- L241-245: These few sentences could be a bit neater and consistent. Perhaps: "Standard hourly meteorological data was inputted in BEPS, including air temperature at 2 m, shortwave radiation, precipitation, relative humidity and wind speed, taken from the FLUXNET database (for sites: AT-Neu, DK-Sor, ES-Lma, FI-Hyy and US-Ha1 see https://fluxnet.org), the AmeriFlux database (for sites: US-Ha1, US-Wrc, see https://ameriflux.lbl.gov) and the ERA5 dataset (for sites: AT-Neu, IT-Soy, US-Ha1 see https://cds.climate.copernicus.eu/cdsapp#!/dataset/reanalysis-era5singlelevels?tab=overview)."

- L247: Move the reference to just before the comma. Also remove "Particularly" at the start of the next sentence.

Response: Corrected.

- L257-258: I think the eddy-covariance and gradient-based approach both require a reference.

Response: Thanks for your comment. We have now added the necessary references here, i.e., Baldocchi (2003), Kohonen et al. (2020) and Wu et al. (2015).

- 258-259: "The COS soil measurements were collected using soil chamber, except at US-Ha1, where a sub-canopy flux-gradient approach was used to calculate the soil COS flux." -> "The COS soil flux measurements were collected using soil chambers, except at US-Ha1, where a sub-canopy flux-gradient approach was used."

Response: Thanks for your comment. We have revised the sentence accordingly.

- L260: "COS flux measurements"

Response: Corrected.

- L262-264: The first sentence of this paragraph needs to be reworded. There is very little context, and the ordering is confusing. Also, what is the aerodynamic gradient method? IS this the same as the gradient-based approach? Perhaps: "US-Wrc utilises the gradient-based approach to measure COS ecosystem flux (reference), however available data is limited to only COS concentration measurements and lacking other parameters required, therefore this site

risks introducing biases." Or similar.

Response: Thanks for your comment. Following the description of COS flux measurement in Rastogi et al. (2018), "the aerodynamic gradient method" were utilized here. However, we do agree with you that such a description may confuse readers. Therefore, we revised the sentence as you suggested: "**US-Wrc utilises the gradient-based approach to measure COS ecosystem flux (Rastogi et al., 2018), however available data is limited to only COS concentration measurements and lacking other parameters required, therefore this site risks introducing biases**" (Line 267-269)

- L268-273: summary of Equation 11 could be much more concise. Perhaps: "This was done using the mean, $\overline{M}$, and standard deviation, σm, of the simulated COS flux to correct the COS flux observations ($O$):

Eq. 11"

where $\overline{O}$ and $\sigma O$ are mean and standard deviation of the observed COS flux series. F is the corrected observed COS flux and the COS simulations were calculated using the prior parameters for the time period corresponding to the COS flux observations.

Response: Thanks for your comment. We have revised the sentences accordingly.

- L273: No need to start new paragraph. Move up to previous.

Response: Thank you for your comment. We have relocated the sentences accordingly.

- L273: "as an estimate"

Response: Corrected.

- L276: "LE data are selected" -> "LE, data were selected"

Response: Corrected.

- L277: This needs a reference.

Response: Thanks for your comment. Now, we have added the necessary references to this sentence, i.e., Wohlfahrt et al. (2012) and Whelan et al. (2018).

- L283: Move this reference to the end of the sentence.

Response: Corrected.

- L306-307: "With reference" -> "Regarding"

Response: Corrected.

- L309: "After the ability of NUCAS to assimilate COS flux data was confirmed by twin experiments, we could then use the system" -> "After the ability of NUCAS to assimilate COS flux data was confirmed by twin experiments, the system was then utilised"

Response: Corrected.

- L312:

Response: Corrected.

- L317-318: put comma (,) after experiment and simultaneously.

Response: Done.

- L325: "in twin experiment" -> "in a twin experiment".

Response: Corrected.

- L326: was calculated "using Eq. 12".

Response: Corrected.

- L328: "where" -> "Where".

Response: Thanks for your comment. As you mentioned before, it would be more appropriate to use "where" as the start of a sentence. Therefore, we reserve the use of "where" here.

- L352: I only see Dfinal in Table S5.

Response: Thanks for your comment. Due to an identical perturbation size (0.2) being applied in all twin experiments, the initial value ($D_{initial}$) of $D_x$ is constant, with a value of 7.48. Thus, we describe $D_{initial}$ in the title of the table.

- L355: "nearly zero with the maximum value below" -> "nearly zero, where the maximum value was below".

Response: Corrected.

- L356: pseudo-observations

Response: Thanks for your detailed comment. We have corrected this typo.

- L358:

Response: Corrected.

- L364:  I read this as 6.35% in Table 2?

Response: Thanks for your detailed comment. We have corrected this error.

- L365: "with the cost function reduction of 16.39% and 15.70%." -> " of 16.39% and 15.70% respectively."

Response: Corrected.

- L366: Include the percentage values for these 2 sites: "FI-Hyy (21.47%)" and "US-Wrc (27.71%)".

Response: Thanks for your detailed comment. We have revised the sentences accordingly.

- L367-368: I believe you mean July of 2015. This sentence needs to be reworded, perhaps: "In August 2014 and July 2015, the cost function reduction was between 40.59 % and 50.94 %, while in July of all other years, the cost function reduction was much lower, ranging from 4.87% to 18.94 %."

Response: Thanks for your comment. Actually, we want to refer to July 2014 and August 2014 here. According to your suggestion, we have revised the sentence: "**In July 2014 and August 2014, the cost function reductions were 20.17% and 38.86% respectively, while in July of all other years, the cost function reductions were are much lower, ranging from 2.84 % to 5.88 %.**" (Line 379-381)

- L371-372: This is a bit of a throwaway sentence. If prior simulations were that good, we wouldn't need an inversion scheme right. I don't think sentence is necessary and almost devalues the posterior results.

Response: Thanks for your valuable comment. We have removed the sentence as you suggested.

- L377: I found this to be a particularly interesting finding. In that the way the COS fluxes are being calculated in the posterior are clearly missing one or more processes to exactly replicate measurements. Certainly something to investigate in future research and perhaps highlight in your conclusions.

Response: Thanks for your valuable comment. We believe there are two main reasons for the ineffective simulation of nighttime COS fluxes, and we have already covered this in the discussion: "**On the one hand, this is due to the substantial gap between current modelled COS soil fluxes and observations (Whelan et al., 2022). On the other hand, this also stems from the fact that the nighttime stomatal conductance was set to a low and constant value (1 mmol m$^{-2}$ s$^{-1}$) in the BEPS model**" (Line 516-518). Of course, we agree with your point that this should be investigated in future research.

- L382: You can remove the sentence starting "Similar to". I believe you raised this in the previous sentence.

Response: Thanks for your valuable comment. We have re-reviewed the article and corrected this.

- L384: 6.94  to 3.09 pmol m$^{-2}$ s$^{-1}$

Response: Corrected.

- L387: missed at: nighttime.

Response: Thank you for your detailed comment. We have corrected this typo.

- L390: "FI-Hyy and US-Wrc have different soil textures, with sandy loam and loam, respectively." -> "FI-Hyy and US-Wrc have different soil textures; sandy loam and loam respectively."

Response: Corrected.

- L391: "took this difference into account and" -> "accounted for this difference appropriately and".

Response: Corrected.

- L392: I calculate this to be 26.28% but perhaps I have mixed up numbers. Please check.

Response: Thanks for your detailed comment. We have mixed up numbers, and we have corrected this error.

- L408: If the two-site assimilation method achieved similar results to the single-site, why do we need the two-site? Was is it more of proof-of-concept? A sentence summarising why it was useful would be helpful.

Response: Thanks for your valuable comment. A detailed description of the purpose of conducting the two-site experiment has been provided in the conclusions, i.e., "Our two-site setup constitutes a challenge for the assimilation system, the model and the observations. In this setup, the assimilation system has to determine a parameter set that achieves a fit to the observations at both sites, and NUCAS passes this important test. It should be noted that NUCAS was designed as a platform that integrates multiple data streams to provide a consistent map of the terrestrial carbon cycle although only ecosystem COS flux data were used to evaluate the performance of NUCAS in this study. The "two-site" assimilation experiment conducted in this study gives us more confidence that the calibrated model will provide a reasonable parameter set and posterior simulation throughout the plant functional type. In other words, what we present here is a pre-requisite for applying the model and assimilation system at regional to global scales."

- L410: regarding 'as mentioned before', it looks like this material has been moved to the appendix. Please check and update accordingly.

Response: Thanks for your detailed comment. We have checked and updated this.

- Section 3.4: It isn't really clear if a positive or negative change is a good thing. Especially as the majority of the summary refers to absolute differences. Could this be elaborated on?

Response: Thank you for your suggestions. In this section, we mainly want to characterize the actual results of parameter changes. Whether the results of parameter tuning are good or bad (as I understand it in terms of improving the model simulation) needs to be judged subsequently by performing an evaluation of the simulation results. Since the parameter changes are relative to the prior values, absolute averages are used here to better reflect the magnitude of fluctuations in the parameter changes.

- Line 419: 45.09% surely this value is hugely skewed by ES-Lma? What is the value excluding this site?

Response: Thank you for your comment. With the exception of ES-Lma, the mean absolute change of $V_{cmax25}$ is 34.94%.

- L425-427: my interpretation of IT-Soy is that on paper the RMSE is ok and improved in the posterior due to improvements during the daytime. However, it's minimal change from the prior suggests it is not particularly sensitive to assimilation of COS flux data.

Response: Thank you for your comment. Since the prior simulations and observations at IT-Soy are already very close, the assimilation effect of the COS flux at this site is limited and the cost function reduction is small.

- L430: remove capital T.

Response: Corrected.

- L432: include at the end of this sentence "(note the difference in x-axis scales)". By eye it is misleading initially.

Response: Thank you for your valuable comment. We have revised the sentence accordingly.

- L435-438: These values are very different. Is it appropriate to be comparing variables like-forlike in this way? A bit more explanation of the implications of the results would be helpful.

Response: Thank you for your valuable comment. Here we have calculated and compared the sensitivity indexes of different parameters using concomitant-based sensitivity analysis. The sensitivity indexes of the different parameters varied significantly, indicating that the impacts of these parameters on the ecosystem COS flux simulation and the cost function were significantly different. A more detailed explanation of the implications are presented in Section 4.2.

- L442: Unless you have specifically excluded DK-Sor, I would remove this.

Response: Thank you for your valuable comment. This is because the sensitivity of VJ_slope at DK-Sor was only 12.05%, which is notably smaller than at the other sites. We have revised the sentence in accordance with your comments: "**With the SIs ranging from 12.05% to 45.71% and 0.94% to 14.43%, VJ_slope and f_leaf also play important roles in the modelling of COS.**" (Line 453-454)

- L444-445: [at end of sentence] respectively.

Response: Corrected.

- L447: Maybe this is clearer: "Our results also suggest that f_leaf tends to play a more important role in the COS assimilation at the forest sites (DK-Sor, FI-Hyy, US-Ha1 and US-Wrc) compared to the low-stature vegetation type sites (AT-Neu, ES-Lma and IT-Soy), with the mean absolute SIs about two times than that of the latter, with the exception of DK-Sor." Optional.

Response: Thank you for your valuable comment. We have revised the sentence as follows: "**Our results also suggest that f_leaf tends to play a more important role in the COS assimilation at the forest sites (except DK-Sor, including FI-Hyy, US-Ha1 and US-Wrc) compared to the low-stature vegetation type sites (AT-Neu, ES-Lma and IT-Soy), with the mean absolute SIs about two times than that of the latter.**" (Line 459-461)

- L445: Does a lower R2 value not suggest that the assimilation has worsened the result? Also do you mean Figure S3?

Response: Thank you for your valuable comment. In terms of $R^2$, the $R^2$ between the posterior simulated GPP and the observed GPP is indeed a bit lower than that of the prior. However, we believe this is acceptable because the cost function itself considers the difference between the model and the observation, not the correlation. And, we do mean Figure S3 here, as we labeled the scatterplot of GPP with $R^2$.

- Section 3.6: Lots of plots being referred to. Include 'see Figure 6c' etc.. where necessary. Help

the reader.

Response: Thank you for your kindly comment. Now, we have given more detailed instructions on this.

- L465: I calculate the 3.81% to be 8.61%. Please check.

Response: Thank you for your comment. We have checked and confirmed the results are correct. For ES-Lma, the reason that you calculated a result of RMSE reduction of 8.61% and we present a result of 8.60%, is because we only present the RMSE with two decimals reserved.

- L471: GPP? Not COS.

Response: Thank you for your comment. The changes in the GPP simulation are caused by the assimilation of COS. Therefore, we present here the posterior simulation results for both COS and GPP. In order to improve the readability of the manuscript, we have added clarifications to the sentence: "In parallel, the model-observation difference **of GPP** also reduced, by 12.36% and 28.10%, respectively." (Line 483-484)

- L472: Drop line after 'underestimated.'

Response: Corrected.

- L484: struggling to get where these 2 percentages have come from.

Response: Thank you for your comment. These two percentages refer to the difference between the RMSE reductions of GPP in the two-site experiment and the RMSE reductions of GPP in the single-site experiments.

- L495: Please refer to figures S4-S7 as they are discussed. Rather than just listing them at the start of a section. Also as a note, if you are having to discuss and refer to figures in supplementary material, it's probably a sign that you are trying to present too much. As mentioned earlier, you could probably remove sections 3.7 and 3.8.

Response: Thank you for your valuable comment. We have removed the section in accordance with your comment.

- L551: Requires more references, bottom-up or top-down. Kooijmans et al. (2021), Ma et al.(2021), Maignon et al. (2021) and Remaud et al. (2022). For example.

Response: Thank you for your comment and the references. We have added more references to the sentence, including those you have listed. The revised sentence reads as: "**This is because COS plant fluxes are much larger than COS fluxes of soil in general (Whelan et al., 2016; Whelan et al., 2018; Spielmann et al., 2019; Kooijmans et al., 2021; Ma et al., 2021; Maignan et al., 2021; Remaud et al., 2022) and the soil hydrology-related parameters cannot directly influence the COS plant uptake.**" (Line 508-511)

- L567-569: "COS plant uptake is governed by the hydrolysis reaction of COS (Wohlfahrt et al.,2012), catalysed by CA, though it can also be degraded by other photosynthetic enzymes, e.g., RuBisCo (Lorimer and Pierce, 1989), and the reaction is not dependent on light (Stimler et al., 2011; Whelan et al., 2018)." I think reads a bit better. Optional.

Response: Thanks for your comment. We have revised the sentences accordingly.

- L586: Proven.

Response: Corrected.

- L593: CA, not carbonic anhydrase.

Response: Corrected.

- L595: "capable to influence" -> "capable of influencing"

Response: Corrected.

- L599: "sensitivity of Vcmax25" -> "sensitivity in Vcmax25"

Response: Corrected.

- L608-611: It's not clear if you're saying your work also found this. Please clarify and amend accordingly.

Response: Thank you for your valuable comment. In the sensitivity analyses conducted in this study, we examined only the sensitivity of the cost function to the model parameters and did not examine the sensitivity of photosynthetic capacity-related variables (e.g., GPP) to the model parameters. So, we can't get to that conclusion. We thus revised the sentence to avoid misunderstanding, the revised sentence reads as: "**Similar findings by Sun et al. (2019) found that the simulated GPP was more sensitive to radiation at forested vegetation types and less sensitive at low-stature vegetation types.**" (Line 566-568)

- L623-624: "In comparison, the RMSEs of GPP simulations were reduced by an average of 25.37% within the assimilation of COS, while that of LE were reduced by 16.27 %." -> "In comparison, the RMSEs of GPP simulations were reduced by an average of 25.37 % as a result of assimilating COS, but reducing LE by only 16.27 %."

Response: Thank you for your valuable comment. We have modified the sentences accordingly.

- L630: at the end of sentence: "via evapotranspiration".

Response: Corrected.

- L635: behaviour

Response: Corrected.

- L639: 'remarkable differences' is an odd phrase. Be specific. Also if you mean large, this is a different narrative to Section 3.

Response: Thank you for your valuable comment. We have revised the sentence, as: "However, our results also show that there are **obvious** discrepancies between the ecosystem COS flux simulations and observations, and that discrepancies cannot be effectively reduced by the adjustment by the photosynthesis related parameters duo to the simplification of BEPS for nighttime stomatal conductance modelling" (Line 596-599). Moreover, we have also deleted the sentence in section 3 you mentioned: "**Results show that the prior simulations can**

"

- L641: BEPS

Response: Corrected.

- L679: move reference to end of sentence.

Response: Done.

- Some of the discussion in Section 4.5 is a bit wordy (mainly last paragraph). Below are a few instances of trying to improve readability and flow.

○ Final paragraph, sentence 2: "As the nighttime COS plant uptake is driven by stomatal conductance (Kooijmans et al., 2021), gn, nighttime COS fluxes can therefore be used to test the capability of BEPS to model $gn$.

Response: Thank you for your comment. In this manuscript, $g_n$ refers specifically to the **nighttime** stomatal conductance, which is slightly different with the stomatal conductance ($g_s$) . Therefore, we retained the original sentence, but deleted "the" in response to your comment: "As the nighttime COS plant uptake is driven by stomatal conductance (Kooijmans et al., 2021),  nighttime COS fluxes can therefore be used to test the accuracy of the model settings for nighttime stomatal conductance ($g_n$)." (Line 652-654)

○ L699: space between 1 and mm.

Response: Corrected.

○ Final paragraph sentence 5: "Similar findings by Resco De Dios et al. (2019), showed that the median $gn$ in the global dataset was 40 mmol m−2 s−1."

Response: Corrected.

○ Final paragraph sentence 8: "As different enzymes have different physiological characteristics, Cho et al. (2023) proposed a new temperature function for the CA enzyme and showcased the considerable difference in temperature response of enzymatic activities of CA and RuBisCo, which provided valuable insights into the modelling and assimilation of COS."

Response: Corrected.

○ L707: CA

Response: Corrected.

○ L709: N = nitrogen?

Response: Thanks for your comment. We have replaced "N" with "nitrogen".

○ L710: in -> by

Response: Corrected.

○ Final paragraph final sentence: "Therefore, using the global microbial C biomass, soil N content and MAP datasets, the relationships between these variables, and the associated COS exchange processes, it is to be expected that a more accurate modelling of terrestrial ecosystem COS fluxes could be achieved, further increasing our understanding of the global COS budget and facilitate the assimilation of COS fluxes."

Response: Thanks for your valuable comment. We have revised the sentence accordingly.

- L724-726: Perhaps: "Fourteen twin experiments, thirteen single-site experiments and one two-site experiment covering the period from 2012 to 2017, were conducted to investigate the capability of NUCAS to assimilate COS fluxes and optimize output parameters and variables. COS flux observations from a range of ecosystems were used, including four PFTs and three soil textures."

Response: Thanks for your comment. We have revised the sentence accordingly.

- L729: COS fluxes

Response: Corrected.

- L749: "throughout the plant function type" -> "for different PFTs".

Response: Thank you for your valuable comment. But for the convenience of our readers, especially those who don't have the time to read the whole manuscript, here we prefer to keep the original sentences without abbreviations.

- L753-754: "However, the 'equifinality' can be avoided by imposing additional observational constraints (Beven, 2006)." Such as? i .e., 'in this instance, we refer to the calculation and assimilation of multiple datasets, other than just COS fluxes.'

Response: Thanks for your comment. The assimilation of multiple datasets, other than just COS fluxes, is just what we want to express.

**Technical Notes:**

- Be consistent with the following phrases single-site, two-site, AmeriFlux, process-based, etc..in terms of the use of hyphen and capitalisation. Please check other potential sources of inconsistency.

Response: Thanks for your valuable comment. We have carefully rechecked the manuscript to avoid inconsistencies terms of the use of hyphen and capitalisation.

- Use non-line-breaking hyphen where possible. This way be resolved in the editing by EGU, but for future reference to avoid a hyphenated phrase, or unit, breaking a line, use ctrl+shit+-.

Response: Thank you for this extremely detailed technical note. We have changed the line-breaking hyphen as you suggested.

- % symbols should be immediately adjacent to values, not with a space

Response: Thank you for your comment. We have corrected the use of "%" and spaces.

**References**

[revised manuscript text omitted]